# Mortality caused by tropical cyclones in the United States

Rachel Young[1,2,3,6] & Solomon Hsiang[3,4,5,6] ✉

Natural disasters trigger complex chains of events within human societies[1]. Immediate deaths and damage are directly observed after a disaster and are widely studied, but delayed downstream outcomes, indirectly caused by the disaster, are difficult to trace back to the initial event[1,2]. Tropical cyclones (TCs)—that is, hurricanes and tropical storms—are widespread globally and have lasting economic impacts[3–5], but their full health impact remains unknown. Here we conduct a large-scale evaluation of long-term effects of TCs on human mortality in the contiguous United States (CONUS) for all TCs between 1930 and 2015. We observe a robust increase in excess mortality that persists for 15 years after each geophysical event. We estimate that the average TC generates 7,000–11,000 excess deaths, exceeding the average of 24 immediate deaths reported in government statistics[6,7]. Tracking the effects of 501 historical storms, we compute that the TC climate of CONUS imposes an undocumented mortality burden that explains a substantial fraction of the higher mortality rates along the Atlantic coast and is equal to roughly 3.2–5.1% of all deaths. These findings suggest that the TC climate, previously thought to be unimportant for broader public health outcomes, is a meaningful underlying driver for the distribution of mortality risk in CONUS, especially among infants (less than 1 year of age), people 1–44 years of age, and the Black population. Understanding why TCs induce this excess mortality is likely to yield substantial health benefits.

Despite attracting widespread cultural, scientific and policy attention, the full impacts of natural disasters on society are not well understood. In particular, effects on human health are challenging to disentangle from numerous other factors that also influence health outcomes, such as behaviour, healthcare systems and pollution. Because of this complexity, many approaches to measuring the mortality impact of disasters focus narrowly on enumerating cases where a disaster is the most immediate and obvious direct cause of death, such as drownings in flood waters. Yet it has been widely hypothesized[1–5,8,9] that tracking only these 'direct deaths' might misrepresent the total mortality that results from disasters, since disasters trigger complex cascades of events that ultimately may cause additional future mortality. To our knowledge, this full excess mortality effect has never been characterized for any class of disaster at population scale, accounting for deaths that may be delayed relative to the physical disaster but are nonetheless traceable to those events. By extension, the full health burden of environments that are chronically disaster-prone also remains poorly understood.

Here we develop a long-run estimate for the overall effect of individual TCs (which include hurricanes and tropical storms) and the TC climate on all causes of mortality across all populations within CONUS. TCs are a frequent hazard for CONUS, causing damage to infrastructure, homes and businesses[10,11]; population relocation[12]; social and economic disruptions[2,3,5,13]; ecological changes[14,15]; reduced access to basic services[16]; increased pollution[7]; crop damage[7]; insurance payouts[17]; and political actions[18]. These and other impacts of TCs might affect human health through complex chains of events that separate the cause (cyclone) from the delayed effect (mortality) so much that affected individuals are themselves unaware that a TC influenced their own health outcome. For example, individuals may use retirement savings to repair damage, reducing future healthcare spending to compensate; family members might move away, removing critical support when something unexpected occurs years later; or public budgets may change to meet the immediate post-TC needs of a community, reducing investments that would otherwise support long-run health. Prior studies of sub-populations[19] for specific events[20,21] or shorter windows of time[2,8,9] suggest that these and other pathways could substantially influence post-TC mortality, but the full long-term impacts of all indirect pathways for all storms across an entire population remains unknown.

In an impossible hypothetical (and unethical) experiment measuring the impact of TCs on mortality, two initially identical populations would be compared after one was 'treated' with a TC and the other was not ('control'). If mortality later increased in the treated population relative to the control, then we would infer that the TC caused this delayed mortality. Here we study an alternative natural experiment, which approximates experimental conditions, in which states in CONUS are randomly hit by TCs over time. Specifically, we study how mortality rates within a state change after the state is hit by a TC. Thus, conceptually, each state before a TC serves as the control for the same state after

[1]Department of Agricultural & Resource Economics, University of California, Berkeley, Berkeley, CA, USA. [2]School of Public and International Affairs, Princeton University, Princeton, NJ, USA. [3]Global Policy Laboratory, University of California, Berkeley, Berkeley, CA, USA. [4]Global Policy Laboratory, Stanford Doerr School of Sustainability, Stanford University, Stanford, CA, USA. [5]National Bureau of Economic Research, Cambridge, MA, USA. [6]These authors contributed equally: Rachel Young, Solomon Hsiang. ✉e-mail: solhsiang@stanford.edu

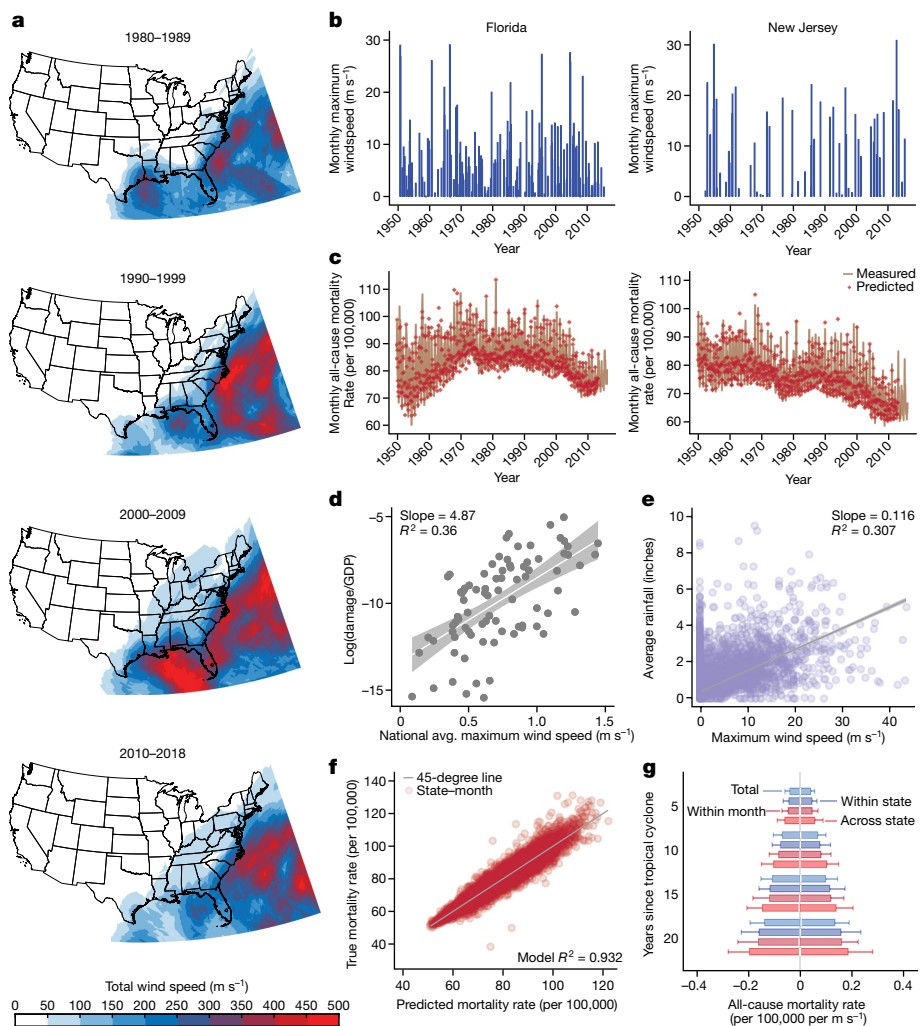

**Fig. 1 | Example data and model validation. a**, Example decadal totals of cumulative monthly TC maximum wind speed computed using LICRICE (Supplementary Fig. 1 shows all decades). **b,c**, Example monthly TC 'impulses' from Florida and New Jersey (**b**) and corresponding monthly all-cause mortality (lines) (**c**). Model predictions for monthly mortality are overlaid as red dots (Extended Data Figs. 1–3 show all states). **d**, log(TC damage/GDP) per storm from ref. 51 versus national average (avg.) maximum wind speed computed using LICRICE (1950–2005, $n = 89$, slope = 4.87, $R^2 = 0.36$; shaded region represents 95% confidence interval). GDP, gross domestic product. **e**, Correlation between state-by-storm average rainfall from NOAA station data (a limited sample) and maximum wind speed computed using LICRICE (263 storms, $n = 12,889$, slope = 0.116, $R^2 = 0.307$). **f**, Comparison of model predictions against observed mortality for all state-by-month observations ($n = 27,216$, $R^2 = 0.93$). **g**, Verifying that the model produces unbiased estimate of TC impacts in four randomization-based placebo experiments (negative exposure controls; see Supplementary Fig. 7) described in text. Distributions are estimated cumulative effects (0 is true) for 5, 10, 15 and 20 years after incidence of a shuffled cyclone event across 1,000 iterations of randomization and model re-estimation. Boxes delineate 25th and 75th percentiles, whiskers span minimum and maximum values. Extended Data Fig. 5 shows all lags.

it receives TC treatment[22]. The exact timing, location and intensity of TC treatments—directed by stochastic and chaotic oceanic and atmospheric processes—are randomly assigned to populations[23]. If mortality systematically rises after TCs, conditional on other factors, we may infer that it was plausibly caused by the TCs[24]. This 'reduced-form' approach captures all potential pathways through which TCs influence mortality, without requiring that every channel is modelled explicitly (Methods).

We study the impact on all-mortality of all 501 TCs that affected the CONUS coastline during 1930–2015 (Fig. 1a, Extended Data Fig. 1 and Supplementary Fig. 1). We estimate changes in monthly state mortality rates (1950–2015) during the 240 months (20 years) after each TC, partially motivated by analyses indicating that some economic effects of TCs persist for more than 15 years[3,20]. We hypothesize that delayed effects exist and design an econometric approach that should reveal them, if they are present.

The potentially long duration of TC impacts poses methodological challenges to measuring the impact of individual TCs, because mortality rates may still be responding to earlier TCs when later TC strikes

occur. In this case, mortality time series would exhibit the superposition of multiple overlapping signals, each a response to individual TC strikes that occurred over a sequence of years. Our solution to separating these overlapping signals is to deconvolve mortality time series, empirically recovering the characteristic mortality impulse-response function that results from a single TC 'impulse'. Deconvolution is an established signal-processing technique[25] used to analyse time-series data in many contexts, including astronomy[26] and economics[27]. Here, short-lived TC events are represented as impulses (Dirac delta functions) scaled by their physical intensity (Fig. 1b, Extended Data Fig. 1 and Supplementary Fig. 1). We examine, via deconvolution (Extended Data Fig. 2), how mortality rates (Fig. 1c and Extended Data Fig. 3) systematically change following a TC while accounting for the effects of previous TCs (Methods).

We rebuild the sequence of physical TC events that each state experiences in each month for 1930–2015 using the Limited Information Cyclone Reconstruction and Integration for Climate and Economics (LICRICE) model[28] (Methods). This results in estimates of maximum

wind speed experienced at each 0.125° × 0.125° ground pixel during each storm (Fig. 1a and Supplementary Fig. 1), a measure of TC incidence that has been shown to strongly predict physical damage and other economic and social impacts globally[2,3,28,29]. We show that this measure predicts direct normalized storm-level damages in CONUS (Fig. 1d and Supplementary Fig. 2) and is correlated with TC rainfall (Fig. 1e and Supplementary Fig. 3). Ground-level wind speed is not the only dimension of TC incidence that affects human outcomes (for example, flooding also causes direct deaths[30]), but, to our knowledge, it is the only metric that can be consistently reconstructed for all storms throughout our entire sample. We therefore consider wind speed a meaningful, albeit imperfect, proxy measure for the physical aspects of storm intensity that impact all-cause mortality. Wind speed is averaged over state land area for each month over 86 years ($n$ = 37,152 state-by-month observations, mean = 0.39 ms$^{-1}$, 95th percentile = 2.33 ms$^{-1}$, maximum = 47.6 ms$^{-1}$; Supplementary Fig. 4) and linked to monthly state mortality records from the US Center for Disease Control and Prevention (CDC) Mortality Statistics of the United States Annual Volumes, Multiple Cause of Death (MCOD) files, and Underlying Cause of Death database[31] (Fig. 1c and Extended Data Fig. 3). Finer resolution mortality data were unavailable for the extended period that we studied.

To measure the effect of TCs on mortality, we deconvolve mortality rates as a series of responses to other factors and continuous TC events that have effects that may unfold over a period of up to 240 months. To identify the effect of TCs on mortality separately from other known and unknown factors that affect mortality across locations and over time, our econometric analysis non-parametrically accounts for: (1) all average differences across states—including culture, state healthcare policy and geographic factors; (2) all average state-specific seasonal patterns—including environmental changes (for example, sunlight) and annual events (for example, holidays); (3) nonlinear state-specific trends—which include changes in demographics, healthcare access, environmental pollution and economic conditions; (4) national month-of-sample effects—which capture all nationally coherent social, economic, political and epidemiological changes (for example, influenza outbreaks[32]); and (5) state-by-month-specific linear trends—capturing policies, technology or climate changes that affect mortality in particular months within a state more than other months (for example, anti-lock brakes reducing winter motor vehicle mortality only in cold states) (Extended Data Fig. 4 and Supplementary Fig. 5). Our analysis also accounts for the documented state-specific nonlinear effect of temperature on mortality[33–35] (Supplementary Fig. 6), since temperature and TC activity are correlated in the North Atlantic[36] (Methods).

We find that our econometric model is the most parsimonious model that adequately captures the rich historical variation in mortality that we observe in data (Fig. 1c, Extended Data Figs. 3 and 4 and Methods). Our model exhibits skill ($R^2$ = 0.932) in predicting month-by-state mortality (red markers in Fig. 1c,f and Extended Data Fig. 3) and, critically, produces unbiased estimates for randomized TC data. In four placebo experiments, we search for (null) associations between true mortality data and randomly reshuffled placebo versions of our TC data that cannot be associated with real world outcomes (Supplementary Fig. 7 and Methods). Our model does not generate spurious associations between mortality and (1) the unconditional distribution of TC events (we shuffle all TC observations; denoted 'total randomization'); (2) average cross-sectional patterns across states (we shuffle the sequence of TCs each state experiences; denoted 'within state' randomization); (3) secular national trends (we shuffle the TC experience across states within each month of sample; denoted 'within month' randomization); or (4) temporal-trends within states (we block-shuffle complete TC time series across states, keeping the sequence of storms as blocks; denoted 'across state' randomization). Our model correctly indicates no associations between mortality and TCs in these experiments (Fig. 1g

and Extended Data Fig. 5), but alternative models missing any of the elements 1–5 above fail one or more of these tests (Methods).

## The impact of individual TCs

We find that state-level all-cause mortality is systematically increased for the 172 months (14.3 years) following a TC (Fig. 2a). In the month of TC landfall, we estimate that monthly mortality rates increase by 0.033 (±0.012) deaths per 100,000 population per ms$^{-1}$ of state-level wind speed incidence ($t$(23,730) = 2.78, $P$ < 0.05). The rise and fall in mortality is well approximated by a quadratic function of time (red line in Fig. 2a) that peaks at 0.042 deaths per 100,000 after 68.6 months after landfall. We compute the cumulative excess mortality following a single TC (summing over months; Fig. 2b), a more intuitive and policy-relevant description of these impacts. For each 1 ms$^{-1}$ of state-level wind speed incidence we estimate an average cumulative 5.37 (±1.8) excess deaths per 100,000 after 172 months ($t$(23,730) = 2.94, $P$ = 0.0038). The average state-level TC event results in state-level winds of 6.9 ms$^{-1}$, implying an average cumulative mortality of 37.05 (±12.4) per 100,000 after 172 months. This estimate diverges from corresponding results in all four randomization-based placebo tests (orange dashed lines in Fig. 2b) and is highly statistically significant ($P$ < 0.012) when using these randomizations in permutation tests (Extended Data Fig. 5 and Methods). This result is robust to using population-based regression weights, count-based models, or accounting for region-by-month-of-sample shocks (Extended Data Fig. 6a). We also test whether this response has changed over time—perhaps owing to changes in technology or policy—but find no evidence that earlier storms had an impact different from later storms (Extended Data Fig. 6b). This indicates that no adaptations[29,31,33–35] over this period have reduced the mortality impact of TCs in CONUS.

## Direct versus indirect deaths

These results indicate that a large number of previously uncounted deaths in CONUS can be traced to TC events. These indirect deaths are deaths that occur earlier than would be expected in the absence of the TC. They are likely to be caused by complex sequences of events that follow in the wake of TCs (such as economic loss[37] or lack of healthcare access[21]) and thus differ from official counts of direct deaths that occur during the short-lived geophysical event and outnumber official direct deaths by orders of magnitude. The US National Oceanic and Atmospheric Administration (NOAA) report[6,7] that the TCs that we study (501 events) directly killed 24 individuals on average (22 without Hurricane Katrina). By contrast, we estimate that the average TC indirectly accelerated the death of roughly 7,170–11,430 individuals, depending on model specification (Extended Data Table 1).

## Mortality by age

We evaluate the effects of TCs on four age categories that are available for the entire sample (Fig. 2c). For the same TC event, cumulative excess mortality risk for 172 months is greatest for infants less than 1 year of age (called 'infants' herein), at 49.8 deaths per 100,000 per ms$^{-1}$ (±11.3, $t$(20,319) = 4.41, $P$ < 0.0001), and second largest for people 65 years of age and older, at 22.8 (±10.0, $t$(20,862) = 2.28, $P$ < 0.05). For people 1–44 and 45–66 years of age, it is smaller, at 2.49 (±0.53, $t$(20,821) = 4.70, $P$ < 0.0001) and 3.50 (±2.4, $t$(20,862) = 1.46, $P$ > 0.05) per 100,000 per ms$^{-1}$. However, because of the age distribution, the group 65 years of age and older has the largest number of total excess deaths (46%), with ages 45–64 (8%), 1–44 (32%) and infants (14%) years together making up the remaining deaths (Extended Data Fig. 7a). We estimate that infants are the most vulnerable group by risk (16 times the risk at age 1–44), but the small population of infants mean they constitute only one-seventh of deaths. Of these infant deaths, 99% occur more than 21 months after the TC, indicating that the infants were not conceived prior to landfall (consistent with ref. 2). This suggests that cascades of

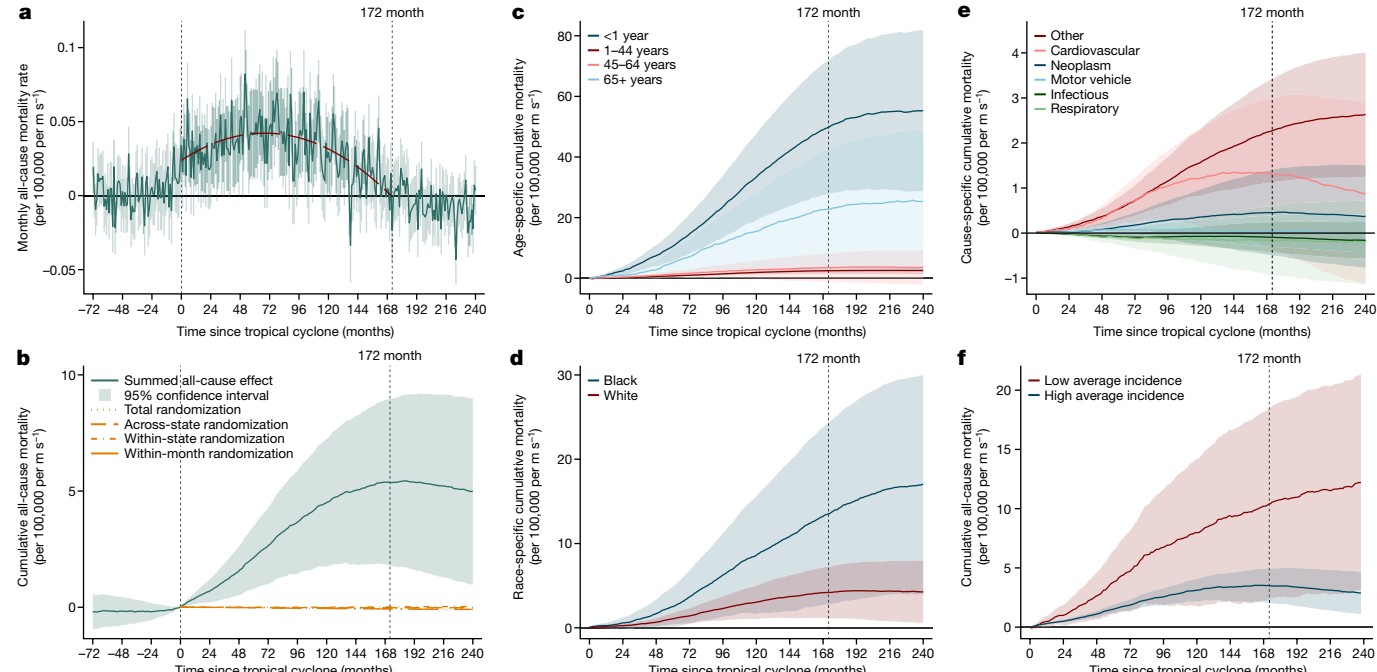

**Fig. 2 | Excess mortality following a TC.** Impulse response of mortality rates per 1 ms$^{-1}$ of state-level wind speed incidence. **a**, Estimated excess all-cause mortality by month before and after a TC. Green line shows monthly coefficients (equation (8)); the shaded area represents 95% confidence intervals. The red dashed line shows a quadratic fit to lag coefficients 0–172: excess mortality = $0.0237 + 0.000535x - 0.0000039x^2$, where $x$ is the time since landfall in months. **b**–**f**, Cumulative excess mortality and 95% confidence intervals (accounting for covariances) from estimated impulse responses after a TC. **b**, Green line shows cumulative effect of coefficients in **a**, and the shaded area represents the 95% confidence interval. Leads (negative months) and orange lines are negative exposure controls. Orange dashed lines are means in the four randomization-based placebo experiments. **c**–**f**, Same as **b**, but estimated separately by age (**c**), race (**d**), official cause of death (**e**) and average TC incidence (**f**). **f**, Low average incidence represents the quartile of states with lowest non-zero average TC incidence; high average incidence represents the top three quartiles. Joint significance test of equality between high and low incidence responses for months 1–240 has $P = 0.0082$ (two-sided $F$-test; Extended Data Fig. 6c shows quartiles separately).

indirect effects following TCs, rather than personal direct exposure to the physical event, generate this mortality.

## Mortality by race

There is growing concern that minority populations may suffer greater harm from environmental conditions[38]. A central hypothesis argues that these populations suffer more than other groups when both experience the same physical event, although quantified evidence for such unequal vulnerability is mixed and inconclusive[39]. We find that when Black and white populations are exposed to the same TC event, Black individuals experience cumulative excess mortality risk of 13.53 (±5.51) per 100,000 per ms$^{-1}$ ($t(19,992) = 2.46$, $P < 0.05$) over the following 172 months, whereas white individuals experience excess risk of 4.19 (±1.56) per 100,000 per ms$^{-1}$ ($t(2,083) = 2.68$, $P < 0.05$) (Fig. 2d). However, because the white population that is exposed to TCs is larger, we estimate that 66% of cumulative excess deaths occur among white individuals, compared to 34% among Black individuals (Extended Data Fig. 7b). We cannot distinguish other race groups owing to limitations in our data (Methods).

## Mortality by cause

The official cause of TC-related excess deaths is almost never recorded as a TC. Examining official causes of death from the CDC Underlying Cause of Death database (Fig. 2e), we find that most TC-related excess deaths (58.9%, $2.27 \pm 0.59$ per 100,000 per ms$^{-1}$) result from 'other' causes, a nonspecific category that includes diabetes, suicide, sudden infant death syndrome and other causes that are not individually recorded. Cardiovascular disease is the second largest cause of TC-related excess deaths (36.0%, $1.30 \pm 0.86$ per 100,000 per ms$^{-1}$) and neoplasms (cancer) is third (11.6%, $0.46 \pm 0.48$ per 100,000 per ms$^{-1}$), consistent with some evidence of stress from extreme weather affecting

long-run health[40]. Infectious diseases, respiratory diseases and motor vehicle accidents are not linked to TCs. Future work should investigate the role of specific causes within the 'other' causes category.

## Mortality by climatological risk

Defensive adaptations may cause TC incidence to have less impact on populations that are frequently affected by TCs compared with those that are infrequently affected[29,33–35]. We study this by stratifying states on the basis of their TC climates, measured by average TC incidence[2,3,29]. States that experience TCs least infrequently (quartile 1) exhibit higher vulnerability to TCs—10.4 deaths per 100,000 per ms$^{-1}$ (±4.11, $t(24,821) = 2.53$, $P < 0.05$) 172 months after a TC, compared to all other states (quartiles 2–4), 3.49 deaths per 100,000 per ms$^{-1}$ (±0.74, $t(24,821) = 4.71$, $P < 0.0001$) (Fig. 2f; vulnerability among upper quartiles are statistically indistinguishable; Extended Data Fig. 6c and Methods). Thus, in states where TCs are uncommon, the mortality impact of physically similar TC events is around 2.8 times greater than in states where TCs are a regular occurrence. This finding is consistent with the hypothesis that populations adapt to their climate, somewhat reducing its effects. However, the levelling-off of vulnerability and adaptation among upper quartiles indicates a limit to how effective these adaptations are in practice (similar levelling-off was observed in refs. 2,3). We additionally evaluate whether the spatial distribution of populations within states alters the mortality response to TCs and find no evidence of such within-state adaptation (Methods).

## Mortality burden of the TC climate

We estimate the impact of the TC climate by computing the expected mortality that resulted from all TC events in our sample, accounting

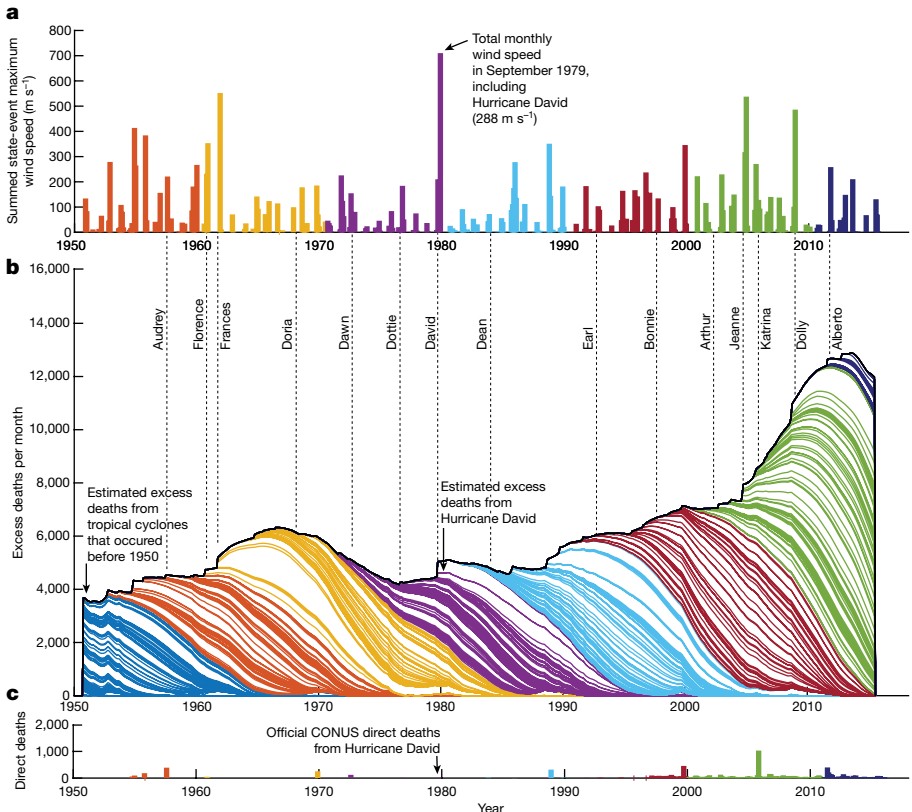

**Fig. 3 | Estimated mortality burden from 501 TCs affecting CONUS during 1930–2015. a**, Total incidence of TCs on CONUS by month. Bar height is sum of average maximum wind speeds for all state-by-storm events. Colours correspond to decades. **b**, Stacked overlapping excess mortality responses to each storm for all of CONUS. Each storm response aggregates state-level responses nationally, accounting for state-level population and adaptation. Outline colours correspond to the decade when the TC occurred. The upper envelope is the total estimated mortality burden for CONUS resulting from all TCs occurring during the prior 172 months (see Supplementary Fig. 3). **c**, Official deaths directly resulting from TCs for each month according to NOAA National Hurricane Center and NOAA National Weather Service[6,7]. The *y*-axis scale is the same for **b** and **c**.

for TC intensity and evolving demographics of affected populations. Individual storms increase state mortality modestly per month (13 deaths on average). However, owing to the long duration (172 months) of elevated mortality after each event and the large number of TC-by-state events (2,748 events between 1930 and 2015), CONUS mortality is always simultaneously affected by the overlapping impacts of numerous prior TCs across many locations (149 in an average month during 1950–2015). Figure 3b illustrates the superposition of expected monthly mortality flows resulting from all TCs affecting CONUS (Fig. 3a). Individual storms contribute only 0.05% of all nation-wide TC-related mortality in a given month on average and the growth and subsidence of post-TC mortality is gradual and without spikes. The flow of total CONUS TC-related mortality appears relatively stable until 2001, when TC-driven mortality grew rapidly—a change that we study below.

We estimate that all TCs combined produced between 4,600 and 7,300 excess deaths per month in CONUS during 1950–2015 (Fig. 3b and Extended Data Table 1). The 501 TCs affecting CONUS between 1930 and 2015 generated a total of 3.6–5.7 million excess deaths (range based on alternative model specifications). A TC of average intensity is linked to roughly 7,170–11,430 deaths. This burden is 300–480 times greater than government (NOAA) estimates of 24 deaths per storm on average (22 without Hurricane Katrina) and 11,937 total TC deaths during 1950–2015 (Fig. 3c). We estimate that the TC climate contributes to 1.9–3.1 excess deaths per 100,000 annually, equal to 3.2–5.1% of all deaths across CONUS. This burden is distributed unevenly across geography, age and race, and explains some patterns of mortality risk across and within these strata.

### Burden across states
Accounting for all TCs in our data, underlying age distributions, demographic changes over time, historical experience with TCs (including associated adaptations) and nonlinear models of TC impact (Methods), we find that southeastern states have the highest proportion of total deaths attributable to TCs (Fig. 4a). For example, 13% of deaths in Florida, 11% of deaths in North Carolina, 9% of deaths in South Carolina and 8% of deaths in Louisiana during this period can be traced to their TC climate.

### Burden by age
We estimate that the TC climate contributes to a large overall fraction of mortality for individuals less than 45 years of age in CONUS. TC-related mortality risk is highest for infants and lowest for people 1–44 years of age (Fig. 2c and Extended Data Table 2). Because both groups have low overall baseline mortality risk, post-TC excess mortality translates into a substantial fraction of overall mortality, which explains 25% and 15% of mortality for infants and for people 1–44 years of age, respectively (Fig. 4b). Total TC-related deaths is largest for people 65 years of age and up, but only accounts for around 3.5% of all 65+ deaths.

### Burden by race
Black populations bear a relatively larger TC mortality burden than white populations, owing to their greater vulnerability (Fig. 2d) and their spatial distribution, which is denser in the southeast where TCs are common[41]. We estimate that, on average, 47,444 and 37,402 excess deaths per year among white and Black populations, respectively, are

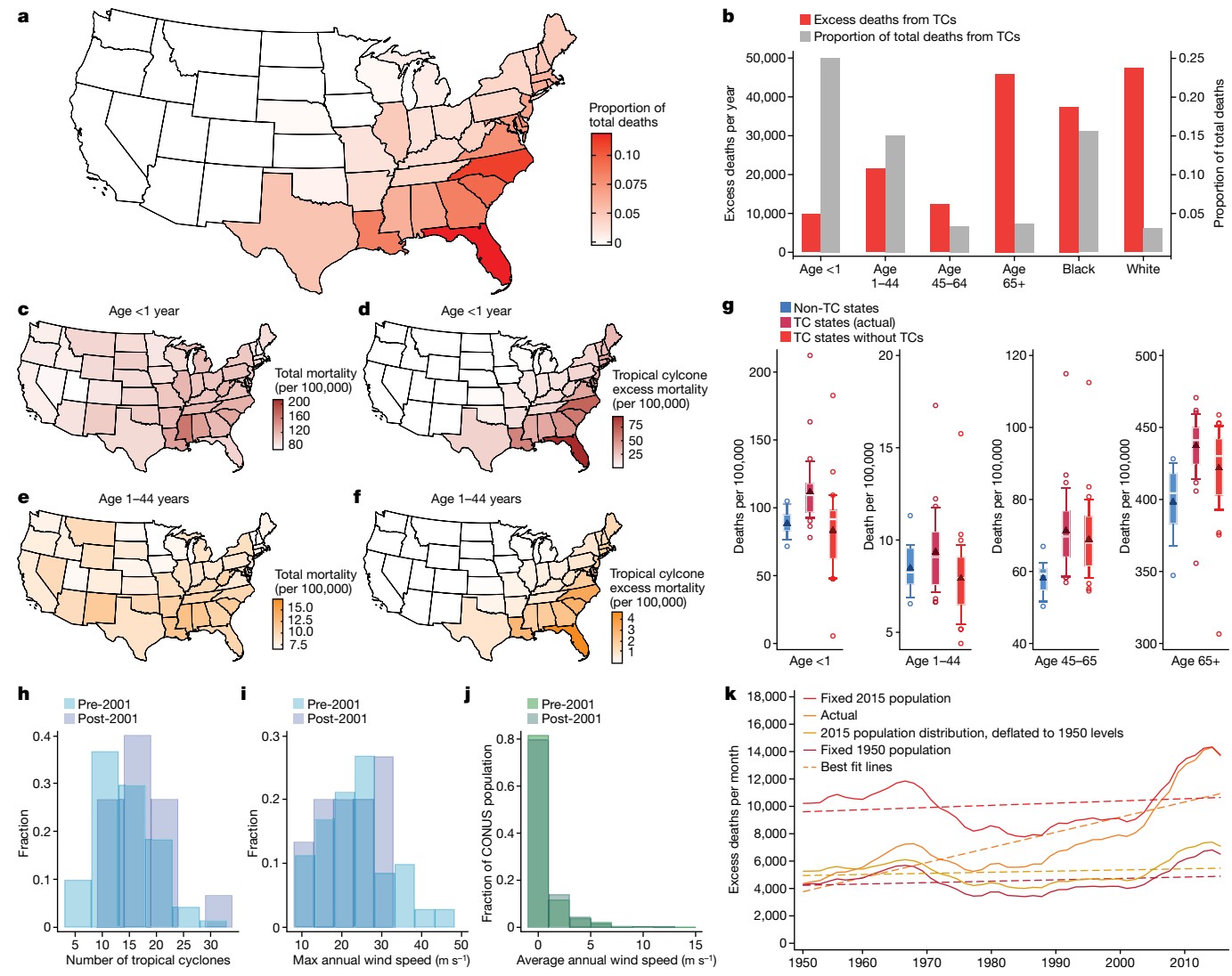

**Fig. 4 | Geographic, demographic and temporal patterns in the mortality burden of the TC climate. a**, Estimated average excess mortality traceable to the TC climate by state during 1950–2015 (all maps are from ref. 52). **b**, Estimated average excess total TC-related deaths per year by demographic group (left axis) and as a fraction of all deaths in each group (right axis). **c**, All-cause mortality rate for infants (age <1 year) by state during 1960–2015. **d**, Estimated all-cause mortality for infants traceable to the TC climate. **e,f**, Same as **c,d**, but for people 1–44 years of age. **g**, Distributions of average state mortality by age group for states that never experience a TC (non-TC states, *n* = 14) and the states that experience non-zero TC incidence (TC states (actual), *n* = 35). Red distributions

(TC states without TCs, *n* = 35) are estimated counterfactual distributions for TC states after mortality burdens from TC climates are removed. In box plots, boxes represent inner quartiles, white bars show the median, triangles indicate the mean, whiskers extend to 10th and 90th percentiles and circles show outliers. **h**, Distribution of TC event counts each year before and after 2001. **i,j**, Distribution of maximum state-level wind speed (**i**) and average TC incidence for CONUS population (**j**) each year before and after 2001. **k**, Actual and simulated trends in the mortality burden of TCs in CONUS. Simulations assume counterfactual population distributions but use actual TC incidence.

traceable to prior TCs. These excess deaths constitute 3.1% and 15.6% of all white and Black mortality in CONUS, respectively (Fig. 4b and Extended Data Table 2).

### Geography of age-specific burden

Overall geographic patterns of mortality for the populations of infants and people 1–44 years of age (Fig. 4c,e) are consistent with the uneven geographic incidence of TCs and their substantial impact on mortality for these ages (Fig. 4d,f). States exposed to TCs ('TC states') have higher mortality on average for all ages compared to states that are never exposed ('non-TC states') (Fig. 4g, blue versus maroon distributions) owing to many factors (such as lifestyle and diet), but TCs are likely to explain some of this gap.

Removing TC-related deaths (red distributions) aligns the distributions of mortality rates between TC and non-TC states for infants and

people 1–44 years of age, suggesting that TCs may have an important role in explaining the difference in these outcomes for these regions. For infants, the average mortality rate for TC states declines from 112 to 83 deaths per 100,000 in TC states when excess TC deaths are removed, more closely matching the 88 deaths per 100,000 in non-TC states, whereas for ages 1–44, the average mortality rate for TC states declines from 9 to 8 deaths per 100,000 in TC states when excess TC deaths are removed, more closely matching the 8 deaths per 100,000 in non-TC states (Fig. 4g). The spatial pattern of the TC climate across states appears to explain much of the difference in mortality for those 0–44 years of age across these regions. We note this result is not a mechanical outcome of our econometric analysis, since our estimates net out average mortality rates in each state and depend only on the timing of TCs within each state. Removing deaths attributable to TCs reduces, but does not close, the gap between TC and non-TC states for ages 45–64

and 65 and over, because TC-related deaths are a smaller fraction of total deaths for these age groups.

### Accelerating trend after 2001

Our reconstruction of TC-related mortality indicates a positive average trend of +9.2 TC-related deaths per month, with a notable increase to +43.3 deaths per month after 2001 (Fig. 3b). This acceleration results from more frequent TCs after 2001 (17 yr$^{-1}$) than before 2001 (14 yr$^{-1}$; Fig. 4h), but these storms were not more intense on average. The maximum intensity of state-level incidence within landfalling storms actually declined from 23.6 ms$^{-1}$ before 2001 to 21.4 ms$^{-1}$ after 2001 (Fig. 4i). The net effect of more frequent storms with slightly weaker average intensity was to increase the intensity of average TC incidence experienced by CONUS populations from 0.125 ms$^{-1}$ to 0.143 ms$^{-1}$ after 2001 (Fig. 4j).

### Long-term trends

In addition to climatological drivers, changes in population distributions have altered exposure to TCs. For example, it is widely argued that increasing TC damages are driven by migration towards risky locations[42]. We therefore decompose the long-term trend in TC-related excess mortality into contributions from climatological factors, shifting spatial distribution of population towards coasts, and demographic trends.

To decompose the long-term trend in the excess mortality, we simulate outcomes while assuming counterfactual population patterns. First, we simulate excess mortality as in Fig. 3, but holding populations fixed so that all changes are driven by changes in the distribution of TC events (Fig. 4k). Fixing populations at 1950 or 2015 distributions, climatological factors contribute +0.848 or +1.34 TC-related deaths per month, respectively, explaining 12% of the estimated trend (red and maroon lines). This is consistent with the generally small climatological shifts we observe before 2001.

Second, we examine whether the evolving spatial arrangement of populations drives the long-term trend, since land development[43] and public insurance[5,44] may have incentivized populations to shift towards risky coastal locations. To test this, we simulate mortality using the relative spatial distribution from 2015, but rescaled to equal the 1950 total population (yellow line). This increases average monthly excess mortality by 653 (14.3%) relative to a fixed 1950 population distribution, indicating that population shifts towards coastal states contributes +0.69 TC-related deaths per month (7.5% of the trend). The small size of this effect is consistent with the limited effect of population distributions within states on the overall mortality impact TCs (Extended Data Fig. 6d).

The remaining 80.5% of the trend in TC-related mortality results from the growth and aging of the CONUS population. In 1950, 131 million individuals lived in TC states, increasing by 85.4% to 243 million by 2015 (the population of non-TC states increased by 255.7%). Thus, we estimate that +7.4 additional TC-related deaths per month results from demographic trends.

## Discussion

In our evaluation of long-run population-wide excess mortality resulting from TCs, we find that indirect deaths triggered by TCs in the 15 years after landfall are substantially higher than official counts of mortality occurring during the geophysical event. Our estimates indicate that the current TC climate of CONUS imposes an annual burden of around 55,280–88,080 excess deaths. During the period of study, we estimate that TCs contributed to more deaths in CONUS (3.6–5.2 million) than all motor vehicle accidents (2.0 million), infectious diseases (1.9 million) or US battle deaths in wars (1.3 million). These findings point to TCs as an important and understudied contributor to health in the United States, particularly for young or Black populations.

We acknowledge that the large difference between the indirect excess mortality burden that we compute and official counts of direct TC deaths is surprising. Indeed, we initially believed that these findings resulted from calculation errors, as the absence of any previous comparable analysis made it difficult to construct an informed prior for these estimates. However, these findings are consistent with the growing literature indicating that climatic conditions generally[45], and TCs specifically[3,40], have larger and more enduring effects than previously recognized. For example, high temperatures cause[31,33–35] roughly 150,910 deaths annually in CONUS (7.0% of total mortality); TCs trigger substantial long-term economic losses at the individual household[2,20] and macroeconomic level[3,5]; and one study[21] of Hurricane Maria estimated that there were 3,000 excess deaths (within 3 months) beyond official counts. Furthermore, our finding of uneven TC impacts across Black and white populations is consistent with prior descriptive results that Black populations tend to be more vulnerable to disasters[46].

We find no evidence of autonomous adaptation over time to TCs (Extended Data Fig. 6b), contrasting with the finding that populations in CONUS adapted to reduce heat-related mortality during this period[31], probably via adoption of air conditioning. We hypothesize that an inability to observe the indirect excess deaths from TCs prevented analogous adaptations to TCs.

Our analysis identifies that TCs cause excess mortality, but it does not identify the underlying mechanisms. This situation is similar to other early statistical studies that linked hazards (such as tobacco and asbestos) to health outcomes before the underlying mechanisms were understood[47]. Future work may disentangle the channels that generate these effects.

We propose five hypotheses that might explain why TCs trigger excess mortality. (1) Economic disruption[2–5] from TCs might change household economic decisions, eventually translating into adverse health outcomes[2]. For example, job loss[5,16] might affect health insurance, or retirement savings could be drawn down to repair property damage, both of which could reduce future spending on healthcare. (2) Social network changes after TCs could affect future health. For example, out-migration of working-age individuals[12,20] could alter social support for older dependents that remain behind. (3) Fiscal adjustments by state or local governments in response to TCs[5,16] may impact future health outcomes. For example, restructuring budgets to support recovery might reduce spending on healthcare infrastructure[16]. (4) Changes in the natural environment after TCs could impact health. For example, ecological changes could redistribute disease vectors or flooding may expose populations to harmful chemicals[48,49]. (5) Heightened physical and mental stress from experiencing TCs may alter long-run health[40]. Future work that determines whether these or other channels generate post-TC mortality will enable the development of effective policy interventions.

Our analysis has several limitations. First, we measure TC incidence by reconstructing wind speeds and do not model storm surges, rainfall or flooding. The effects of these variables are partially represented in our estimates by proxy based on their correlation with maximum winds (Fig. 1e and Supplementary Fig. 3). However, fully accounting for these effects explicitly might alter overall estimates of TC impact. Second, we analyse state-level mortality data containing limited demographic detail because they are available consistently since 1950. More recent granular data are available, but these time series are too short to study 20-year effects. Third, we do not explicitly quantify the interacting effect of multiple TC events that occur in rapid succession or TCs that occur immediately after non-TC events. Such cases are uncommon but are likely to be impactful. Nonetheless, our results are average effects that include the impacts of these compound events. Fourth, this analysis does not capture the effect of TCs on non-fatal outcomes[19], thus our estimates may understate the public health burden of TCs. Finally, our estimates do not account for individuals who migrate outside of their

state after a TC, although they adjust for population changes within the origin state, and TC-induced migration is too small to explain these findings[50]. One case study[20] of Hurricane Katrina suggested that migration was important for determining health outcomes for that event, although Katrina was unusual in many respects and we cannot test or validate those results using these data.

Overall, our findings identify the TC climate as a driver of broad public health outcomes. This suggests that critical healthcare needs of TC-affected populations are not being fully addressed and many affected individuals probably do not realize the extent to which their own health was affected by a TC. Identifying the underlying origin of these health outcomes should prompt research and policy to mitigate this human toll. Additionally, we hypothesize that other environmental conditions might also generate unmeasured mortality burdens that can be identified using this approach.

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

## Methods

### Data

**Wind speed data.** We measure TC incidence using the LICRICE model (version 4)[28]. LICRICE is a parametric wind-field model that estimates the maximum sustained winds experienced by every location throughout the lifetime of each TC recorded in the International Best Track Archive for Climate Stewardship (IBTrACS) database[2,29,31,37,53–56]. LICRICE uses observed maximum wind speeds to reconstruct wind fields throughout the storm based on internal storm structure, location, and each storm's observed translational velocity. Many summaries of storm wind incidence are possible to construct using LICRICE, such as integrated power dissipation[28,29]; however, prior analyses have shown that the maximum wind experienced within storms is the most predictive of social and economic outcomes among numerous parsimonious metrics previously analysed. This result is consistent with most components of built structures failing catastrophically based on whether or not a threshold of stress is applied. LICRICE does not explicitly model storm surge, rainfall or flooding, however these dimensions of impact are captured in the analysis to the extent that they are correlated with integrated maximum wind speeds. For example, we find that LICRICE wind speed and NCEP rainfall from TCs are correlated within storms at the pixel level (Supplementary Fig. 3) and at the state-by-storm level ($R^2 = 0.31$, $P < 0.001$; Fig. 1e) for a limited sample of storms for which granular rainfall data are available. Iterations of the LICRICE model has been used to measure various social and economic impacts of TCs, including direct deaths and damages[29], changes to household income and expenditure[2], infant mortality[2], GDP growth[28,37] and depreciation[3]. We also compare our measure of wind speed against total direct national economic damages (normalized by GDP) from a limited sample of TCs estimated by Nordhaus[51]. We find that national wind speed exposure is a meaningful predictor of total national damages (Fig. 1d and Supplementary Fig. 2), although we note that this outcome is highly uncertain and widely understood to be biased.

Here we use a new reconstruction of incidence at the sub-national-level within CONUS. We reconstruct incidence for 0.125° × 0.125° pixel of CONUS in each of 1,230 Atlantic storms between 1930 and 2015. Supplementary Fig. 1 shows all decadal averages of these output (four example maps are also shown in Fig. 1a), illustrating the TC climatology for CONUS–however these aggregates over time are not themselves used in subsequent analysis.

To match state-level mortality data, TC incidence is collapsed from pixels to states for every month. If multiple storms impact a cell within a month, the maximum incidence at the cell level is recorded, and monthly averages are computed across pixels in each state. This spatial averaging causes our measures of incidence to be substantially lower than the maximum sustained wind speed commonly reported for storms, since only a small number of pixels experience those extreme conditions within each storm. We note that TC events generate the highest average state wind speeds compared to wind speeds from other intense storm phenomena, such as tornadoes. For reference, the minimum monthly state average wind speed we compute from TCs in our sample is $3.34 \times 10^{-4}$ ms$^{-1}$ and the 1st percentile is $8.4 \times 10^{-3}$ ms$^{-1}$. By contrast, the maximum monthly state average wind speeds from tornadoes in CONUS between 1950 and 2022 is $9.6 \times 10^{-4}$ ms$^{-1}$ and the 99th percentile is $3.8 \times 10^{-5}$ ms$^{-1}$, and for non-TC and non-tornado wind/hail events the maximum is $1.1 \times 10^{-3}$ ms$^{-1}$ and the 99th percentile is $1.3 \times 10^{-4}$ ms$^{-1}$. Therefore, the maximum non-TC wind events are comparable to the minimum (non-zero) TC events; and absent a TC, states do not experience average wind speeds of a similar magnitude as those from TCs.

Prior analysis by Hsiang & Jina[3] demonstrated that spatial aggregates of TC exposure can be used as independent variables in a regression framework to obtain unbiased average effects that are expressed at finer spatial resolutions (footnote 13 on pages 16–17 of ref. 3). As long as there is no systematic correlation between the average intensity of a storm and the likelihood that the most intense regions within that storm strike the most populated (or economically active or vulnerable) pixels within a state, regression coefficients will not be biased by spatial aggregations. This condition would be violated if, for example, there were systematic patterns such that the eyes of a Category 3 hurricanes tended to pass directly over dense cities, but the eyes of Category 2 hurricanes tended to miss cities. However, given that the paths of storms are primarily controlled by random steering winds at high altitude, interacting with the beta-effect induced by the Earth's meridional vorticity gradient, we have strong reason to believe that the spatial distribution of TC incidence within each state is orthogonal to the spatial distribution of underlying populations; and further that this covariance is independent of average TC intensity. Thus far, we know of no evidence that the trajectory of stronger (or weaker) storms systematically strike more vulnerable locations on land.

Of the 1,230 TCs that we reconstruct, 501 come within 250 km of a CONUS coastline. Intersecting these storms with state boundaries generates a total of 3,317 state-by-TC events. These longitudinal data reveals rich variation in the timing and intensity of TC incidence for individual states[52] (Extended Data Fig. 1). Within-state variation in incidence season-to-season and month-to-month provides substantial variation in TC impulses that enable us to identify the impulse-response of mortality empirically.

**All-cause mortality data.** We analyse all-cause mortality at the state-year-month level between 1930 and 2015 using data from multiple sources. Data from 1900 to 2004 were digitized and assembled by Barreca et al.[31] in their report identifying the impact of temperature on mortality. According to Barreca, this is the most comprehensive data on mortality assembled in this context. The remaining data was assembled by the authors using the CDC Underlying Cause of Death database. Data prior to 1959 was digitized from the Mortality Statistics of the United States annual volumes and is not otherwise available in a machine-readable format. Therefore, data in years prior to 1959 do not include cause of death (for example, cardiovascular disease) or demographic information (for example, age 1–44, Black)[31]. From 1959 to 2004, the data are from the machine-readable MCOD files, which include cause of death and demographic data.

For the years 2005–2015, we analyse mortality data from the public CDC Underlying Cause of Death database (2017). The data are based on death certificates for U.S. residents, which gives a single underlying cause of death and demographic data. Cause of death prior to 2000 was indexed using the four-digit ICD-9 code and 2000 onwards the index changed to the four-digit ICD-10 code. Cause of death was indexed using a four-digit ICD-10 code. We harmonized the cause of death into five categories that matched the cause of death variables from Barreca et al. We also construct a 6th category which is the difference between all-cause mortality and the sum of the 5 cause-specific categories, called 'other'. Notably, the change in CDC ICD code methodology resulted in a shift in the counts of deaths from specific causes, particularly infectious diseases and cardiovascular disease.

To account for differences in underlying age-specific mortality we decomposed the effect of TCs on all-cause mortality by four age groups in the data: <1, 1–44, 45–64, and 65+ years of age. We were limited to these age groups because these are the designations in our historical data. We computed mortality with respect to the underlying population by these same four age groups. We compute mortality by race with respect to the population by race in each state and year. Black and white are the only race categories available for the entire sample. Extended Data Fig. 3 shows the monthly all-cause mortality rate, and our predicted monthly mortality rate, for all the states in our sample.

**Direct deaths from TCs.** Direct deaths from TCs are deaths that are officially attributed to a storm by the US government. We combine

official death counts from two NOAA data sources. For storms between 1950 and 1996 we use the NOAA National Hurricane Center and Central Pacific Hurricane Center's Hurricane in History[6]. Storms from 1997 to 2015 are from the NOAA Storm Events Database[7].

**Population data.** We normalize state mortality by the population (per 100,000 people) in the state each month. Similar to the all-cause mortality data, these data must be combined from multiple sources. Pre-1968 population estimates are from Haines[57]; estimates for 1969–2000 are from the National Cancer Institute (2008); estimates for 2000–2010 are from the US Census Bureau, Intercensal Population and Housing Unit Estimates: 2000 to 2010 (ref. 58); estimates for 2010–2017 are from the US Census Bureau, US Population Estimates[59].

**Temperature data.** Average monthly temperature data are from Berkeley Earth Surface Temperatures (BEST) land surface air temperature. BEST provides a monthly mean of average, minimum and maximum surface air temperature over land covering 1753 to the present[56,60]. The temperature data are based on a large inventory of observations from over 30,000 weather stations. Using these observations gridded temperature fields are reconstructed statistically, incorporating the reliability of individual weather stations and spatial variability of temperature[56,60]. Gridded BEST temperature data are then spatially aggregated, weighted by population, to the state-month-level.

## Analysis

The econometric approach that we apply here is a top-down strategy, commonly called a 'reduced-form' analysis, that describes the overall net change of an aggregate outcome $y$ (mortality) in response to exogenous treatments $z$ (TC incidence). Under suitable conditions, this approach can identify causal effects on the outcome $y$ induced by exogenous changes in independent variable $z$ without explicitly describing all underlying mechanisms that link $z$ to $y$, without observing intermediary variables $x$ (for example, retirement savings accounts or healthcare infrastructure) that might link $z$ to $y$, or without explicitly tracking other determinants of $y$ unrelated to $z$ (such as demographic trends or health policy), denoted $w$ (refs. 23,61,62). Let $f(\cdot)$ describe a complex and unobserved process that generates state-level mortality rates $y_{t_2}$, occurring at time $t_2$ based on $x$, $w$ and $z$ that occur both at times $t_1$ and $t_2$ ($t_1 < t_2$):

$$y_{t_2} = f(z_{t_2}, x_{t_1}^1(z_{t_1}), ..., x_{t_1}^K(z_{t_1}), \ x_{t_2}^1(z_{t_1}, z_{t_2}), ..., x_{t_2}^K(z_{t_1}, z_{t_2}), \\ w_{t_1}^1 ... w_{t_1}^J, \ w_{t_2}^1 ... w_{t_2}^J) \tag{1}$$

where $x_{t_1}^k(z_{t_1})$ indicates that the $k$th factor $x^k$, which influences mortality rates $y$, at time $t_1$ is itself affected by TCs at time $t_1$. At time $t_2$, $x^k$ may be influenced both by TCs at $t_2$ and those that occur in the past at $t_1$. Here, we let there be $K$ pathways through which $y$ is impacted by intermediary variables ($x$) and $J$ ways through which determinants unrelated to TCs ($w$) impact $y$.

In this framework, the direct mortality impact of TC incidence usually reported by government agencies are the partial derivative:

$$direct\_deaths_{t_2} = \frac{\partial y_{t_2}}{\partial z_{t_2}} \tag{2}$$

which are the deaths that occur contemporaneously and directly as a result of the geophysical event itself, holding fixed all other factors.

In this analysis, we directly estimate the total change in mortality that results from TC incidence in both current and prior moments in time, allowing for the possibility that changes in other factors that are influenced by TCs may indirectly affect mortality. In this case, the overall total change in mortality from TC incidence in $t_2$ is the total derivative:

$$\frac{d}{dz_{t_2}}(mortality\_rate_{t_2}) = \frac{dy_{t_2}}{dz_{t_2}} = \underbrace{\frac{\partial y_{t_2}}{\partial z_{t_2}}}_{direct\ deaths} + \underbrace{\sum_{k=1}^{K} \frac{\partial y_{t_2}}{\partial x_{t_2}^k} \frac{\partial x_{t_2}^k}{\partial z_{t_2}}}_{indirect\ deaths} \tag{3}$$

which includes both direct deaths and deaths that result from any of the $K$ possible pathways that depend on the intermediate variables $x^k$ at time $t_2$. Empirically, we find that direct deaths are much smaller than indirect deaths in CONUS.

In addition, we also account for the possibility that deaths are delayed relative to TC incidence. Because direct deaths are usually tabulated immediately following storms, there are negligible direct deaths that are delayed. However, once we begin considering indirect deaths, it becomes possible for substantial delays to emerge due to the dynamics of different pathways $x^k$. In our analysis, we also estimate the total deaths that occur at time $t_2$ as a result of TC incidence that occurs at an earlier time $t_1$, which is the total derivative

$$\frac{d}{dz_{t_1}}(mortality\_rate_{t_2}) = \frac{dy_{t_2}}{dz_{t_1}} = \underbrace{\sum_{k=1}^{K} \frac{\partial y_{t_2}}{\partial x_{t_2}^k} \frac{\partial x_{t_2}^k}{\partial z_{t_1}}}_{indirect\ via\ x_{t_2}} + \underbrace{\sum_{k=1}^{K} \frac{\partial y_{t_2}}{\partial x_{t_1}^k} \frac{\partial x_{t_1}^k}{\partial z_{t_1}}}_{indirect\ via\ x_{t_1}} \tag{4}$$

This expression does not contain a term for direct deaths, but it contains two summations which capture the effects of past TC incidence ($z_{t_1}$) on current mortality ($y_{t_2}$) via past intermediate variables ($x_{t_1}$) and current intermediate variables ($x_{t_2}$). In practice, we explore the possibility of indirect effects that emerge over the course of 240 months following TC incidence, one could generalize this framing to a corresponding number of summations.

The possibility of delayed indirect deaths has two major implications regarding how indirect mortality is estimated and how those results are interpreted. First, because TC incidence at multiple points in the past, as well as the present, might affect current mortality, we must account for both the present and past influence of TC incidence simultaneously for each instance of the outcome. This is accomplished via deconvolution[25–27,56,63,64], implemented here using a distributed lag-model solved via ordinary least squares, detailed below.

Second, each TC event affects mortality outcomes at multiple points in time, thus computing the full impact of a TC event requires summing these impacts that might emerge gradually. In the simplified two-period framework above, the total impact from TC incidence at $t_1$ is then

$$\frac{d}{dz_{t_1}}(mortality\_rate_{t_1} + mortality\_rate_{t_2}) = \frac{dy_{t_1}}{dz_{t_1}} + \frac{dy_{t_2}}{dz_{t_1}} \tag{5}$$

which can be expanded further by substituting from the equations above. These terms, if plotted separately, characterize the impulse response of $y$ in reaction to the TC impulse $z_{t_1}$. In our actual analysis, we compute the average cumulative impact of a single TC that occurs at $t_0$ over 240 subsequent months ($t_1 - t_{240}$). Following substitution and simplification, this can be expressed as

$$\frac{d}{dz_{t_0}}\left(\sum_{\ell=0}^{240} mortality\_rate_{t_\ell}\right) = \underbrace{\frac{\partial y_{t_0}}{\partial z_{t_0}}}_{direct\ deaths} + \underbrace{\sum_{\ell=0}^{240}\sum_{k=1}^{K}\left(\sum_{j=\ell}^{240} \frac{\partial y_{t_j}}{\partial x_{t_\ell}^k}\right)\frac{\partial x_{t_\ell}^k}{\partial z_{t_0}}}_{indirect\ deaths} \tag{6}$$

which describes the overall total impact of a storm through all pathways across all possible delays $\ell = [0, 240]$. Note that neither $K$ nor $x^k$ need ever be specified explicitly in our estimation below. This expansion reveals that, accounting for numerous possible pathways operating over different delays, a single TC event can potentially generate a total mortality impact much larger than the direct deaths traditionally reported.

To compute the overall mortality burden imposed by the TC climate of CONUS, we compute the full mortality response across all age groups in each state, accounting for the incidence of each storm on the state:

$$mortality\_burden = \sum_{\ell=0}^{240} \sum_{t \in months} \sum_{s \in storms} \sum_{i \in states} population_{i,t+\ell}$$
$$\cdot z_{sit} \cdot \frac{d}{dz}(mortality\_rate)_{i,t+\ell} \qquad (7)$$

where $z_{sit}$ is the TC incidence of storm $s$ on state $i$ in month $t$, $population_{i,t+\ell}$ is the population in state $i$ in month $t + \ell$, and $\frac{d}{dz}(mortality\_rate)_{i,t+\ell}$ is our estimate for the total impact of TC incidence on the mortality rate in state $i$ in month $t + \ell$. In practice, this effect is nonlinear, but it is expressed linearly here for simplicity. Information on state $i$ affects the impulse responses used in these calculations because TC risk is computed by state and affects the structure of the impulse-response function.

## Econometric implementation

**Identification.** Our econometric analysis exploits the quasi-random variation in the location and intensity of TC incidence to estimate the impact of TCs on mortality separately from other known and unknown factors that affect mortality across locations and over time. As described above, this reduced-form approach captures the effect of all possible channels of influence that may increase mortality after a TC[33,65]. Because the location, timing and intensity of TC incidence is determined by oceanic and atmospheric conditions that are beyond the control of individual states, we assume mortality TC incidence is as good as randomly assigned[23,61]. For reference, Extended Data Fig. 1 shows the sequence of monthly TC incidence by state for all the states in our sample.

We note that some early analyses of natural disaster impacts utilized social outcomes (for example, direct economic damage[66] or direct mortality) as a proxy measure of physical hazard severity. However, it is now understood that use of these metrics as independent variables may confound estimated treatment effects, since they are endogenously determined by many of the same underlying covariates (for example, healthcare, infrastructure, inequality and institutions) that mediate other outcomes from disasters[22,29,33–35,67]. Thus, use of these proxy measures for hazard severity exposes analyses to selection biases, since population characteristics may cause observational units to 'select' into more or less severe treatment[23]. We therefore focus this analysis strictly on independent variables that are physical measures of TC incidence (wind speed), because they are exogenous and cannot be influenced by the populations that are impacted[22].

**Deconvolution.** In considering the long-run impact of TCs on mortality, we hypothesize that there may be a delay between the geophysical event and components of the mortality response. Because TCs are regular events that occur frequently in CONUS, the possibility of this delay means that the time series of mortality outcomes we observe in data may be the result of overlapping responses from multiple storms. Extended Data Fig. 2 displays a cartoon of this data-generating process. In such a context, the empirical challenge is isolating the impact from individual storms which might be partially confounded by the overlapping TC signals from earlier or later storms. We use the well-established signal-processing approach of deconvolution[25–27,33,63,64] to recover the characteristic impulse-response function for a TC impulse. Conceptually, this approach searches for an impulse-response function that, if applied to all TCs in the data simultaneously, best fits the observed outcome data. Stated another way, this approach estimates the effect of each TC accounting for the potential overlapping impact of all other TCs, subject to the constraint that TCs share a characteristic impulse-response function.

This method assumes that the overlapping responses influencing mortality at a moment in time are additively separable, an assumption that we think is reasonable given the overall small impact that any individual storm event has mortality rates at a moment in time in a particular region (0.019% on average, 0.04% for storms at the 95th percentile). We solve for the structure of the impulse response, characterized by a set of coefficients $\beta$, using ordinary least squares (OLS). This is a standard procedure that is commonly applied in a wide range of disciplines[26]. In some fields, such as econometrics, deconvolution is frequently described as estimation of distributed lags[27].

**Baseline specification.** Our main results are based on a linear model of TC incidence on mortality rate. Indexing states by $i$ and month of sample by $t$, we solve the model

$$mortality\_rate_{it} = \sum_{\ell=-72}^{240} (\beta_\ell \cdot wind\_speed_{i,t-\ell})$$
$$+ \delta_{1,i} \cdot temp_{it} \cdot s_i + \delta_{2,i} \cdot temp_{it}^2 \cdot s_i + \mu_1 \cdot m_{it} \qquad (8)$$
$$+ \sum_{n=1}^{8} (\eta_n \cdot s_i \cdot t^n) + \mu_2 \cdot m_{it} \cdot t + \mu_3 \cdot h_t + \epsilon_{it}$$

via OLS. Here $wind\_speed_{i,t-\ell}$ is TC maximum wind speed $\ell$ months prior to month $t$, $s_i$ is a vector of state-specific dummies, $m_{it}$ are state-by-month-of-the-year dummies (for example, an indicator variable for whether state = Florida and month = January), $h_t$ are month-of-sample dummies (for example, an indicator variable for whether the month = January, 1974), $temp_{it} \cdot s_i$ is month-of-sample temperature interacted with state dummies, and $temp_{it}^2 \cdot s_i$ is squared month-of-sample temperature (also interacted with state). Each coefficient $\beta_\ell$ measures the marginal effect of an additional ms$^{-1}$ of wind speed incidence on mortality $\ell$ months after a TC conditional on the effect of any prior TC. We include 72 lead terms in equation (8) as a falsification test, also known as negative exposure controls, since idiosyncratic future TC incidence should not alter current health outcomes.

This model accounts for state-specific quadratic effects of temperature on mortality based on prior literature, which has shown that very hot and very cold temperatures cause higher levels of mortality relative to more moderate temperatures[31,33–35]. Each state is allowed to express a different mortality response to temperature extremes, implemented via interaction with the state dummy variable $s_i$. Supplementary Fig. 6 shows the state-specific shape of the quadratic functions we estimate for the temperature-mortality response. Consistent with prior findings studying patterns of adaptation[31,33–35], we observe that some states have a flatter response at temperatures that are more common for that state (for example, cold in Minnesota) while other states have steeper curves at those same temperatures if they are less common (for example, cold in Florida). In an effort to balance parsimony with model richness, we omit extended lags of temperature based on prior literature demonstrating that impacts on mortality dissipate within a month[33,68].

This model also non-parametrically accounts for:
- State-by-month-specific constants (fixed effects) that capture average differences between states, as well as unique seasonable patterns within states ($\mu_1 \cdot m_{it}$). These terms will account for differences in mortality driven by unobserved factors at the state level, such as health policies, as well as factors that cause seasons within a state to exhibit higher mortality, such as holidays.
- State-specific nonlinear trends in mortality, captured by eighth-order polynomials in month-of-sample interacted with state fixed effects ($\sum_{n=1}^{8} (\eta_n \cdot s_i \cdot t^n)$). These trends account for unobserved factors that have caused mortality within states to change over time, such as changing health policies or demographic trends.
- Trends in state-specific seasonal patterns of mortality, captured by a linear trend in month-of-sample interacted with state and month fixed effects ($\mu_2 \cdot m_t \cdot t$). These trends are additive to the state-specific

polynomial and allow for the model to express gradual convergence or divergence in the seasonality of mortality within a year, and allows for these changes to differ by state. These trends account for unobserved factors that drive gradual changes over time that may cause mortality in certain times of year (for example, January) to change relative to other times of year (for example, June). For example, if adoption of safety standards has reduced wintertime mortality from motor vehicle accidents or improvements in medical care have reduces summertime deaths from infectious diseases. Extended Data Fig. 4a illustrates the combined effect of these state-specific seasonal trends, state-specific polynomials, and state-by-month-specific constants on model predictions for Florida and New Jersey. For example, the seasonality of mortality in Florida has lessened over time, in conjunction with other nonlinear trends.

- National month-of-sample fixed effects that capture nonlinear and/or discontinuous changes in mortality rates nation-wide ($\mu_3 \cdot h_t$). These terms are particularly important for capturing idiosyncratic spikes in mortality that result from nation-wide conditions, such as influenza outbreaks, as well as any systematic changes in the accounting methodology of mortality by the CDC. Comparisons of Extended Data Fig. 4b,c illustrates how the inclusion of these terms in the model alters the ability of the model to capture unusual spikes in mortality that are not captured by other model elements, including the trends listed above, TCs, and temperature.

Overall, the fit for this model is high (in-sample adjusted $R^2 = 0.93$ with 25,062 degrees of freedom). Extended Data Fig. 3 overlays predictions with observations for all states (same as in Fig. 1c). We find that all of the non-parametric controls listed above are important for passing standard specification checks. For example, failure to account for trends flexibly enough causes estimated leads to deviate from zero or randomization-based placebo tests (described below) to recover non-zero central estimates. These results are unchanged if we use a Poisson regression specification (Extended Data Fig. 6a).

In a robustness test, we interact the month-of-sample fixed effects with 3 region indicator variations. We continue to obtain our main findings after introducing these additional 2,062 parameters to the model, although the estimates become much noisier and attenuate slightly. Both of these effects are well understood results of including a large number of highly flexible variables that absorb a meaningful fraction of the true variation in the independent variable[69].

We evaluate the distribution of the unmodelled variation represented by the error term $\epsilon_{it}$ and find that it essentially follows a Normal distribution except with slightly positive kurtosis (Supplementary Fig. 5a). The distribution of these residuals appears stationary throughout the sample period and independent over time (Supplementary Fig. 5b). The consistency of the distribution of these errors is attributed to the high degree of flexibility in the non-parametric terms of our econometric specification, which are able to capture those components of the data-generating process that would otherwise appear as auto-correlated errors. On the basis of this evaluation, we construct OLS standard-error estimates as the underlying assumptions for these estimates appear to be reasonably satisfied. In addition, we find strong support for this modelling choice when we conduct a variety of permutation tests for statistical significance (Extended Data Fig. 5), all of which indicate that our asymptotic estimates for confidence intervals are correctly (and possibly conservatively) sized and our tests for statistical significance correctly powered. Notably, these permutation tests do not rely on the assumptions used to estimate these confidence intervals, thus they can be considered independent corroboration for the validity of this approach.

**Cumulative effects.** To compute the total effect after the TC makes landfall, we estimate the cumulative sum of $\beta_\ell$ for each $\ell \in [-72, 240]$. We compute $\Omega_\ell = \sum_{k=o}^{\ell} \beta_k$, which denotes the cumulative impact of an additional $1\,ms^{-1}$ wind speed incidence on mortality $\ell$ months after a TC event. We account for the estimated covariance of $\beta$ when estimating uncertainty in $\Omega$. We normalize the sums relative to the impact one month prior to the TC, $\ell = -1$ such that $\Omega_{-1} = 0$.

**Randomization-based placebo tests.** Given the complexity of our model, the long delays we study, and the absence of prior analyses of long-run total mortality from TCs, it is not possible to subjectively evaluate our econometric analysis against any prior benchmark. In such a context, there is risk of unknowingly recovering a spurious estimate generated as an artefact of our model specification. A strong test designed to avoid such artefacts is to ensure that model estimates of TC impacts on mortality are unbiased in a variety of situations where the structure of the association has been manipulated. In four tests, we shuffle the true TC data in different ways. In each case, this shuffling should break any correlation between TC incidence and mortality such that an unbiased estimate of the effect of shuffled TCs on mortality is zero. However, in each case, some of the structure in the original TC data are allowed to remain in the shuffled TC data. For example, randomization within a state over time retains the average cross-sectional patterns of TC incidence, but destroys any time-series structure. Thus, these tests allow us to examine whether, in each case, the remaining structure generates artefacts in the model that would produce a spurious result, also known as a negative exposure control[70]. Any non-zero correlation, on average, would indicate a biased model where the bias is driven by the non-randomized components of the original TC data.

Within each type of randomization we scramble TC assignment 1,000 times and run the linear version of the model (equation (8)) on each re-sampled version of the data. Our four randomizations are illustrated graphically in Supplementary Fig. 7 and described below:

- Total randomization shuffles TC events across all state-by-month observations. This tests whether the unconditional marginal distribution of TC events, which has a long right tail, could generate bias. Results are shown by light blue boxes in Fig. 1g.
- Within-state randomization shuffles the sequencing of TCs that a state experiences over time. TCs are always assigned to the correct state, but the month and year assigned to each storm is random. The cross-sectional average pattern of storm incidence is preserved in the data. Thus, this tests whether time-invariant cross-sectional patterns across states generate spurious correlations. Results are shown by dark blue boxes in Fig. 1g.
- Within-month randomization shuffles the TC incidence across states within each month-of-sample. TCs are always assigned to the correct month and year, but the state assigned to each storm is random. The average time-series structure of TC incidence nation-wide is preserved. Thus, this tests whether national or seasonal trends, which are nonlinear, could bias estimates produced by this model. Results are shown by maroon boxes in Fig. 1g.
- Across-state shuffles complete TC times-series across states, keeping the timing and sequence of storms correct as blocks. TCs are always assigned to the correct month and year, and the sequence of storms experienced by a state is always a continuous sequence that is observed in the data. However, the state that is assigned that sequence is randomly chosen. This tests whether trends within a state and within the sequence of storms that a state experiences could generate bias. This test differs from the within month randomization because state-level trends often differ across states (see Extended Data Fig. 1 and Extended Data Fig. 3) and there are complex seasonal patterns that could potentially affect estimates. Results are shown by red boxes in Fig. 1g.

The estimated impact of TCs in each of these placebo tests is zero on average, to within a high degree of precision. Extended Data Fig. 5 illustrates distributions of estimates for all lags. These results demonstrate that non-exchangeability across states within a month, across months

within a state, or across states (conditional on month of sample) does not confound our analysis; indicating that the rich set of fixed effects and trends successfully adjust for many patterns of TC incidence and/or mortality such that the remaining conditional variation is as good as random.

**Permutation tests for statistical significance.** In addition to establishing the unbiasedness of our main point estimates, the four randomizations above can be utilized to serve a second function: estimating statistical significance of our estimates. These randomizations enable approximate permutation tests[71], allowing for different types of autocorrelation to remain in the TC data. We use these randomizations to examine the likelihood of randomly obtaining an entire impulse-response function similar our actual estimate, if in reality no such relationship exists in the data. To do this, we jointly test the significance of all true cumulative estimates $\Omega_\ell$ against the null hypothesis that a similarly extreme sequence of estimates is generated randomly. Extended Data Fig. 5 overlays the true estimates for $\Omega_\ell$ on distributions of similar estimates from each randomization. $P$ values for individual lag terms ($p_\ell = \Pr(|\Omega_\ell^{\text{randomized}}| > |\Omega_\ell|)$) are plotted in the right subpanels and are all individually statistically significant ($P < 0.05$) for $\ell < 150$ months in each randomization. However, the significance of the complete sequence of coefficients that together compose the entire impulse-response function is far greater. We compute a joint $P$ value for the full impulse response between 0 and 172 months ($p_\ell = \Pr(\cap_{\ell=0}^{172}(|\Omega_\ell^{\text{randomized}}| > |\Omega_\ell|))$) that ranges from $P = 0.012$ to $P = 0.0012$ across the randomization approaches (Extended Data Fig. 5). We conclude that it would be extremely unlikely to obtain an impulse-response function as extreme as our main result due to chance.

**Subsamples by age, race and cause of death.** In addition to the all-cause mortality rate for the entire population, we also present the results stratified by age, race, and cause of death. For the six cause-specific mortality rates we compute mortality per 100,000 of the total population in that state in the time period (for example, total number of deaths from cardiovascular disease divided by total population times 100,000). 'Other' mortality is the difference between the total deaths and the sum of all the other causes. For age groups and race we report mortality risk as the outcome of interest. For example, for the Black population, we construct the mortality rate for Black people as the number of deaths of Black people in the state divided by the Black population. We do the same procedure by age group. We also report the mortality by these strata as a proportion of the total deaths traceable to TC incidence, see Extended Data Fig. 7.

**Subsamples by average TC risk.** To evaluate whether there is heterogeneity in the mortality response of states that are frequently exposed to TCs compared to those infrequently exposed, we stratify the sample by the average TC incidence they experience. We allow the mortality impulse response to differ based on quartiles of states, sorted by their average TC incidence. We implement this by including and interaction with an indicator variable for the quartile of their average wind speed incidence, following the general approach for modelling adaptation developed in refs. 29,33. Average wind incidence is a measure of the expected TC risk a population bears, which informs preventive risk reduction investments, behaviours, or other adaptive actions they take to reduce the expected harm from TCs. We approximate this measure by computing as the mean wind speed in each state $i$ across the period $t$ in our sample

$$wind\_speed_i = \frac{\sum_{t=1}^{1,032} wind\_speed_{it}}{1,032}$$

and assigning the quartiles of these means to each state. We estimate a model that allows each quartile to express a different impulse response

to TCs and observe little difference in the impact of TCs on mortality between the second through fourth quartile (Extended Data Fig. 6c). The effect for the second through fourth quartile are not statistically significantly different than the effect for the second through fourth quartile, combined ($P = 0.38$). Thus, to improve the efficiency of our model and limit unnecessary noise in our estimates, we pool quartiles 2–4 to create the 'high average incidence' group in the main results (shown in Fig. 2f) ('high incidence' in equation (9)). The 'low average incidence' group is the first quartile of average wind speed alone ('low incidence' in equation (9)). We additionally evaluate whether the spatial distribution of populations relative to the coast, within states, alters the mortality response to TCs. Stratifying states on the basis of the average fraction of the population that lives in coastal counties, we fail to find evidence that states with high concentrations of coastal populations are systematically different from states with little or no coastal population (Extended Data Fig. 6d). Lastly, we evaluate whether the overall spatial correlation between populations and average wind speed incidence, within each state, alters the mortality response to TCs. Stratifying states on the basis of within-state spatial correlation across $0.125° \times 0.125°$ pixels, we fail to find evidence that states with higher spatial correlations are systematically different from states with little or negative correlations (Extended Data Fig. 6e).

**Nonlinear effects of TCs.** We evaluate whether the mortality impact of a TC is nonlinear in the physical intensity of the event. This could occur, for example, if more extreme TC events generate exponentially more physical damage[51,72] or if they elicit different government responses[18,73]. Empirically, we find that excess mortality 180 months after a TC is well approximated by a linear function of max wind incidence, particularly for TCs with area-average max wind speeds between 0 and 20 ms$^{-1}$, which is the majority of events (93%) in our sample (Extended Data Fig. 8). However, for the most extreme events (>30 ms$^{-1}$, 1.4% of events) we find that excess mortality is generally lower than a linear function would predict, although these nonlinear effects are not themselves statistically significant. We lack the data to fully evaluate the underlying causes of this nonlinearity, but believe it is an important topic for future study. For example, it is possible that societal responses to the most extreme events (for example, disaster relief) are more effective at alleviating mortality impacts of TCs because these events attract a disproportionate quantity of attention, compared to less extreme events that are also harmful but less salient[74]. Regardless of their cause, we account for these non-linearities in calculations below as they contain information on how populations in CONUS have adapted to their TC climates[5,73].

To estimate nonlinear effects of TCs, we estimate a model that is identical to the benchmark linear model in equation (1), but it allows the magnitude of the TC mortality impulse-response function to be cubic in TC incidence. The motivation for this approach is the possibility that the relationships between wind speed and long-run mortality does not increase linearly. For example, very high wind speeds may cause extreme damages and/or elicit greater governmental and humanitarian responses, which would mean that a unit of increase from 40 to 41 ms$^{-1}$ wind speed may have a larger or small mortality impact compared to an increase from 5 to 6 ms$^{-1}$. Since the nonlinear impact of TCs may be influenced by the historical TC experience and baseline TC risk of populations, this model also allows for the nonlinear response to differ based on the risk categorization for each state:

$$
\begin{aligned}
mortality\_rate_{it} = & \\
\sum_{\ell=-72}^{240} \sum_{r=1}^{3} (&\theta_{r,\ell}^{low\_incidence} \cdot wind\_speed_{i,t-\ell}^{r} \cdot Q_i^{low\_incidence} \\
+ &\theta_{r,\ell}^{high\_incidence} \cdot wind\_speed_{i,t-\ell}^{r} \cdot Q_i^{high\_incidence}) \\
+ &controls + \epsilon_{it}
\end{aligned}
\tag{9}
$$

where $r$ indicates an exponent and the controls are identical to those in equation (8). $Q_i^{low\_incidence}$ ($Q_i^{high\_incidence}$) is an indicator variable that is set to one if state $i$ is in the low incidence (high incidence) group. The coefficients $\theta_{1\ell}$, $\theta_{2\ell}$ and $\theta_{3\ell}$ separately capture the cubic relationship between wind speed incidence and mortality for low and high-risk states in each lag period. Extended Data Fig. 8 displays the cumulative impact estimated using both the linear and nonlinear models after 180 months. These results are unchanged if we alternatively use a cubic spline regression specification (Extended Data Fig. 8). The low-risk response is slightly convex relative to the linear estimate, while the high-risk response is slightly concave. In both cases, impacts are relatively well approximated by the linear version of the model and only diverge (insignificantly) at very high levels of incidence that are rare in sample. Distributions for in sample frequency are shown in lower panels of Extended Data Fig. 8.

**Computing mortality burdens.** The total impacts of all TCs on mortality are estimated using each version of the model, presented in Supplementary Table 1. We compute the excess mortality from TCs by state, month, and TC. These estimates are presented in Figs. 3 and 4 and Supplementary Tables 1 and 2.

Figure 4 displays the estimated excess mortality from TCs by state, age and race, computed using equation (8) applied to equation (7). Figure 4a presents our estimated full TC mortality burden, similar to equation (7) but by state ($mortality\_burden_{it}$) as an average proportion of total deaths ($mortality_{it}$) in each state between 1950 and 2015:

$$proportion_i = \frac{\sum_{t \in \text{month}} mortality\_burden_{it}}{\sum_{t \in \text{month}} mortality_{it}}$$

Similarly, we estimate the proportion by state and age group ($proportion_{i,a}$), shown in Fig. 4d,f. Proportion and total excess mortality for the Black population is based on mortality burden estimated with the mortality risk for the Black population, therefore $proportion_{Black} = \frac{\sum_{t \in \text{month}} \sum_{i \in \text{state}} mortality\_burden_{it,Black}}{\sum_{t \in \text{month}} \sum_{i \in \text{state}} mortality_{it,Black}}$.

Figure 4g illustrates the impact of the TC climate on the geographic differences in average annual all-cause mortality rate between states that do not experience TCs ('non-TC states') and states that do ('TC states (actual)'), in this context. We also subtract the average annual TC mortality burden from the actual average annual mortality for each TC-impacted state ('TC states without TCs').

**Decomposing trends in mortality burden.** We examine the differences in TC events and population distribution before 2001 and after 2001 in order to understand why the mortality burden after 2001 is increasing more rapidly than it did before 2001 (Fig. 4h–j). Figure 4h shows the distribution of the number of TCs that made landfall each year before 2001 and after 2001. Similarly, Fig. 4i shows the maximum annual wind speed per year and Fig. 4j plots the average wind speed per year as experienced by a proportion of the CONUS population. The changes in the distribution of TC events affecting CONUS after 2001 were themselves probably caused by a combination of factors, including warmer sea surface temperatures[11,75] and reductions of anthropogenic aerosol emissions[76,77] (which create an environment more amenable to TC intensification); and shifts in steering winds[78] (which direct a larger fraction of TCs to landfall in CONUS after formation). We note that identifying factors driving the TC climate remains an active area of research[56,79].

To understand the 1950 to 2015 trend in the national aggregate mortality burden, we re-estimate $mortality\_burden_t$ for each month of the sample with different populations to decompose the long-term trend based on various population patterns. We first replace $population_{i,t+\ell}$ from equation (7) with fixed 1950 or 2015 populations (Fig. 4k, red and maroon lines). To generate the yellow line in Fig. 4k, we replace $population_{i,t+\ell}$ with an estimate of the 2015 population 'deflated' to 1950 levels. Specifically, we first compute a national population deflation fraction,

$$\Delta = \frac{\sum_{i \in \text{state}} population_{i,2015} - \sum_{i \in \text{state}} population_{i,1950}}{\sum_{i \in \text{state}} population_{i,1950}}$$

where $population_{i,2015}$ is the state-specific population in 2015 and $population_{i,1950}$ is the state population in 1950. We then calculate

$$deflated\_population_{i,2015} = \frac{\sum_{i \in \text{state}} population_{i,2015}}{1 + \Delta}$$

and apply $deflated\_population_{i,2015}$ to equation (7). This value is an adjusted state-level population that allows the total national population to match 1950 level but have a relative spatial distribution that reflects 2015.

## Reporting summary

Further information on research design is available in the Nature Portfolio Reporting Summary linked to this article.

## Data availability

All data are available for download at Zenodo (https://zenodo.org/uploads/10459719 (ref. 80)). The full data processing code is not included but the collected full dataset needed for the main analysis (Figs. 1–4) is provided in the file DATA_hurricane_mortality_temp_month_state_19302015.dta. This includes the matched LICRICE-generated TC wind speed and the all-cause mortality data from the CDC Mortality Statistics of the United States annual volumes, the MCOD files and Underlying Cause of Death database; the population data from the Inter-university Consortium for Political and Social Research and the US Census Bureau, Intercensal Population and Housing Unit Estimates; and the temperature data from BEST. We also provide additional datasets required for the supplementary information and for plotting the maps, which are described in the ReadMe file.

## Code availability

Code used in our analyses is available in an open-source repository and includes a ReadMe file describing the code and data: https://github.com/Global-Policy-Lab/young_hsiang_tc_mortality.git.

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

**Competing interests** The authors declare no competing interests.

**Additional information**
**Correspondence and requests for materials** should be addressed to Solomon Hsiang.

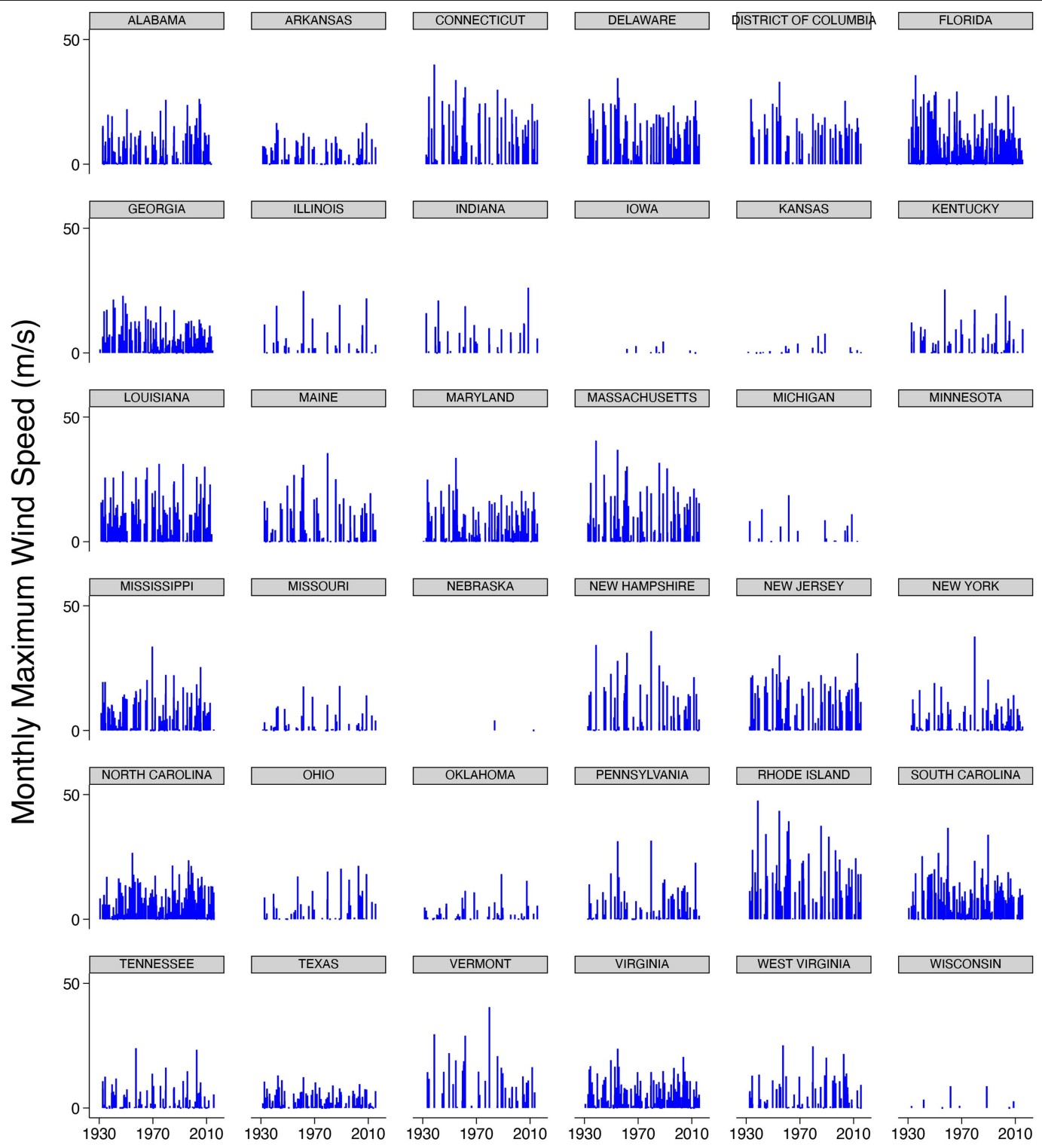

**Extended Data Fig. 1 | State monthly maximum wind speed from tropical cyclones.** LICRICE modeled monthly maximum wind speed from tropical cyclones between 1930 and 2015. State tropical cyclone wind speeds are averages across pixels.

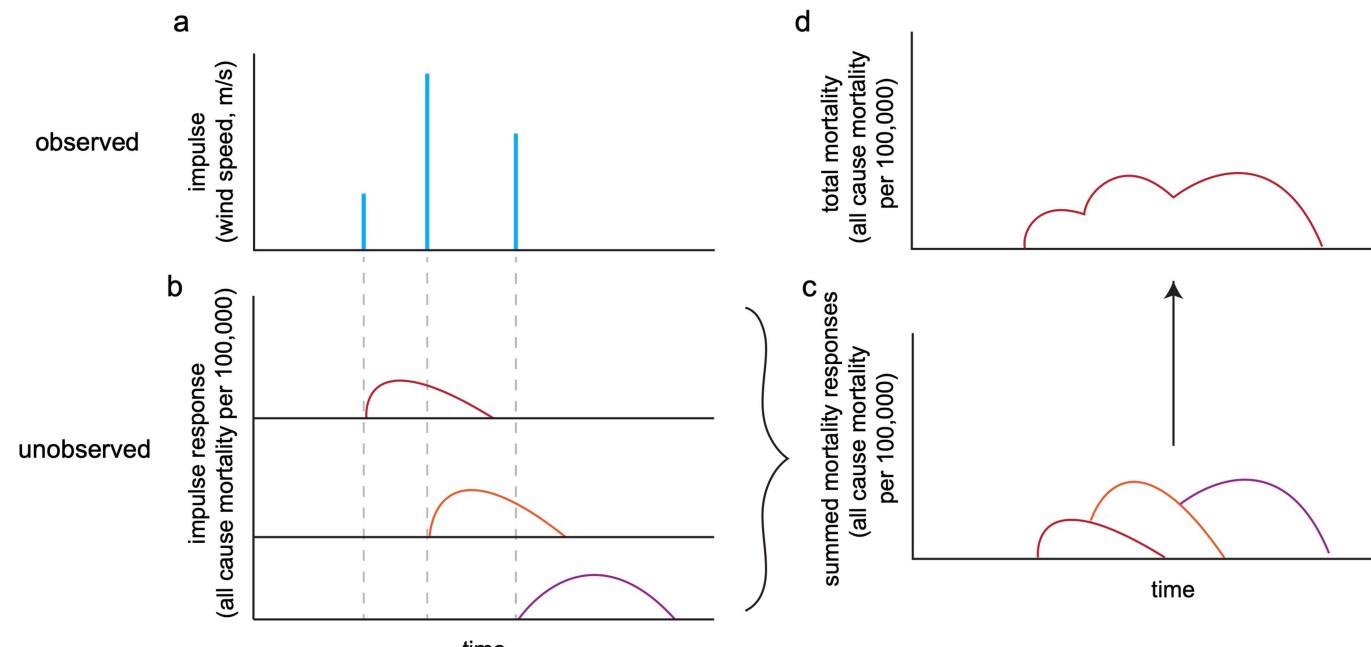

**Extended Data Fig. 2 | Breaking down the effect of tropical cyclones on mortality into analytical components.** (**a**) mapping a sequence of tropical cyclone wind speed incidences (an "impulse" modeled as Dirac delta functions). (**b**) Each impulse has an underlying, unobserved, response function ("impulse response"). (**c**) Superposition of overlapping impulse responses over time. (**d**) Observed envelope of the overlapping impulse responses, used to estimate the average impulse response function (see Methods).

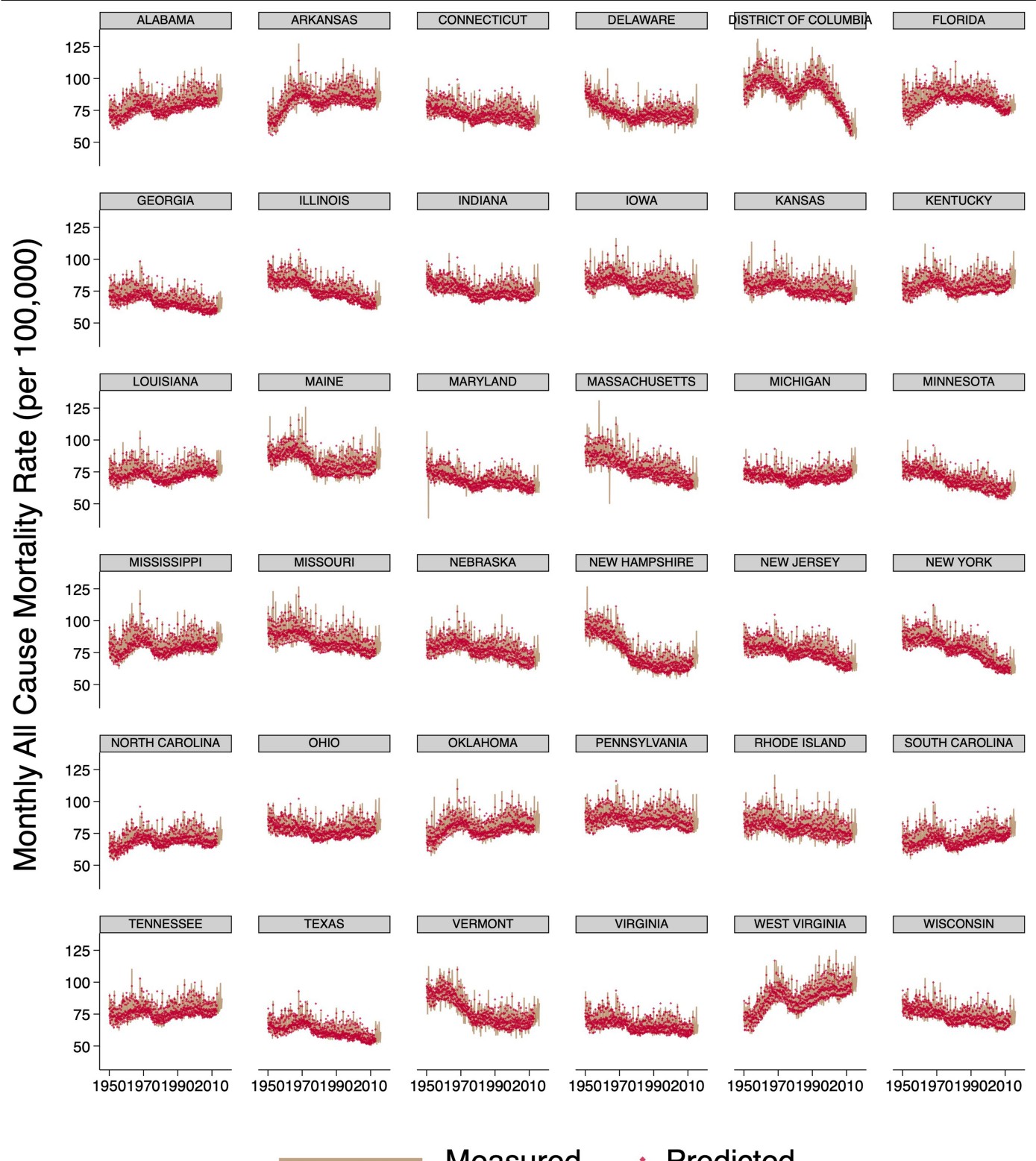

**Extended Data Fig. 3 | State monthly all-cause mortality rate.** Monthly state all-cause mortality rate (per 100,000) between 1950–2015 (orange line) and predicted mortality rate from Eq. 7 (red dots).

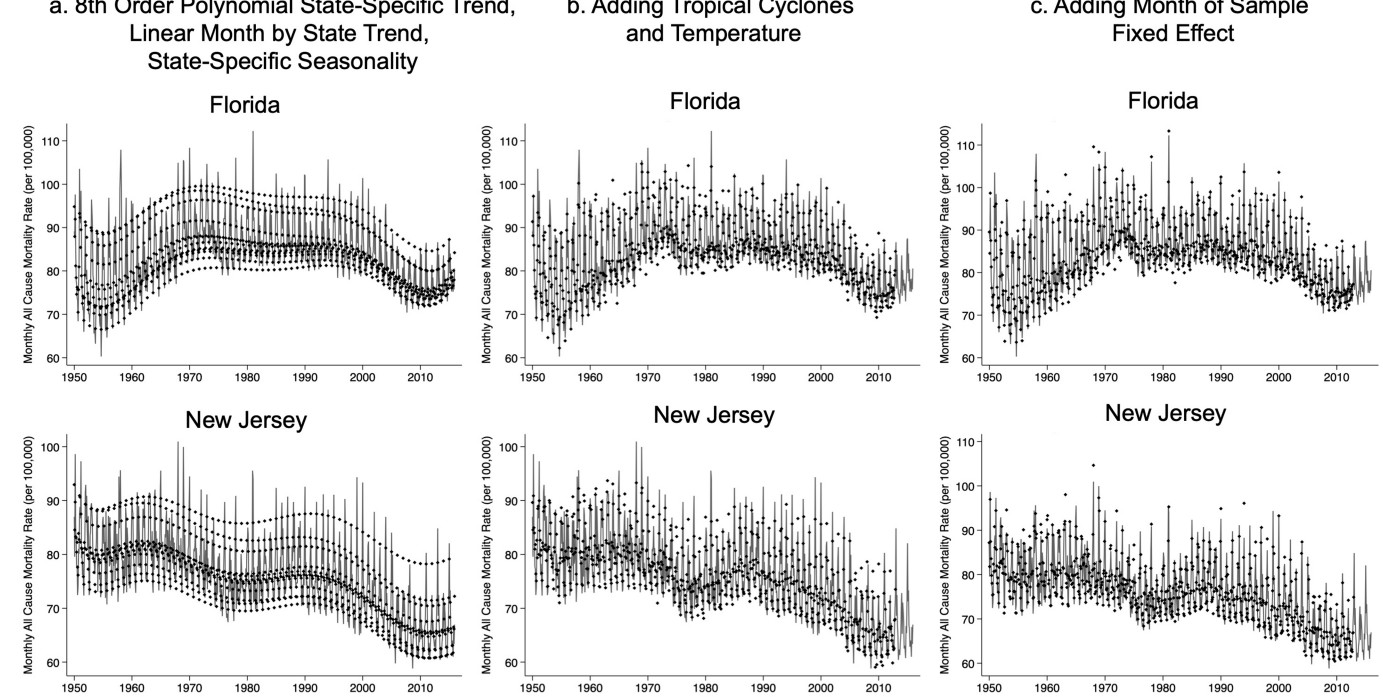

**Extended Data Fig. 4 | Examining model fit with fixed-effects, time trends, tropical cyclones, and temperature.** All cause monthly mortality observations, per 100,000, (grey line) and predicted excess all cause monthly mortality (black dots), in Florida and New Jersey. (**a**) predictions estimated from a model with eighth-order polynomial state-specific trend, linear month-by-state trend, and state-specific seasonality. (**b**) Predictions including elements in (a) plus quadratic temperature and linear wind speed effects. (**c**) predictions including elements in (b) and month of sample fixed-effects, equivalent to Eq. 7 (see Methods).

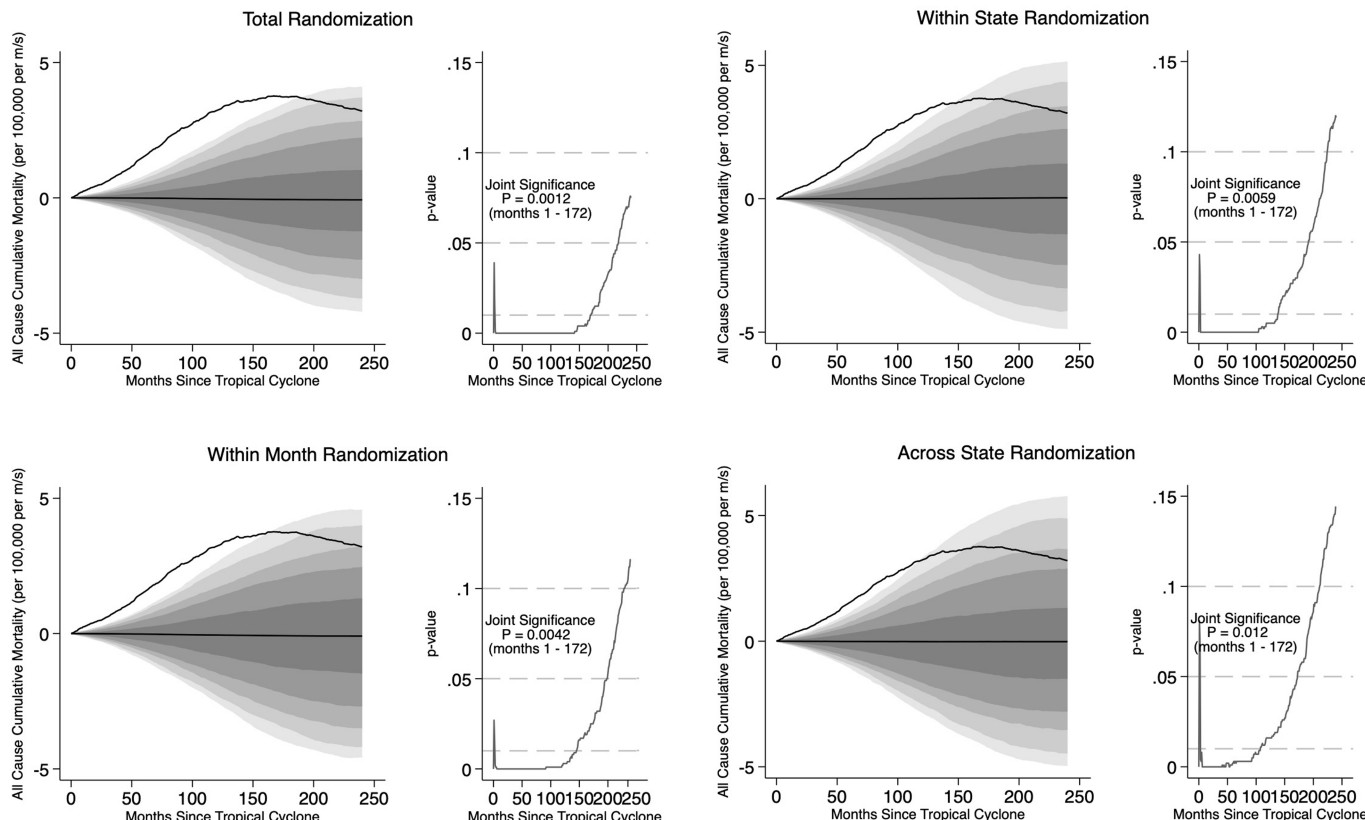

**Extended Data Fig. 5 | Randomization-based placebo tests and permutation tests.** In left panels are true estimates for $\Omega_\ell$ (black line), plus quartiles and 1st, 2nd, 5th, 10th, 90th, 95th, 98th, 99th percentile of randomized estimates, $\Omega^{randomized}$, (grey shaded plumes) for each of the four randomizations described in the main text and illustrated in Fig. SI7. P-values ($p_\ell$) for individual lag terms in right panels are for a two-sided joint permutation tests of the cumulative effect accounting for estimated parameter covariances (see Methods).

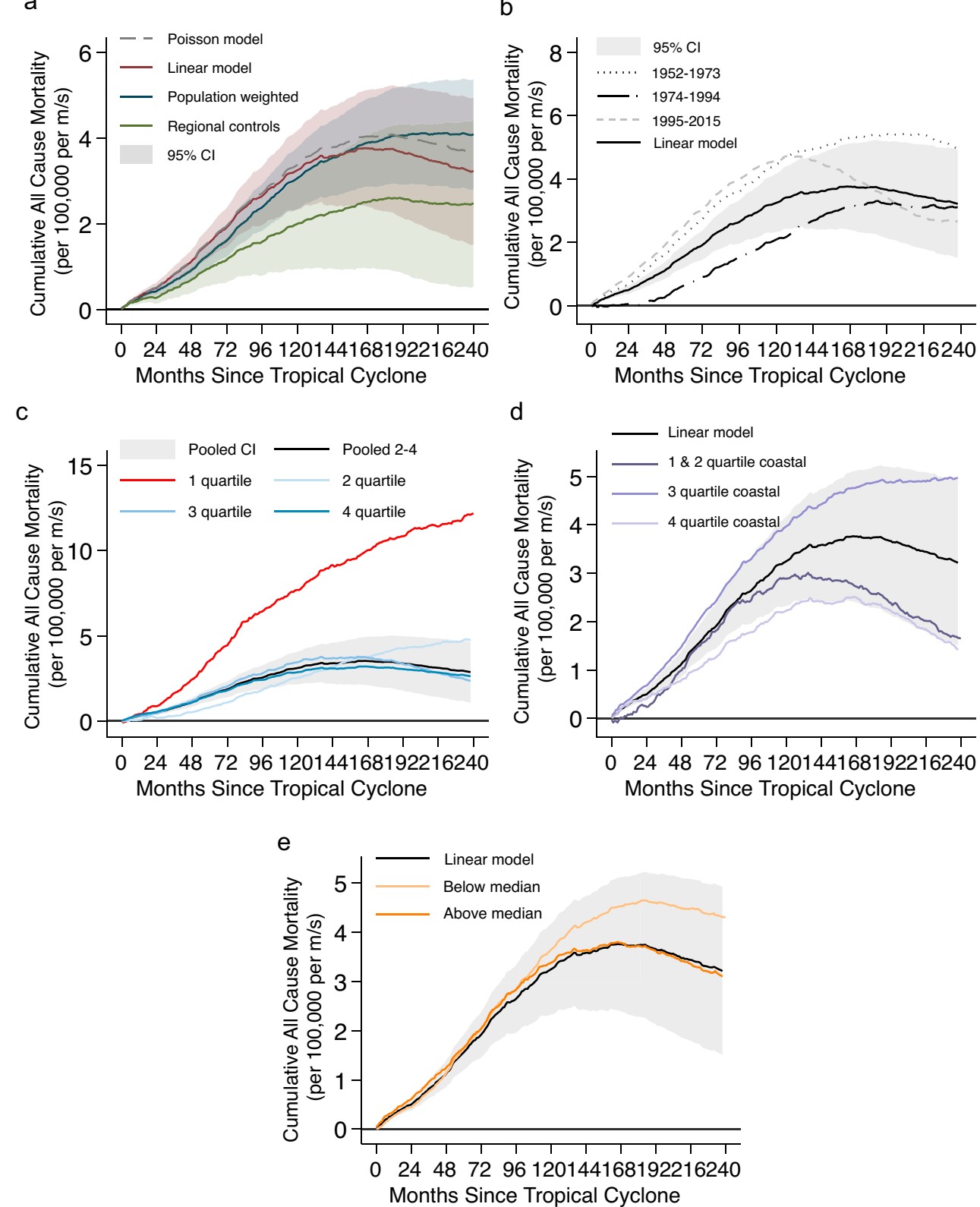

**Extended Data Fig. 6** | See next page for caption.

**Extended Data Fig. 6 | Excess mortality following a tropical cyclone, adjusting for population, model specification, changes over time, average wind speed, and coastal population density.** (**a**-**c**) Estimated cumulative excess all-cause mortality and 95% C.I. (**a**) Red = cumulative effect; blue line = cumulative effect using state population as regression weights; green line = cumulative effect including regional month of sample fixed effects; gray dashed line = cumulative effect using Poisson model specification. (**b**) Solid black line = cumulative effect, equal to Fig. 2b; dashed lines splits the effect by 3 time period trends (1952–1973, 1974–1994, and 199–2015). (**c**) average TC incidence: Red line = *low average incidence*, first quartile of states with non-zero average TC incidence (equivalent to Fig. 2f). Blue shade lines = 2nd through 4th quartile state average TC incidence. Black line = *high average incidence* combined effect of the three upper quartiles with 95% C.I. shaded (same as in Fig. 2f). (**d**) average fraction of state population in coastal county: purple shade lines 1st through 4th quartiles, 1st and 2nd quartile combined. Solid black line = cumulative effect, and 95% C.I. shaded. (**e**) within-state correlation between population density and wind speed incidence: light orange below median and dark orange above median. Solid black line = cumulative effect and 95% C.I. shaded.

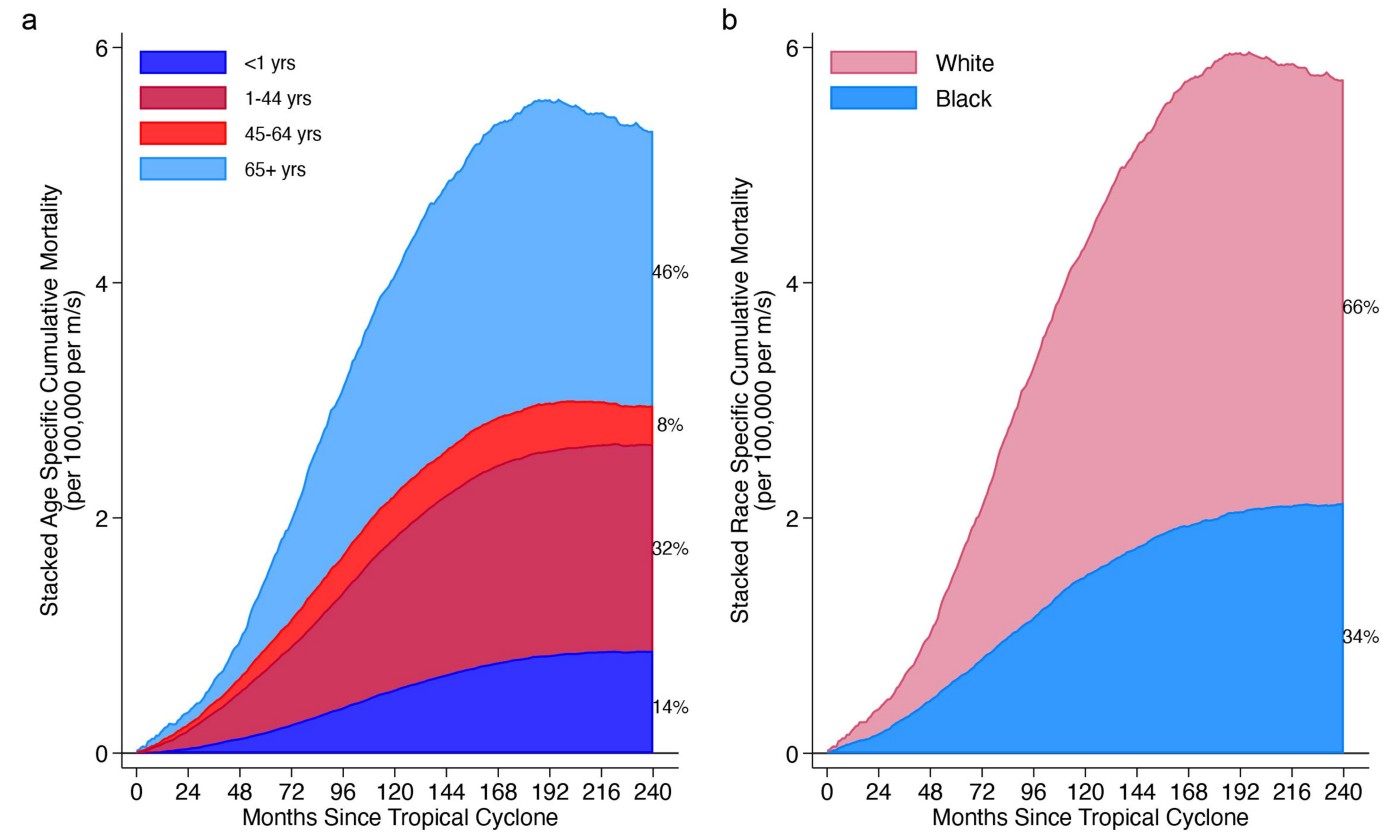

**Extended Data Fig. 7 | Stacked cumulative mortality by demographic groups.** (**a**) Stacks the cumulative effect of a tropical cyclone on all-cause mortality where each wedge corresponds to the effect for a specific age group. (**b**) The stacked summed effect for black and white individuals (all ages).

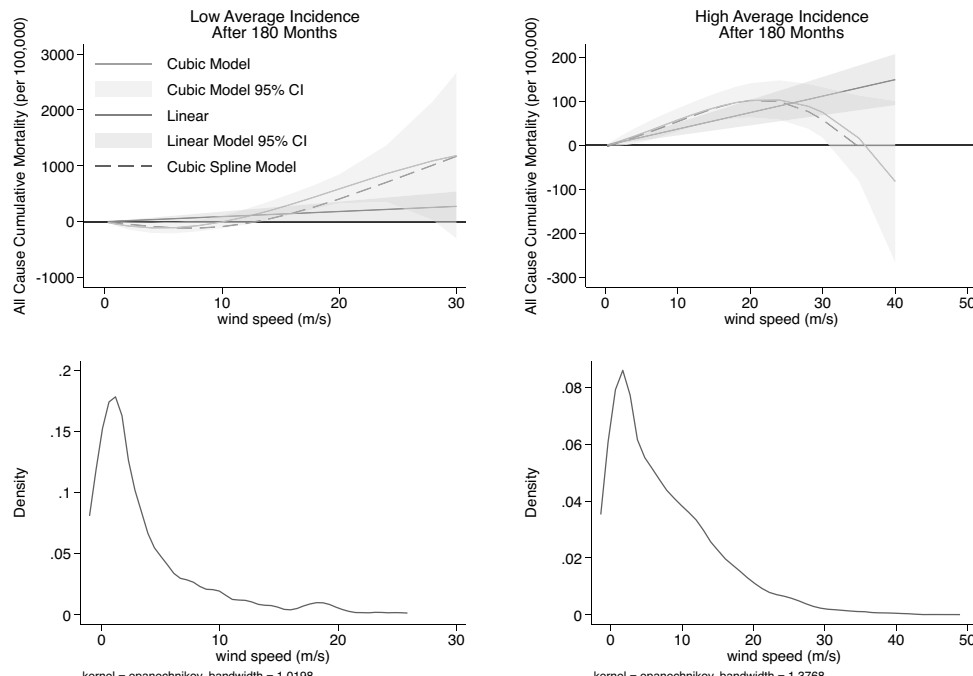

**Extended Data Fig. 8 | Excess mortality as a cubic function of wind speed.** TC mortality impulse response function is cubic in TC incidence in high and low average TC incidence states. Top row, light grey lines are effects as a function of wind speed (ms$^{-1}$) at $\ell = 180$, with 95% C.I. shaded. Dark grey lines are the same but with TC-mortality impulse response function is linear in TC incidence. Dashed lines are response functions from a cubic spline version of the model. The bottom row are kernel density plots of the TC incidence wind speeds in the low and high average incidence states.

**Extended Data Table 1 | Mortality burden estimated with nine models**

| | Linear | Quadratic | Cubic | Linear with Leads | Linear Adaptation with 4 groups | Linear Adaptation with 2 groups | Quadratic Adaptation with 4 groups | Quadratic Adaptation with 2 groups | Cubic Adaptation with 2 groups | Official Direct Deaths |
|---|---|---|---|---|---|---|---|---|---|---|
| Percent of Total Deaths (1950-2015) | 3.2% | 4.3% | 4.5% | 4.5% | 3.6% | 3.3% | 4.6% | 4.7% | 5.1% | 0.009% |
| Average Deaths/Month (1950-2015) | 4,613 | 6,293 | 6,554 | 6,533 | 5,141 | 4,803 | 6,648 | 6,810 | 7,349 | 50 |
| Average Deaths/Month (2000-2015) | 7,347 | 9,926 | 10,193 | 10,371 | 7,836 | 7,488 | 9,691 | 10,478 | 11,206 | 44 |
| Average Deaths per Storm | 7,172 | 9,785 | 10,191 | 10,158 | 7,993 | 7,468 | 10,337 | 10,589 | 11,427 | 24 (22 w/o Katrina) |
| Total TC Deaths (1950-2015) | 3.6 million | 4.9 million | 5.1 million | 5.1 million | 4.0 million | 3.8 million | 5.2 million | 5.3 million | 5.7 million | 11,937 |
| Adjusted R-squared | 0.9262 | 0.9270 | 0.9276 | 0.9245 | 0.9261 | 0.9262 | 0.9274 | 0.9268 | 0.9272 | - |

Total mortality burden (*mortality_burden*, see equation (6)), from all TCs across all states from 1950–2015, estimated with TC-mortality impulse response function linear, quadratic, and cubic, and with low and high average TC incidence states. The excess mortality calculations for each version of the model is presented in each column. "Linear" = excess mortality estimated with TC-mortality impulse response function linear in TC incidence, Eq. 7. "Cubic Adaptation with 2 groups" = excess mortality estimated with TC-mortality impulse response function cubic in TC incidence with high and low average incidence states, Eq. 8. "Official Direct Deaths" = Direct deaths from TCs are deaths that are officially attributed to a storm by the US government (see Data). "Percent of Total Deaths" = $\frac{mortality\_burden}{\sum_{i \in states} \sum_{t \in months} all\_cause\_mortality_{it}}$. "Average Deaths/Month" = $\frac{mortality\_burden}{\sum_{t \in months} t}$. "Average Deaths per Storm" = $\frac{mortality\_burden}{\sum_{s \in storms} s}$.

**Extended Data Table 2 | Mortality burden by demographic groups**

| | Average deaths per month (1960-2015) | Percent of deaths (1960-2015) | Total TC deaths (1960-2015) |
|---|---|---|---|
| Age <1 | 527 | 25.0% | 537,272 |
| Age 1 - 44 | 1,158 | 15.0% | 1,179,790 |
| Age 45 - 64 | 666 | 3.4% | 678,817 |
| Age >65 | 2,481 | 3.7% | 2,528,325 |
| Black | 2,019 | 15.6% | 2,057,149 |
| White | 2,561 | 3.1% | 2,609,418 |

Total mortality burden from all TCs across all states from 1960–2015, estimated with TC-mortality impulse response function linear in TC incidence, by age group and race ($mortality\_burden_a$ and $mortality\_burden_{race}$, where $race \in [Black, white]$). "Percent of Total Deaths" $= \frac{mortality\_burden_a}{\Sigma_{i \in states} \Sigma_{t \in months} all\_cause\_mortality_{it,a}}$. "Average Deaths/month" $= \frac{mortality\_burden_a}{\Sigma_{t \in months} t}$.

# Reporting Summary

## Statistics

For all statistical analyses, confirm that the following items are present in the figure legend, table legend, main text, or Methods section.

| n/a | Confirmed | |
|---|---|---|
| ☐ | ☒ | The exact sample size (*n*) for each experimental group/condition, given as a discrete number and unit of measurement |
| ☐ | ☒ | A statement on whether measurements were taken from distinct samples or whether the same sample was measured repeatedly |
| ☐ | ☒ | The statistical test(s) used AND whether they are one- or two-sided<br>*Only common tests should be described solely by name; describe more complex techniques in the Methods section.* |
| ☐ | ☒ | A description of all covariates tested |
| ☐ | ☒ | A description of any assumptions or corrections, such as tests of normality and adjustment for multiple comparisons |
| ☐ | ☒ | A full description of the statistical parameters including central tendency (e.g. means) or other basic estimates (e.g. regression coefficient) AND variation (e.g. standard deviation) or associated estimates of uncertainty (e.g. confidence intervals) |
| ☐ | ☒ | For null hypothesis testing, the test statistic (e.g. $F$, $t$, $r$) with confidence intervals, effect sizes, degrees of freedom and $P$ value noted<br>*Give P values as exact values whenever suitable.* |
| ☒ | ☐ | For Bayesian analysis, information on the choice of priors and Markov chain Monte Carlo settings |
| ☐ | ☒ | For hierarchical and complex designs, identification of the appropriate level for tests and full reporting of outcomes |
| ☐ | ☒ | Estimates of effect sizes (e.g. Cohen's *d*, Pearson's *r*), indicating how they were calculated |

*Our web collection on statistics for biologists contains articles on many of the points above.*

## Software and code

Policy information about availability of computer code

| Data collection | Code was not used to collect data for the study. |
|---|---|
| Data analysis | Stata 14, Matlab 2019a, R 4.2.0 |

For manuscripts utilizing custom algorithms or software that are central to the research but not yet described in published literature, software must be made available to editors and reviewers. We strongly encourage code deposition in a community repository (e.g. GitHub). See the Nature Portfolio guidelines for submitting code & software for further information.

## Data

Policy information about availability of data

All manuscripts must include a data availability statement. This statement should provide the following information, where applicable:
- Accession codes, unique identifiers, or web links for publicly available datasets
- A description of any restrictions on data availability
- For clinical datasets or third party data, please ensure that the statement adheres to our policy

All data is available for download here: https://zenodo.org/uploads/10459719.

The full data processing code is not included but the collected full data set needed for the main analysis (figures 1-4) is provided in DATA_hurricane_mortality_temp_month_state_19302015.dta }. This includes the matched LICRICE generated TC wind speed and pddi; the all-cause mortality data

from the Center for Disease Control and Prevention (CDC) Mortality Statistics of the United States annual volumes, the Multiple Cause of Death (MCOD) files, and Underlying Cause of Death database; the population data from the Inter-university Consortium for Political and Social Research and the US Census Bureau, Intercensal Population and Housing Unit Estimates; and the temperature data from the Berkeley Earth Surface Temperatures (BEST).

We also provide the follow datasets required for the analysis:

- Shapefiles of the U.S. states : cb_2016_us_state_20m
- LICRICE generated TC wind speed and pddi by state and month : panel_by_storm__NA_USA_density_8_yr_1930_2018.csv
- NOAA TC direct deaths : directdeaths.csv
- Nordhaus TC damages and LICRICE, national : nordhaus_LICRICE_USA_merged.dta
- TC rainfall data : rainfall_idw_state_storm.csv
- CDC mortality data for all states : mortality_19002015.dta (needed for SI figures)
- Counties on the coastline : coastline-counties-list.xlsx (needed for SI figures)
- County population by age : us.1969_2020.19ages.adjusted.txt (needed for SI figures)
- List of TC names : storm_list.txt
- Wind speed and population by pixel : wind_state_pop_export.csv (needed for Figure SI13e)
- LICRICE all storms pixel-level : NA_USA_density_8_yr_1930_2018_storm_specific.mat (needed for Figure SI6)
- Shapefile of US states : plotting_maps/s_11au16/s_11au16.shp (needed for Figure SI1)
- Hurricane direct death data : directdeaths.csv (needed for Figure 3)
- TC unique serial number and storm name : storm_id_name_raw.mat (needed for Figure 3)
- TC unique serial number : stormnamelist.mat (needed for Figure 3)

# Research involving human participants, their data, or biological material

Policy information about studies with human participants or human data. See also policy information about sex, gender (identity/presentation), and sexual orientation and race, ethnicity and racism.

| Reporting on sex and gender | Information on sex and gender were unavailable for analysis. |
|---|---|
| Reporting on race, ethnicity, or other socially relevant groupings | In one of our analyses, we stratify the sample based on whether individuals are identified as "Black" or "White", excluding other races, because these two categories are the only consistent categories available in our sample. These are recorded in administrative data maintained by the US Centers for Disease Control. |
| Population characteristics | See above |
| Recruitment | No recruitment was done for the study |
| Ethics oversight | We received a waiver from the UC Berkeley IRB |

Note that full information on the approval of the study protocol must also be provided in the manuscript.

# Field-specific reporting

Please select the one below that is the best fit for your research. If you are not sure, read the appropriate sections before making your selection.

☐ Life sciences  ☒ Behavioural & social sciences  ☐ Ecological, evolutionary & environmental sciences

For a reference copy of the document with all sections, see nature.com/documents/nr-reporting-summary-flat.pdf

# Behavioural & social sciences study design

All studies must disclose on these points even when the disclosure is negative.

| Study description | Quantitative analysis of longitudinal administrative data combined with reconstructions of physical geophysical events. |
|---|---|
| Research sample | Sample is representative of the contiguous United States because it includes all individuals living in the contiguous United States during 1930-2015. We chose this study sample because of data availability. Detailed all-cause mortality data is not available for this sample period outside of the United States. All cause mortality data was collected from the Center for Disease Control and Prevention (CDC) Mortality Statistics of the United States annual volumes, the Multiple Cause of Death (MCOD) files, and Underlying Cause of Death database (2017). State population data was combined from Inter-university Consortium for Political and Social Research and the US Census Bureau, Intercensal Population and Housing Unit Estimates. |
| Sampling strategy | Our sample is comprehensive administrative data, representing the universe of recorded deaths in the contiguous United States during the period of study. We included data from 1930-2015 because it was all of the publicly available digitized data. This long time period allows us to estimate the lagged effects for 20 years after a TC. |
| Data collection | Administrative data on all-cause mortality and population is collected by local, state, and federal government agencies. The data collection did not involved experimental conditions or a study hypothesis. |

| Timing | Monthly data during 1930-2015. |
|---|---|
| Data exclusions | No data is excluded. |
| Non-participation | No participants were involved in the study. |
| Randomization | The research design assumes that the timing and intensity of tropical cyclone incidence within a given location is as good as randomly assigned to populations. All quasi-experimental comparisons are within a population over time and outcomes are not compared across locations. |

# Reporting for specific materials, systems and methods

We require information from authors about some types of materials, experimental systems and methods used in many studies. Here, indicate whether each material, system or method listed is relevant to your study. If you are not sure if a list item applies to your research, read the appropriate section before selecting a response.

## Materials & experimental systems

| n/a | Involved in the study |
|---|---|
| ☒ | ☐ Antibodies |
| ☒ | ☐ Eukaryotic cell lines |
| ☒ | ☐ Palaeontology and archaeology |
| ☒ | ☐ Animals and other organisms |
| ☒ | ☐ Clinical data |
| ☒ | ☐ Dual use research of concern |
| ☒ | ☐ Plants |

## Methods

| n/a | Involved in the study |
|---|---|
| ☒ | ☐ ChIP-seq |
| ☒ | ☐ Flow cytometry |
| ☒ | ☐ MRI-based neuroimaging |

## Plants

| Seed stocks | n/a |
|---|---|
| Novel plant genotypes | n/a |
| Authentication | n/a |

