## [Peer Review File · Nature]

Manuscript Title: Mortality Caused by Tropical Cyclones in the United States

Reviewer Comments & Author Rebuttals

Reviewer Reports on the Initial Version:

Referees' comments:

Referee #1:

As the frequency and magnitude of natural disasters increase globally, quantifying their effects on population mortality is of vital importance to public health agencies. Large swaths of machine-readable data captured in the 21st century have significant potential to support the provision of immediate aid, and the development of resilient health policy. As countries globally continue to develop policy responses to climate change, it remains increasingly important to leverage retrospective data to identify factors that have short- and long-term effects on population well-being.

The current study develops a long-run estimate for the overall effect of individual TCs on all-cause mortality within the CONUS. Specifically, the authors use a reduced-form econometric approach to identify and attribute the causal effect of all TCs between 1930-2015 on human mortality (both direct and indirect deaths), accounting for the delayed, overlapping, and long-run effects of TCs. Overall, the presented research question was well defined, and the authors successfully justified the role of their analysis as a first attempt at a long-run estimate amongst the existing evidence base within the field. They carefully catalogue existing theoretical relationships and previously used methods that apply to their research question, leverage appropriate data sources, and justify their selected approaches to data processing and analysis, with sufficiently presented placebo tests, stratified sample results, and sensitivity analyses. The authors' graphical representation of results, commentary around their importance and validity, and grounding in literature were excellent and appropriate to the journal readership.

However, there remain a few open questions around the methods applied during processing and analyses that may help clarify the interpretation of their estimates, as well as minor in-text and graphical errors.

Major Comments

1. The authors note abrupt changes after 2001 in lines 263-274, and attribute an abrupt increase in deaths/month to observed increased frequency of TCs. They also analyze the effects of TCs on different causes of death. However, cause-of-death attribution may also be affected by changes in documentation practices in the underlying data, specifically the MCOB's shift in use of ICD-9 to ICD-10 1999 onwards. Notably, this change was accompanied by an increase in deaths attributed to infectious disease, and cardiovascular disease among others (see Comparability of Cause of Death Between ICD-9 and ICD-10: Preliminary Estimates. National Vital Statistics Report 2001; Volume 49,

Number 2).

2. In several places, the authors mention that TC wind speed is a reasonable indicator to use for the impact of TCs overall and that they are not parametrizing storm surge/rainfall. To what extent is wind speed correlated with those other potential metrics of TC damage? This may not be possible to do for the full historical record, but if we are to interpret the results as the effects of TCs rather than high winds, then some indication of correlation with flood-related TC damage would be valuable.

3. On a related note to #2, there are high wind events that are not associated with TCs. If the key channels for mortality are mostly through wind, then we might expect those non-TC wind events to have a similar impact; on the other hand, if the flood impacts are important, then they wouldn't. It would be helpful for the authors to discuss the choice to only consider TC-related winds and the implications for interpretation.

4. In the case of TCs that were contemporaneous or occurring around the same time-period with overlapping signals, the authors use deconvolution to recover single-TC impulses and estimate the effects of a singular TC. Regarding the models that are non-linear models for TC incidence lines 879-895, would non-linearity of effects violate necessary assumptions made for deconvolution – i.e., does modeling mortality as a non-linear function of TC incidence lead to non-separability and non-additivity of individual TC effects? Are the non-linear models presented as a robustness test but not used in any of the other reported results?

5. To what extent do differences in the population distribution within states affect the results + associated interpretation? The share of the population on the coast of Florida is quite different than North Carolina, for example. The shift in the distribution over time may also contribute to the observed (non)-adaptation over time within states.

Minor comments

1. The authors present pooled 95% confidence intervals for their final models. Given the observed heterogeneity in mortality responses across states due to varying exposure to TCs, it would be useful to clarify whether these 95% CI are calculated using cluster-robust standard errors at the state-level to account for this.

2. There is one study of mortality post-Maria that is in the references, but the result seems like the most comparable point estimate (there may be other Maria papers that are also relevant here), so I suggest that the authors actually use the result to contextualize their own findings. While clearly a different type of hazard, there is also some long-term mortality evidence from the Indonesian tsunami (<https://pubmed.ncbi.nlm.nih.gov/29731526/>, <https://pubmed.ncbi.nlm.nih.gov/33681474/>).

3. The research design does not allow for estimation of different effect sizes due to temporal compounding (i.e., the possibility that a second TC shortly after a first TC has a worse impact), as far as I can tell. This should be noted in the limitations.

4. The Abrupt changes after 2001 paragraph is difficult to read and contains several typos.
5. Line 819: The reference to Fig. 1c may be erroneously labelled; the authors may have intended to reference Fig. 1e.
6. The legend in Fig. S9c plots “quantiles”. The text refers to “quartiles”, which is likely what the authors intended on communicating.
7. Lines 874-875: Minor sp. inconsistency for “fourth/forth”.

Referee #2:

Summary of the key results:

Post-tropical cyclone (TC) mortality over a very long period (1930-2015) was evaluated for the entire continental US (CONUS) region. The paper estimates total mortality – not just mortality directly attributed to TCs and occurring in the immediate aftermath of a storm, but all excess mortality attributed to 501 TCs that reached CONUS over this period. They report an average of 7,000-11,000 total excess deaths per storm, occurring over fifteen years. This is far more than the 24-death estimate by the US Government based on direct attribution in the immediate aftermath of a TC. They conclude that this heretofore uncalculated mortality burden comprises a substantial fraction of mortality rates along the Atlantic coast, around 3.2-5.1% of all deaths. They identified populations at the highest risk: infants, people ages 1-44, and Black people.

Originality and significance:

The work is both original and highly significant.

Data & methodology:

The authors have employed a valid, entirely original approach and, in my view, very effectively allowed them to test their findings in the complete absence of any control group. The presentation of results is beautiful; excellent charts and graphics make it easier to understand their methodologies and present their findings and statistical uncertainties.

Storm Data: I am concerned about using the Limited Information Cyclone Reconstruction and Integration for Climate and Economics (LICRICE) model as the only surrogate for TC-related exposures to human populations. As you note, many TCs with lower windspeed have had substantial population impacts because of flooding and power outages. This metric may be underestimating the impacts of TCs on mortality. Averaging maximum windspeed over the entire land area of each state results in a picture (Fig. S12) that appears to show Connecticut with much higher TC severities than Texas (and the label for the figure does not say that the metric was averaged over land area).

Mortality Data: The most impressive increase in mortality was for “other” causes. The CDC’s

underlying cause of death data does not include the “other” category, which is, in this paper, a grab bag containing diabetes and all the other diseases that were not assessed separately.

[See the following from CDC: how ICD codes have (and have not) been consistent over the years: US Centers for Disease Control and Prevention. (2015). Comparability across revisions for selected causes. Retrieved from <https://www.cdc.gov/nchs/data/dvs/comp2.pdf>.]

It must be that the authors themselves created this category, which may have come about because they used data from the authors they cite for their mortality database.

[A. Barreca, K. Clay, O. Deschenes, M. Greenstone, and J. S. Shapiro, “Adapting to climate change: The remarkable decline in the US temperature-mortality relationship over the twentieth century,” *Journal of Political Economy*, vol. 124, no. 1, pp. 105–159, 2016.]

The Barreca mortality analysis focused on the impacts of air conditioning on heat-related mortality for five underlying causes of death: cardiovascular, respiratory, motor vehicle injuries, infectious, and neoplasms, the same specific underlying causes as in this manuscript. Also, as a correction, Barreca et al. only digitized some of the data from prior to 1959. (Bacerra, in an earlier paper, describes digitizing the earlier data.) Data from 1959 onward are downloadable from the NBER and other sources:
<https://www.nber.org/research/data/mortality-data-vital-statistics-nchs-multiple-cause-death-data>.

Given that the “other” causes predominate, it would be preferable to disaggregate these deaths and examine, at least, the significant causes of death that comprise the “other” category. The lack of identification of individual underlying causes makes it difficult to determine what, if anything, might have been done to prevent these deaths. Some significant causes which may be important in the aftermath of TCs include diabetes; intestinal infections (infants); chronic liver disease and cirrhosis; chronic kidney disease, homicide, drug poisoning; suicide; unintentional injuries (other than a motor vehicle); congenital malformations (infants); and conditions originating in the perinatal period (infants). Possibly the authors could undertake a sub-analysis for just the years from 1959 onward if they cannot obtain from Bacerra the complete data set that includes data from the period between 1930 and 1958 (or cannot, themselves, digitize the earlier data, which are available from the NCHS). As another possibility, the authors could remove the cause-specific mortality analysis and conduct a more thorough review in a second publication.

Appropriate use of statistics and treatment of uncertainties:

As was implied above, I am very favorably impressed by how the authors used statistics and identified statistical uncertainties.

Conclusions: robustness, validity, reliability:

Mortality is the totality of health impacts from TCs; one or more adverse health events precede every death. It would be expected that not all these events would cause premature mortality. This indicator should be discussed within the context of broader public health impacts, not all of which

resulted in death. Assessing mortality alone significantly underestimates the overall public health burden of TCs in CONUS.

The authors noted migration as an issue that could not be addressed in this long-range and state-by-state analysis. This reviewer was privileged to do a study of mortality after Hurricane Maria in Puerto Rico in which we had access to flight logs that allowed for understanding population flux over several years. As demographers know, people do not migrate randomly. Very likely in the context of this study is that the inability to account for migration would underestimate risk. First, people whose lives and livelihoods are most impacted may be more likely to leave coastal areas for opportunities elsewhere. These moves may be within the same state, but if not, their subsequent mortality will be lost to analysis. Second, people moving to the coasts from other parts of the country will be counted in the population data upon which mortality risks are computed, yet, if they are moving in from non-TC impacted states, they lack exposure to TCs. Both of these situations would result in underestimating mortality risk in TC-impacted states.

Santos-Burgoa, C., Sandberg, J., Suarez, E., Goldman-Hawes, A., Zeger, S., Garcia-Meza, A., ... Goldman, L. R. (2018). Differential and persistent risk of excess mortality from Hurricane Maria in Puerto Rico: a time-series analysis. *Lancet Planet Health*, 2, e478-e488. doi:10.1016/S2542-5196(18)30209-2.

The discussion of black/white racial inequities could benefit from the citation of literature documenting how historical racial inequality (e.g., redlining in housing markets) has placed black coastal communities in harm's way (i.e., in more flood-prone areas, in housing that is less protective and in communities where there has been a failure to invest in levees and other community level protective measures). The authors should be aware that HHS, EPA, FEMA and the states are increasingly attempting to identify communities at highest risks using social vulnerability indices. I am not suggesting that such data are available back to 1930, but these issues are as, or more, significant than those included in the discussion. To cite but a few examples of literature that might be cited (but there is much more):

US EPA. (2021). *Climate Change and Social Vulnerability in the United States: A Focus on Six Impacts*. (EPA 430-R-21-003). Washington, DC Retrieved from www.epa.gov/cira/social-vulnerability-report

Hardy, R. D., Milligan, R. A., & Heynen, N. (2017). Racial coastal formation: The environmental injustice of colorblind adaptation planning for sea-level rise. *Geoforum*, 87, 62-72. doi:<https://doi.org/10.1016/j.geoforum.2017.10.005>

Tee Lewis, P. G., Chiu, W. A., Nasser, E., Proville, J., Barone, A., Danforth, C., ... Craft, E. (2023). Characterizing vulnerabilities to climate change across the United States. *Environment International*, 172, 107772. doi:<https://doi.org/10.1016/j.envint.2023.107772>

Morello-Frosch, R., & Obasogie, O. K. (2023). The Climate Gap and the Color Line — Racial Health Inequities and Climate Change. *New England Journal of Medicine*, 388(10), 943-949. doi:10.1056/NEJMs2213250

Implications: In the discussion, the authors are focused on providing health care to address these regional inequities. They should also consider including discussions of how prevention activities could, in the future, contribute to preventing the excess mortality that they are documenting. They

did not present evidence that there are too few such facilities in the region. To the extent this is true, one might recognize the lack of demand for services caused by people who lack adequate healthcare coverage, and historically and even currently, as many coastal states have refused to expand Medicaid. However, there is a diversity of policies and doctor: patient ratios across the CONUS TC impacted states, and more likely other factors should be considered. As an alternative, the authors may consider the role of prevention, particularly in the context of stress, as a causal factor in several diseases that lead to premature mortality over many years, namely hypertension (cardiovascular disease, chronic renal disease, poor pregnancy outcomes); diabetes (via stress-induced mechanisms as well as the impacts of storms on food systems and perhaps eating behaviors); disruption of HPA axis relationships and the profound impact of that across populations on birth outcomes and infant mortality even when the stress occurs preconceptionally. (Hopefully, the implication of this is not lost on the authors when discussing the high rates of infant mortality, years after TCs, that they have documented.) The implications of this are not to build more hospitals but to address factors like hypertension and blood sugar control, psychosocial stress, and of course, anything in the system that can be done to reduce stress and uncertainty in the environment around people. Only treating the heart attack, or a child born preterm, is not an adequate response.

Class, Q. A., Khashan, A. S., Lichtenstein, P., Långström, N., & D'Onofrio, B. M. (2013). Maternal stress and infant mortality: the importance of the preconception period. *Psychol Sci*, 24(7), 1309-1316. doi:10.1177/0956797612468010

Ghosh, A. K., Demetres, M. R., Geisler, B. P., Ssebyala, S. N., Yang, T., Shapiro, M. F., . . . Abramson, D. (2022). Impact of Hurricanes and Associated Extreme Weather Events on Cardiovascular Health: A Scoping Review. *Environ Health Perspect*, 130(11), 116003. doi:10.1289/ehp11252

Cohen, S., Janicki-Deverts, D., & Miller, G. E. (2007). Psychological Stress and Disease. *Jama*, 298(14), 1685-1687. doi:10.1001/jama.298.14.1685

Duthie, L., & Reynolds, R. M. (2013). Changes in the Maternal Hypothalamic-Pituitary-Adrenal Axis in Pregnancy and Postpartum: Influences on Maternal and Fetal Outcomes. *Neuroendocrinology*, 98(2), 106-115. doi:10.1159/000354702

Suggested improvements: experiments, data for possible revision

Ideally, as noted above, there would be the possibility of disaggregating the "Other" mortality category used to analyze underlying causes of death. I think that other suggestions are more relevant to the conclusions.

Several references are suggested.

Clarity and context: lucidity of abstract/summary, appropriateness of abstract, introduction, and conclusions

In conclusion, this paper is ground-breaking and incredibly important to understanding the public health significance of tropical cyclones. It is very well written. Though it may surprise many that TCs are such a significant risk factor for mortality, as noted above, if anything, this paper provides a modest estimate of public health risk. From the outset, collaboration with an epidemiologist would have benefitted the work. Although I suggested the possibility of a second paper to improve the

epidemiology analysis, I very much like the scope and breadth of this paper and hope it is feasible for the authors to address these concerns! Congratulations to the authors for a great contribution!

Referee #3:

This paper estimates the number of excess deaths in the 20 years following a tropical cyclone exposure. The premise of this paper is significant. However, this paper has several limitations that need further analysis. Below please find my specific comments.

1. A few times in the manuscript the authors state the assumption that TC exposure is random. This is clearly not true, though, as TCs have very distinct spatial and temporal patterns. For example, June in Texas is not exchangeable with August in North Carolina. More sensitivity analyses to evaluate the robustness of the results to this assumption are definitely warranted.

2. I am concerned about the statewide analysis. In addition to differences in TC exposure across states, there are distinct patterns within each state as well (e.g., western NC does not experience the same TC exposure as eastern NC, and the populations living in these two areas are quite different as well). I understand that the authors do not have access to finer-resolution mortality data that go back to the beginning of the study period. However, such data (at count and month level) are available since the mid-80's (if not earlier). Therefore, a sensitivity analysis at the county level for those years for which data are available, and comparison with the main statewide analysis, is warranted.

3. Fig. 1b: Why do the x-axes only go back to 1950?

4. Fig. 1, "cumulative monthly TC maximum wind speed": What does this mean and how was it estimated? And what is its relevance to the rest of the paper? That is, why not show average speeds?

5. Fig. 1e: What is plotted on the x-axis exactly? What are the "cumulative effects"?

6. Lines 115-116: How common is it for a state to experience monthly average winds of 6.9 m/sec when not exposed to TC? How does this impact interpretation of results?

7. Lines 166-168: But aren't these results surprising, especially given other recent publications (e.g., Parks et al., JAMA, 2022)?

8. Lines 174-175: But are these estimates statistically different from each other? I am not sure that they are given the 95% CIs. If not, this finding is overinterpreted, please tone down.

9. Nonlinear TC effects: It is not clear to me which model (linear vs nonlinear) actually yielded a better fit. How did the authors compare the two models? Whichever model yielded the best fit is the one that should be used for the calculation of excess deaths.

10. What is the predictive accuracy of LICRICE?
11. Building up on comment #2: Assigning a monthly statewide average of TC wind speeds is a bit concerning for me, and I am worried that it may bias results quite a bit.
12. Line 645: Why not 1-18 and 19-44?
13. Eqn 6: What is K?
14. Lines 744-746: Wouldn't that depend on storm intensity and characteristics and could thus vary across storms? Was this taken into account?
15. Why use an OLS for rates and not a (quasi-)Poisson or negative binomial model?
16. Why were there not more lags of temperature considered?
17. The temperature-mortality association is not really quadratic, though. Why not use a more flexible form (e.g., a 3- or 4-df natural spline)?
18. Eqn 8: What is the difference between m_t and t ?
19. Line 786: What is h : nationwide monthly mortality rate?
20. Lines 786-787: But these outbreaks tend to be more localized and not nationwide. Why not use regional terms instead?
21. Line 791: How was this R^2 estimated? Cross-validation? If not, it may be just due to overfitting.
22. Line 797: Is the period of 72 months prior to TC exposure included as a negative exposure control? Please clarify.
23. Lines 865-878: I am not sure I understand the purpose of this analysis or how it was actually conducted. How is this different from the association being nonlinear?
24. It is not clear to me if there is statistical interaction by quartile of incidence or not in the end.
25. Eqn 9: Why cubic and not a more flexible form (e.g., natural or penalized splines)?

Author Rebuttals to Initial Comments:

Reviewer comments in black

Our replies in blue

New text in the manuscript in red

Reviewer #1:

As the frequency and magnitude of natural disasters increase globally, quantifying their effects on population mortality is of vital importance to public health agencies. Large swaths of machine-readable data captured in the 21st century have significant potential to support the provision of immediate aid, and the development of resilient health policy. As countries globally continue to develop policy responses to climate change, it remains increasingly important to leverage retrospective data to identify factors that have short- and long-term effects on population well-being.

The current study develops a long-run estimate for the overall effect of individual TCs on all-cause mortality within the CONUS. Specifically, the authors use a reduced-form econometric approach to identify and attribute the causal effect of all TCs between 1930-2015 on human mortality (both direct and indirect deaths), accounting for the delayed, overlapping, and long-run effects of TCs. Overall, the presented research question was well defined, and the authors successfully justified the role of their analysis as a first attempt at a long-run estimate amongst the existing evidence base within the field. They carefully catalogue existing theoretical relationships and previously used methods that apply to their research question, leverage appropriate data sources, and justify their selected approaches to data processing and analysis, with sufficiently presented placebo tests, stratified sample results, and sensitivity analyses. The authors' graphical representation of results, commentary around their importance and validity, and grounding in literature were excellent and appropriate to the journal readership.

Our reply: We thank the reviewer for their thoughtful evaluation of our analysis.

However, there remain a few open questions around the methods applied during processing and analyses that may help clarify the interpretation of their estimates, as well as minor in-text and graphical errors.

Major Comments

1. The authors note abrupt changes after 2001 in lines 263-274, and attribute an abrupt increase in deaths/month to observed increased frequency of TCs. They also analyze the effects of TCs on different causes of death. However, cause-of-death attribution may also be affected by changes in documentation practices in the underlying data,

specifically the MCODE's shift in use of ICD-9 to ICD-10 1999 onwards. Notably, this change was accompanied by an increase in deaths attributed to infectious disease, and cardiovascular disease among others (see Comparability of Cause of Death Between ICD-9 and ICD-10: Preliminary Estimates. National Vital Statistics Report 2001; Volume 49, Number 2).

Our reply:

First, we note that the rapid increase in mortality referenced by the reviewer is an increase in estimated excess mortality after 2001, i.e. an increase in the computed mortality burden that we construct based on the results of our econometric analysis. Changes in documentation practices in 1999 in the underlying data may have affected our econometric analysis, but those changes could not have driven an abrupt change in our estimate of excess mortality simply because the computation of excess mortality does not allow the underlying econometric model to change that abruptly. Thus, to address this issue and avoid future readers experiencing confusion on this point, we have retitled the subsection to “**Accelerating trend in excess mortality** after 2001” to be clear that this is not a trend in raw mortality data.

Second, we thank the reviewer for the valuable note about the shift in cause of death methodology from the CDC. We had been previously aware of various changes to the CDC's methodology over time and had tried to address these concerns by including national month-of-sample fixed effects that account for any systematic changes in underlying data collection methodology that occur at the national (i.e. CDC) level. The inclusion of this model element was described in the original text, however we did not explicitly state that it would address changes in data collection over time. Thus, to clarify this point, we have revised the text describing this model element to now read:

Revised text from line 880-885:

*“National month-of-sample fixed effects that capture nonlinear and/or discontinuous changes in mortality rates nationwide ($\mu_3 \cdot h_t$). These terms are particularly important for capturing idiosyncratic spikes in mortality that result from nation-wide conditions, such as influenza outbreaks **as well as any systematic changes in the accounting methodology of mortality by the CDC.** Comparisons of Panel b and c of Figure S18 illustrates how the inclusion of these terms in the model alters the ability of the model to capture unusual spikes in mortality that are not captured by other model elements, including the trends listed above, TCs, and temperature.”*

And we now also explicitly detail the changes in CDC data collection methodology flagged by the reviewer in our description of the CDC data:

Revised text from line 724-732:

“For the years 2005 - 2015, we analyze mortality data from the public Center for Disease Control and Prevention (CDC) Underlying Cause of Death database (2017). The data is based on death certificates for U.S. residents, which gives a single underlying cause of death and demographic data. Cause of death prior to 2000 was indexed using the 4-digit ICD-9 code and 2000 onwards the index changed to the 4-digit ICD-10 code. We harmonized the cause of death into 5 categories that matched the Barreca et al. cause of death variables. We also construct a 6th category which is the difference between all-cause mortality and the sum of the 5 cause-specific categories, called “other”. Notably, the change in CDC ICD code methodology resulted in a shift in the counts of deaths from specific causes, particularly infectious diseases and cardiovascular disease.”

Finally, we have rerun our analysis, explicitly checking for the impact of this change in the accounting of deaths from infectious diseases in 2000. First, we estimated the effect of TCs on infectious diseases, excluding years after 1999 (shown below, left panel, blue line). In our original analysis, we estimated that on average throughout our sample, infectious diseases did not exhibit a significant nor substantial response to TCs (Fig 2e of manuscript, also green line below). We find that prior to 2000, TCs *reduced* the number of excess deaths attributed to infectious disease in the years after the storm, presumably because individuals that would have died of infectious disease (in the absence of the storm) died of an alternative cause.

Second, we also estimated the effect of TCs on all cause mortality, removing infectious diseases (total mortality minus mortality from infectious diseases). These results are shown below in the right panel, and should be compared to our main results in Fig 2b of the main text. We find that removing infectious diseases does not alter our main estimates for all cause mortality responses. This finding appears reasonable, since infectious diseases are only a small fraction of overall mortality (both before and after 2000) and it does not exhibit a substantial systematic response to TCs.

Thus, overall, we find that the increase in excess mortality after 2000 is not driven by a change in reporting of infectious diseases at that time. Further, this change in reporting does not alter our main findings.

We thank the reviewer for encouraging us to check these points.

2. In several places, the authors mention that TC wind speed is a reasonable indicator to use for the impact of TCs overall and that they are not parametrizing storm surge/rainfall. To what extent is wind speed correlated with those other potential metrics of TC damage? This may not be possible to do for the full historical record, but if we are to interpret the results as the effects of TCs rather than high winds, then some indication of correlation with flood-related TC damage would be valuable.

Our reply:

We thank the reviewer for this important question. We now have conducted extensive new analysis to address this concern.

It is nearly impossible to validate the wind speed data against ground-based wind speed measures, since most wind data collection systems are not designed to withstand/measure TC strength winds. However, we are able to evaluate the extent to which our measure of wind speed correlates with rainfall during storms and overall reported damages from a significant prior study (Nordhaus, 2010). [Although, we note that a similar evaluation of storm surge is beyond the scope of this analysis but is the subject of follow-up work we are conducting along with a large team of collaborators.]

We examine whether wind speed is associated with rainfall by reconstructing the rainfall field for a limited number of TCs where storm-specific gauge data is available. We emulate the approach of NOAA in constructing rain fields by interpolating rainfall, and then we compare these fields against the wind fields of the same storms. Our new Figure SI6, pasted below, illustrates this comparison for six example storms and reports state-level correlations between our measure of wind speed exposure and an analogously computed rainfall exposure. The r-squared of these correlations range from 0.36 to 0.74, indicating that at the level of our analysis, wind speed is a fairly good proxy for rainfall within a storm, as hypothesized in our original manuscript.

Figure S16. Tropical cyclone rainfall and wind speed incidence comparison. Left panel, NCEP tropical cyclone rainfall weather station data (inches) inverse squared distance weighted interpolation at each $1^\circ \times 1^\circ$ pixel. Right panels, LICRICE modeled cumulative maximum wind speed ($m s^{-1}$) from tropical cyclones at each $0.1^\circ \times 0.1^\circ$ pixel. R-squared values from the linear correlation between state area average rainfall and state maximum wind speed for each storm.

We cannot conduct this analysis for all storms, but we now pool the data from all storms for which rainfall data is available and estimate an average association with r -squared = 0.31, indicating that state-level wind speed is a reasonable, albeit imperfect, proxy for

state-level rainfall. This result is now presented in a new figure panel of the main text (Figure 1e):

Figure 1e: Association between state-by-storm average rainfall (inches) from NOAA station data and maximum wind speed ($m s^{-1}$) computed using LICRICE (263 storms, $N=12,889$, slope=0.116, R -squared=0.307).

This finding is now reflected in the main text, which has been revised to state:

Revised text from line 675-679:

LICRICE does not explicitly model storm surge, rainfall or flooding, however these dimensions of impact are captured in the analysis to the extent that they are correlated with integrated maximum wind speeds. For example, we find that LICRICE wind speed and NCEP rainfall from TCs are correlated within storms at the pixel level (Figure SI6) and at the state-by-storm level (r -squared = 0.31, Figure 1e) for a limited sample of storms where granular rainfall data is available.

We have also evaluated the extent to which our measure of wind speed is correlated with estimates for total direct TC damage, i.e. monetary losses that occur during the storm event. Total storm damage is notoriously difficult to estimate, since many damages are to public infrastructure, are not insured, or are insured by various disparate entities. Thus, there is no definitive record of total damage and it is widely known that all efforts to compute such damages are coarse approximations. Nonetheless, we obtain total damage estimates from a prominent study by Nordhaus (2010) in which the author undertook extensive efforts to correct for many known

sources of bias. We then evaluate the extent to which LICRICE wind metrics predict the log of total damages as a fraction of GDP – a standard, but imperfect, normalization used throughout the literature. Because Nordhaus’s estimates of damage are national, we aggregate wind nationally and evaluate associations at the storm level, as well as aggregated across all storms reported by Nordhaus in each year (which is not a complete record).

These associations are shown in new Figure SI5, pasted below. Within a broad class of simple models, we find that the fit is maximized by a log-linear model that is a piecewise linear spline with a single kink. Above the kink, average damages appear to be a constant fraction of GDP, regardless of estimated higher wind speed exposure for a few outlying storms. The model fit is $r^2=0.39$ across storms and $r^2=0.32$ across years.

Figure SI5. Tropical cyclone damage and wind speed incidence comparison. (a) Total logarithm tropical cyclone national economic damage per national gross domestic product (GDP), assembled for Nordhaus 2010, of tropical cyclones between 1950-2005 and national average maximum wind speed ($m s^{-1}$) estimated from Limited Information Cyclone Reconstruction and Integration for Climate and Economics (LICRICE). Discrete spline estimated relationship, split at 1.2 maximum wind speed ($m s^{-1}$). (b) corresponding annual tropical cyclone log damage per GDP and national average maximum wind speed ($m s^{-1}$).

We also now present the linearly increasing segment of the storm-level analysis in the main text in a new panel Figure 1d, pasted below:

Figure 1d: Correlation between log TC damage / GDP for each storm computed by Nordhaus (2010) and national average maximum wind speed (m/s) computed using LICRICE (1950-2005, N=89, slope=4.87, R-squared=0.36).

And have updated the main text to report this result:

Revised text from line 679-685:

Iterations of the LICRICE model has been used to measure various social and economic impacts of TCs including direct deaths and damages (Hsiang & Narita 2012), changes to household income and expenditure (Anttila-Hughes & Hsiang 2013), infant mortality (Anttila-Hughes & Hsiang 2013), Gross Domestic Product growth (Hsiang 2010, Hsiang & Jina 2015), and depreciation (Hsiang & Jina 2014). We also compare our measure of wind speed against total direct national economic damages (normalized by GDP) from a limited sample of TCs estimated by Nordhaus (2010). We find that national wind speed exposure is a meaningful predictor of total national damages (Figure 1d and Figure SI5), although we note that this outcome is highly uncertain and widely understood to be biased.

We thank the reviewer for pushing us to investigate these issues, as we believe these new analyses substantially strengthen the manuscript.

3. On a related note to #2, there are high wind events that are not associated with TCs. If the key channels for mortality are mostly through wind, then we might expect those non-TC wind events to have a similar impact; on the other hand, if the flood impacts are important, then they wouldn't. It would be helpful for the authors to discuss the choice to only consider TC-related winds and the implications for interpretation.

Our reply:

We focus on wind speed because it is the only metric that can be consistently reconstructed for all storms going back to 1930, and we consider it a proxy for other dimensions of storm intensity that are not measured directly in this analysis. The analysis above, responding to the reviewer's comment #2, suggests wind speed is a strong predictor of damages and it is also a valid proxy for rainfall. Thus, to address this concern, we have added text to the manuscript discussing these points:

Revised text from line 63-72:

“This reconstruction results in estimates of maximum wind speed experienced at each 0.1 X 0.1 pixel on the ground during each storm (Figure 1a and Figure SI1), a measure that has been shown to strongly predict physical damages and other economic and social impacts globally. We additionally verify that this measure predicts direct normalized damages in CONUS, estimated in a prior prominent analysis, (Figure 1d and Figure SI5) and it serves as a meaningful proxy for other TC-related weather, such as rainfall (Figure 1e and Figure SI6). Ground-level wind speed exposure is not the only dimension of TCs that impact human outcomes, however it is the only metric that can be consistently reconstructed for all storms going back throughout our entire sample, to the best of our knowledge. We thus consider wind speed as a meaningful, albeit imperfect, proxy measure for the physical aspects of storm intensity that impact all-cause mortality.”

4. In the case of TCs that were contemporaneous or occurring around the same time-period with overlapping signals, the authors use deconvolution to recover single-TC impulses and estimate the effects of a singular TC. Regarding the models that are non-linear models for TC incidence lines 879-895, would non-linearity of effects violate necessary assumptions made for deconvolution – i.e., does modeling mortality as a non-linear function of TC incidence lead to non-separability and non-additivity of individual TC effects?

Our reply:

The non-linear TC models we present are nonlinear in storm intensity, but additively separable across storms. Thus, the nonlinearity expressed in our results does not contradict the implicit assumptions required by linear deconvolution. (Total mortality in each year is the linear sum of storm-specific impacts, but storm-specific impacts are allowed to be a nonlinear function of each storm's intensity.)

Are the non-linear models presented as a robustness test but not used in any of the other reported results?

Our reply:

We present results from our linear model in Fig 2, since it is the simplest to interpret, allows easy direct comparison across strata in the sample, and it is a very good approximation of overall structure of the mortality response. However, because the non-linear model accounts for patterns of adaptation that we believe are important to account for when computing excess mortality burdens, we use the non-linear model to compute results in Figures 3-4. We present the range of total mortality estimates that result from all versions of the model in Table SI1. These details are all described in the Methods section of the manuscript.

5. To what extent do differences in the population distribution within states affect the results + associated interpretation? The share of the population on the coast of Florida is quite different than North Carolina, for example. The shift in the distribution over time may also contribute to the observed (non)-adaptation over time within states.

Our reply:

To address this concern, we have developed a new test to evaluate whether the spatial distribution of populations, relative to coast, affects the mortality response of states. We first compute the average fraction of each state’s population that lives in a coastal county, ranking states by this measure. We then break our sample into quartiles based on this ranking and evaluate whether the mortality-response to TCs differs across these quartiles (the first two quartiles are combined, since they have no coasts). The results are now presented in Figure SI13d, pasted below:

“Figure SI13: (d) average fraction of state population in coastal county: purple shade lines 1st through 4th quartiles, 1st and 2nd quartile combined. Solid black line = cumulative effect and 95% C.I. shaded.”

We find no systematic pattern in the mortality response across these strata that is distinguishable from statistical noise. The central estimate for the average response within the third quartile is higher than average, while the central estimate for quartiles 1, 2 and 4 are lower. This leads us to conclude that our results (both average effects and trends) are unlikely to be driven by systematic changes in the spatial distribution of populations within states. This finding is now described in the main text:

Added Text from line 189-193:

“We additionally evaluate whether the spatial distribution of populations relative to the coast, within states, alters the mortality response to TCs. Stratifying states based on the average fraction of the population that lives in coastal counties, we fail to find evidence that states with high concentrations of coastal populations are systematically different from states with little or no coastal population (Figure SI13d).”

We also now note that this finding is consistent with the small trend associated with the spatial rearrangement of the population:

Added Text from line 310-312:

“The smallness of this trend component is broadly consistent with our finding that the spatial arrangements of populations within states have had limited effect on the overall mortality impact of individual storms (Figure SI13d).”

Minor comments

1. The authors present pooled 95% confidence intervals for their final models. Given the observed heterogeneity in mortality responses across states due to varying exposure to TCs, it would be useful to clarify whether these 95% CI are calculated using cluster-robust standard errors at the state-level to account for this.

Our reply:

We thank the reviewer for identifying that we overlooked including this detail in the original manuscript. As the reviewer correctly points out, in many contexts errors in regression models are highly structured with non-Normal distributions, often because the underlying regression models are coarse approximations for the data generating process and do not describe patterns in the data sufficiently that distributions of

residuals are rendered stationary and i.i.d. For example, use of insufficiently flexible trends (e.g. linear trends when actual trends are nonlinear) frequently causes unmodeled autocorrelation to remain in the estimated residuals. One approach to addressing this issue is to estimate “clustered” standard errors that are inflated based on the degree of unmodeled heterogeneity within each cluster group. This approach aims to reduce bias in estimates of variance in regression coefficients, although it does not correct for bias in the regression coefficients themselves. Further, cluster-robust standard errors are known to be strongly biased when the count of clusters is low, as would be the case if we clustered standard errors at the state level in this context (N=36 states in our sample) (Cameron & Miller, *J. of Human Resources*, 2015). Thus, we opt for an alternative approach of adopting a rich set of non-parametric controls that accounts for the underlying structure of the data-generating process well, such that there is very little unmodeled variation with autocorrelated structure remaining in our errors. This approach has the dual benefit of accounting confounding variation in omitted variables, such that our coefficient estimates are unlikely to be biased, and generating residual variation that is normally distributed and i.i.d., such that the assumptions of OLS standard errors are satisfied. This also allows us to circumvent the challenge of having too few clusters.

In order to test the validity of this approach, as described in the original manuscript, we are able to use permutation tests to estimate confidence intervals and statistical significance. Computing significance via permutation test is an entirely different approach that does not require assumptions about error structure. These permutation tests indicate that our standard error estimates based on asymptotic assumptions are appropriately sized, further supporting our approach for estimating uncertainty. Thus, our estimates of uncertainty are corroborated through two entirely independent procedures.

Thus, to address the reviewer’s concern, we have added text to the manuscript to explain how and why we compute our estimates of uncertainty. To support these modeling choices, we have also added a new diagnostic Figure SI9 that allows readers to verify key assumptions. We also now explain that the permutation tests should be considered as an independent check on the validity of our estimated uncertainty. The text now states:

Revised text from line 898-909:

“We evaluate the distribution of the unmodeled variation represented by the error term ϵ_{it} and find that it essentially follows a Normal distribution except with slightly positive kurtosis (Figure SI9a). The distribution of these residuals appears stationary throughout the sample period and independent over time (Figure

SI9b). The consistency of the distribution of these errors is attributed to the high degree of flexibility in the non-parametric terms of our econometric specification, which are able to capture those components of the data-generating process that would otherwise appear as auto-correlated errors. Based on this evaluation, we construct OLS standard-error estimates as the underlying assumptions for these estimates appear to be reasonably satisfied. In addition, we find strong support for this modeling choice when we conduct a variety of permutation tests for statistical significance (Figure SI12), all of which indicate that our asymptotic estimates for confidence intervals are correctly (and possibly conservatively) sized and our tests for statistical significance correctly powered. Notably, these permutation tests do not rely on the assumptions used to estimate these confidence intervals, thus they can be considered independent corroboration for the validity of this approach.”

And the new figure displaying errors added to the text is:

Figure SI9: Distribution of the residuals from the main linear model. (a) Density plot of the residuals from the impulse response of mortality rates following state-level wind speed incidence. Gray line = normal distribution. (b) Percentiles of residuals across each month between 1950 - 2015.

2. There is one study of mortality post-Maria that is in the references, but the result seems like the most comparable point estimate (there may be other Maria papers that are also relevant here), so I suggest that the authors actually use the result to

contextualize their own findings. While clearly a different type of hazard, there is also some long-term mortality evidence from the Indonesian tsunami (<https://pubmed.ncbi.nlm.nih.gov/29731526/>, <https://pubmed.ncbi.nlm.nih.gov/33681474/>).

Our reply:

Thank you for this suggestion. We have added some additional text on line 338 as well as the two citations suggested:

Revised Text from line 330-342:

“We acknowledge that the large difference between the indirect excess mortality burden we compute and official counts of direct TC deaths is surprising. In fact, the authors initially believed these findings resulted from calculation errors, as the absence of any previous comparable analysis made it difficult to construct an informed prior for these estimates. However, we note that these findings are consistent with the growing literature indicating that climatic conditions generally, and TCs specifically (Ho 2017; Frankenberg 2020), have larger and more enduring impacts on human well-being than previously recognized. For example, high temperatures have been implicated as causing roughly 150,910 deaths annually in CONUS (7.0% of total mortality), roughly double the burden of TCs. Furthermore, TCs have been shown to trigger substantial long-term economic losses at both the individual household and one recent study (Kishore et al. 2022) estimated that there were 3,000 additional excess deaths beyond the official government count in Puerto Rico three months after Hurricane Maria. In addition, our finding that post-TC mortality risk is uneven across Black and white populations is consistent with prior descriptive results that minority populations may experience greater vulnerability to disasters and receive less government support in their wake.”

3. The research design does not allow for estimation of different effect sizes due to temporal compounding (i.e., the possibility that a second TC shortly after a first TC has a worse impact), as far as I can tell. This should be noted in the limitations.

Our reply:

We thank the reviewer for this comment. We added additional text on line 376 to clarify this point.

Revised Text from line 376-383:

“Our analysis faces several limitations due to our data that should be improved upon in future work... Third, this analysis does not explicitly measure the

interacting effect of multiple climatic events that arrive in rapid succession, such as multiple TC events striking the same population in a row or a TC right after a non-TC event. Such cases are uncommon but likely impactful, and should be studied in future analysis. Nonetheless, our results report average effects that include the impact of these compound events and we believe that the overall impact of the TC climate is well represented in our estimates.”

4. The Abrupt changes after 2001 paragraph is difficult to read and contains several typos.

Our reply:

Thank you for identifying this confusing text. We have revised the paragraph substantially to improve its clarity and fix typos. The text now reads:

Revised text from line 276-288:

Accelerating trend in excess mortality after 2001 Our reconstruction of TC-related mortality exhibits a positive average trend of +9.2 TC-related deaths per month, with a notable increase to +43.3 deaths per month after roughly 2001 (Figure 4b). This acceleration is due to the higher frequency of TC landfalls post-2001 (17/yr) compared to pre-2001 (14/yr; Figure 4h), but not due to these arriving storms being more intense on average. The maximum intensity of state-level incidence within landfalling storms actually declined slightly from 23.6 m/s pre-2001 to 21.4 m/s post-2001 (Figure 4i). The net effect of these two factors (more frequent storms with slightly weakening average intensity) increased the intensity of average TC incidence experienced by CONUS populations from 0.125 m/s to 0.143 m/s post-2001 (Figure 4j). The changes in the distribution of TC events affecting CONUS after 2001 were themselves likely caused by a combination of factors including warmer sea surface temperatures (Vecchi et al 2007, Xie et al 2010, Webster et al 2005, Emanuel 2005) and reductions of anthropogenic aerosol emissions (Sobel & Camargo 2016, Wang et al 2014) (which create an environment more amenable to TC intensification); and shifts in steering winds (Moon & Camargo 2020) (which direct a larger fraction of TCs to landfall in CONUS after formation). We note that identifying factors driving the TC climate remains an active area of research (Bhatia 2022, IPCC).

5. Line 819: The reference to Fig. 1c may be erroneously labeled; the authors may have intended to reference Fig. 1e.

Our reply:

Thank you for catching this error, it has been corrected.

6. The legend in Fig. S9c plots “quantiles”. The text refers to “quartiles”, which is likely what the authors intended on communicating.

Our reply:

Thank you for catching this error, it has been corrected.

7. Lines 874-875: Minor sp. inconsistency for “fourth/forth”.

Our reply:

Thank you for catching this error, it has been corrected.

We thank the reviewer for their thoughtful comments that pushed us to analyze additional questions. We sincerely believe that this additional analysis has substantially improved the strength of the manuscript.

Reviewer comments in black

Our replies in blue

New text in the manuscript in red

Reviewer #2:

Summary of the key results:

Post-tropical cyclone (TC) mortality over a very long period (1930-2015) was evaluated for the entire continental US (CONUS) region. The paper estimates total mortality – not just mortality directly attributed to TCs and occurring in the immediate aftermath of a storm, but all excess mortality attributed to 501 TCs that reached CONUS over this period. They report an average of 7,000-11,000 total excess deaths per storm, occurring over fifteen years. This is far more than the 24-death estimate by the US Government based on direct attribution in the immediate aftermath of a TC. They conclude that this heretofore uncalculated mortality burden comprises a substantial fraction of mortality rates along the Atlantic coast, around 3.2-5.1% of all deaths. They identified populations at the highest risk: infants, people ages 1-44, and Black people.

Originality and significance:

The work is both original and highly significant.

Our reply:

We thank the reviewer for their positive evaluation of the manuscript.

Data & methodology:

The authors have employed a valid, entirely original approach and, in my view, very effectively allowed them to test their findings in the complete absence of any control group. The presentation of results is beautiful; excellent charts and graphics make it easier to understand their methodologies and present their findings and statistical uncertainties.

Our reply:

We sincerely appreciate the reviewer's evaluation and generous words.

Storm Data: I am concerned about using the Limited Information Cyclone Reconstruction and Integration for Climate and Economics (LICRICE) model as the only surrogate for TC-related exposures to human populations. As you note, many TCs with lower windspeed have had substantial population impacts because of flooding and power outages. This metric may be underestimating the impacts of TCs on mortality. Averaging maximum windspeed over the entire land area of each state results in a picture (original Fig. SI2) that appears to show Connecticut with much higher TC severities than Texas (and the label for the figure does not say that the metric was averaged over land area).

Our reply:

To address this concern, we have verified that the wind speed metric is explicitly described as being a spatial average in the Methods description, and we have now clarified the caption in original Figure SI2 to read:

Figure SI2: State monthly maximum wind speed from tropical cyclones. LICRICE modeled monthly maximum wind speed from tropical cyclones between 1950 and 2015, by state. State tropical cyclone wind speeds are averages across grid cells.

In addition, in response to specific requests by Reviewer #1, we have now linked the LICRICE data to national direct damages data and a limited number of rainfall fields provided by NOAA. This has resulted in substantial new analysis verifying that the LICRICE windfield, spatially aggregated, predicts normalized damages and is a meaningful proxy of other non-wind TC weather. This new analysis has resulted in new figure panels in the main text (Figure 1d-e) and new Supplementary Figures SI5 and SI6. These new findings are discussed in the data portion of the methods and are now summarized in the main text:

Revised text from line 63-71:

“This reconstruction results in estimates of maximum wind speed experienced at each 0.1 X 0.1 pixel on the ground during each storm (Figure 1a and Figure SI1), a measure that has been shown to strongly predict physical damages and other economic and social impacts globally. We additionally verify that this measure predicts direct normalized damages in CONUS, estimated in a prior prominent analysis, (Figure 1d and Figure SI5) and it serves as a meaningful proxy for correlation with other TC-related weather, such as rainfall (Figure 1e and Figure SI6). Ground-level wind speed exposure is not the only dimension of TCs that impact human outcomes, however it is the only metric that can be consistently

reconstructed for all storms going back throughout our entire sample, to the best of our knowledge. We thus consider wind speed as a meaningful, albeit imperfect, proxy measure for the physical aspects of storm intensity that impact all-cause mortality.”

Mortality Data: The most impressive increase in mortality was for “other” causes. The CDC’s underlying cause of death data does not include the “other” category, which is, in this paper, a grab bag containing diabetes and all the other diseases that were not assessed separately.

[See the following from CDC: how ICD codes have (and have not) been consistent over the years: US Centers for Disease Control and Prevention. (2015). Comparability across revisions for selected causes. Retrieved from <https://www.cdc.gov/nchs/data/dvs/comp2.pdf>.]

It must be that the authors themselves created this category, which may have come about because they used data from the authors they cite for their mortality database.

[A. Barreca, K. Clay, O. Deschenes, M. Greenstone, and J. S. Shapiro, “Adapting to climate change: The remarkable decline in the US temperature-mortality relationship over the twentieth century,” *Journal of Political Economy*, vol. 124, no. 1, pp. 105–159, 2016.]

The Barreca mortality analysis focused on the impacts of air conditioning on heat-related mortality for five underlying causes of death: cardiovascular, respiratory, motor vehicle injuries, infectious, and neoplasms, the same specific underlying causes as in this manuscript. Also, as a correction, Barreca et al. only digitized some of the data from prior to 1959. (Bacerra, in an earlier paper, describes digitizing the earlier data.) Data from 1959 onward are downloadable from the NBER and other sources: <https://www.nber.org/research/data/mortality-data-vital-statistics-nchs-multiple-cause-death-data>.

Given that the “other” causes predominate, it would be preferable to disaggregate these deaths and examine, at least, the significant causes of death that comprise the “other” category. The lack of identification of individual underlying causes makes it difficult to determine what, if anything, might have been done to prevent these deaths. Some significant causes which may be important in the aftermath of TCs include diabetes; intestinal infections (infants); chronic liver disease and cirrhosis; chronic kidney disease, homicide, drug poisoning; suicide; unintentional injuries (other than a motor vehicle); congenital malformations (infants); and conditions originating in the perinatal period

(infants). Possibly the authors could undertake a sub-analysis for just the years from 1959 onward if they cannot obtain from Bacerra the complete data set that includes data from the period between 1930 and 1958 (or cannot, themselves, digitize the earlier data, which are available from the NCHS). As another possibility, the authors could remove the cause-specific mortality analysis and conduct a more thorough review in a second publication.

Our reply:

We thank the reviewer for this thoughtful comment. We construct the “other” category based on the difference between total all-cause mortality and the cause-specific mortality as harmonized and observed in the Bacerra et al. and CDC data. We agree that there are significant causes that are within the “other” category that, if disaggregated, would help inform interventions to help reduce mortality after a TC. However, after careful examination of alternative non-digitized data sources, we have determined that it is beyond the scope of this analysis to assemble a new data set to evaluate sub-components of this category. Nonetheless, to address this concern we have added additional text to the paper to help highlight these important points about the other category. Our description of the mortality data now reads:

Revised text from line 724-732:

“For the years 2005 - 2015, we analyze mortality data from the public Center for Disease Control and Prevention (CDC) Underlying Cause of Death database (2017). The data is based on death certificates for U.S. residents, which gives a single underlying cause of death and demographic data. Cause of death prior to 2000 was indexed using the 4-digit ICD-9 code and 2000 onwards the index changed to the 4-digit ICD-10 code. We harmonized the cause of death into 5 categories that matched the Barreca et al. cause of death variables. We also construct a 6th category which is the difference between all-cause mortality and the sum of the 5 cause-specific categories, called “other”. Notably, the change in CDC ICD code methodology resulted in a shift in the counts of deaths from specific causes, particularly infectious diseases and cardiovascular disease.”

We agree with the reviewer that the large effects in the “other” category deserve further examination in a future analysis. However, we have chosen to retain the limited cause-specific analysis that we are able to conduct as we believe it nonetheless provides important information about the drivers for roughly half of the mortality increases after a TC. Further, it provides important information about some drivers that *do not* contribute to post-TC mortality, such as motor vehicle accidents. However, to address the reviewer’s concern, we have now added text explaining that future work should investigate the the components of the “other” category:

Revised text from line 176-177:

“Given the large role of the “other” category to the overall TC-mortality response, we believe that future work should investigate the role of specific causes that contribute to this category.”

Appropriate use of statistics and treatment of uncertainties:

As was implied above, I am very favorably impressed by how the authors used statistics and identified statistical uncertainties.

Our reply:

We thank the reviewer for the affirmation of our methodology.

Conclusions: robustness, validity, reliability:

Mortality is the totality of health impacts from TCs; one or more adverse health events precede every death. It would be expected that not all these events would cause premature mortality. This indicator should be discussed within the context of broader public health impacts, not all of which resulted in death. Assessing mortality alone significantly underestimates the overall public health burden of TCs in CONUS.

Our reply:

To address this concern, we now describe in the Discussion that the absence of non-fatal outcomes is a shortcoming of our analysis and area for future investigation:

Revised text from line 381-383:

“Our analysis faces several limitations due to our data that should be improved upon in future work.... Fourth, this analysis does not capture the effect of TCs on non-fatal outcomes (Parks et al. 2021). If morbidity resulting from TCs is similar in structure and scale to the mortality response, then our estimates may substantially understate the public health burden of TCs.”

The authors noted migration as an issue that could not be addressed in this long-range and state-by-state analysis. This reviewer was privileged to do a study of mortality after Hurricane Maria in Puerto Rico in which we had access to flight logs that allowed for understanding population flux over several years. As demographers know, people do

not migrate randomly. Very likely in the context of this study is that the inability to account for migration would underestimate risk. First, people whose lives and livelihoods are most impacted may be more likely to leave coastal areas for opportunities elsewhere. These moves may be within the same state, but if not, their subsequent mortality will be lost to analysis. Second, people moving to the coasts from other parts of the country will be counted in the population data upon which mortality risks are computed, yet, if they are moving in from non-TC impacted states, they lack exposure to TCs. Both of these situations would result in underestimating mortality risk in TC-impacted states.

Santos-Burgoa, C., Sandberg, J., Suarez, E., Goldman-Hawes, A., Zeger, S., Garcia-Meza, A., ... Goldman, L. R. (2018). Differential and persistent risk of excess mortality from Hurricane Maria in Puerto Rico: a time-series analysis. *Lancet Planet Health*, 2, e478-e488. doi:10.1016/S2542-5196(18)30209-2.

Our reply:

We agree with the reviewer that an important limitation of this study is our inability to fully account for migration. This limitation was explicitly stated in the original manuscript (line 382)

“Lastly, we are unable to track migration and mortality simultaneously using these data, thus if individuals migrate outside of their state within the 20 years after a storm, our estimates will not account for their experience after they move; although our estimates do adjust for population changes within the origin state.”

Further, in our analysis of trends (Figure 4k) we evaluated whether changes in the spatial correlation between persistent cyclone risk and population growth could explain the trend in mortality burden that we compute. We find that roughly 14.3% of the trend can be accounted for by the redistribution of populations toward coastal states.

We believe that both of these elements of the original manuscript largely address much of the reviewers' concern. However, to ensure we address the reviewer's concern, we have developed a new test to evaluate whether the spatial distribution of populations, relative to coast *within* states, affects the mortality response of those states. We first compute the average fraction of each State's population that lives in a coastal county, ranking states by this measure. We then break our sample into quartiles based on this ranking and evaluate whether the mortality-response to TCs differs across these quartiles (the first two quartiles are combined, since they have no coasts). The results are now presented in Figure SI13d (formally Figure SI9), pasted below:

“Figure SI13: (d) average fraction of state population in coastal county: purple shade lines 1st through 4th quartiles, 1st and 2nd quartile combined. Solid black line = cumulative effect, and 95% C.I. shaded.”

This new finding is now reported in the main text:

Added text from line 189-193:

“We additionally evaluate whether the spatial distribution of populations relative to the coast, within states, alters the mortality response to TCs. Stratifying states based on the average fraction of the population that lives in coastal counties, we fail to find evidence that states with high concentrations of coastal populations are systematically different than states with little or no coastal population (Figure SI13d).”

Overall, we believe this collection of results points towards migration as having a meaningful but modest influence on the structure of TC-related mortality within CONUS.

The discussion of black/white racial inequities could benefit from the citation of literature documenting how historical racial inequality (e.g., redlining in housing markets) has placed black coastal communities in harm’s way (i.e., in more flood-prone areas, in housing that is less protective and in communities where there has been a failure to invest in levees and other community level protective measures). The authors should be aware that HHS, EPA, FEMA and the states are increasingly attempting to identify communities at highest risks using social vulnerability indices. I am not suggesting that such data are available back to 1930, but these issues are as, or more, significant than those included in the discussion. To cite but a few examples of literature that might be cited (but there is much more):

US EPA. (2021). Climate Change and Social Vulnerability in the United States: A Focus on Six Impacts. (EPA 430-R-21-003). Washington, DC Retrieved from

www.epa.gov/cira/social-vulnerability-report

Hardy, R. D., Milligan, R. A., & Heynen, N. (2017). Racial coastal formation: The environmental injustice of colorblind adaptation planning for sea-level rise. *Geoforum*, 87, 62-72. doi:<https://doi.org/10.1016/j.geoforum.2017.10.005>

Tee Lewis, P. G., Chiu, W. A., Nasser, E., Proville, J., Barone, A., Danforth, C., ... Craft, E. (2023). Characterizing vulnerabilities to climate change across the United States. *Environment International*, 172, 107772.

doi:<https://doi.org/10.1016/j.envint.2023.107772>

Morello-Frosch, R., & Obasogie, O. K. (2023). The Climate Gap and the Color Line — Racial Health Inequities and Climate Change. *New England Journal of Medicine*, 388(10), 943-949. doi:10.1056/NEJMsb2213250

Our reply:

We thank the reviewer for this comment and highlighting this important and growing area of research and government action. We have added three of the four additional citations listed above.

Line 248-250:

“Black populations bear a relatively larger TC-mortality burden than white populations, due to the combination of their greater vulnerability (Figure 2d) and their spatial distribution, which is denser in the southeastern CONUS where TC incidence is more common (EPA 2021).”

Line 154-155:

“There is growing concern that minority populations and vulnerable groups may suffer relatively greater harm from both natural and anthropogenic environmental conditions (Cutter et al. 2003, Tee Lewis et al 2023, Morello-Frosch & Obasogie 2023).”

Implications: In the discussion, the authors are focused on providing health care to address these regional inequities. They should also consider including discussions of how prevention activities could, in the future, contribute to preventing the excess mortality that they are documenting. They did not present evidence that there are too few such facilities in the region. To the extent this is true, one might recognize the lack of demand for services caused by people who lack adequate healthcare coverage, and historically and even currently, as many coastal states have refused to expand Medicaid. However, there is a diversity of policies and doctor: patient ratios across the

CONUS TC impacted states, and more likely other factors should be considered. As an alternative, the authors may consider the role of prevention, particularly in the context of stress, as a causal factor in several diseases that lead to premature mortality over many years, namely hypertension (cardiovascular disease, chronic renal disease, poor pregnancy outcomes); diabetes (via stress-induced mechanisms as well as the impacts of storms on food systems and perhaps eating behaviors); disruption of HPA axis relationships and the profound impact of that across populations on birth outcomes and infant mortality even when the stress occurs preconceptionally. (Hopefully, the implication of this is not lost on the authors when discussing the high rates of infant mortality, years after TCs, that they have documented.) The implications of this are not to build more hospitals but to address factors like hypertension and blood sugar control, psychosocial stress, and of course, anything in the system that can be done to reduce stress and uncertainty in the environment around people. Only treating the heart attack, or a child born preterm, is not an adequate response.

Class, Q. A., Khashan, A. S., Lichtenstein, P., Långström, N., & D'Onofrio, B. M. (2013). Maternal stress and infant mortality: the importance of the preconception period. *Psychol Sci*, 24(7), 1309-1316. doi:10.1177/0956797612468010

Ghosh, A. K., Demetres, M. R., Geisler, B. P., Ssebunya, S. N., Yang, T., Shapiro, M. F., . . . Abramson, D. (2022). Impact of Hurricanes and Associated Extreme Weather Events on Cardiovascular Health: A Scoping Review. *Environ Health Perspect*, 130(11), 116003. doi:10.1289/ehp11252

Cohen, S., Janicki-Deverts, D., & Miller, G. E. (2007). Psychological Stress and Disease. *Jama*, 298(14), 1685-1687. doi:10.1001/jama.298.14.1685

Duthie, L., & Reynolds, R. M. (2013). Changes in the Maternal Hypothalamic-Pituitary-Adrenal Axis in Pregnancy and Postpartum: Influences on Maternal and Fetal Outcomes. *Neuroendocrinology*, 98(2), 106-115. doi:10.1159/000354702

Our reply:

To address the reviewer's concern, we have also added additional discussion of this point as suggested.

Added Text from line 363-365:

“(5) Heightened physical and mental stress from TC incidence, or the related economic impacts, may have long-run effects on individual's health, increasing

future mortality risks (Cohen et al. 2007, Class et al. 2013, Duthie & Reynolds 2013, Ghosh et al. 2022)."

And have added the citations suggested by the reviewer.

Suggested improvements: experiments, data for possible revision

Ideally, as noted above, there would be the possibility of disaggregating the "Other" mortality category used to analyze underlying causes of death. I think that other suggestions are more relevant to the conclusions.

Several references are suggested.

Our reply:

See response to these comments above.

Clarity and context: lucidity of abstract/summary, appropriateness of abstract, introduction, and conclusions

In conclusion, this paper is ground-breaking and incredibly important to understanding the public health significance of tropical cyclones. It is very well written. Though it may surprise many that TCs are such a significant risk factor for mortality, as noted above, if anything, this paper provides a modest estimate of public health risk. From the outset, collaboration with an epidemiologist would have benefitted the work. Although I suggested the possibility of a second paper to improve the epidemiology analysis, I very much like the scope and breadth of this paper and hope it is feasible for the authors to address these concerns! Congratulations to the authors for a great contribution!

Our reply:

We sincerely thank the reviewer for the detailed and thoughtful comments and the affirmation of the importance of our findings.

Reviewer comments in black

Our replies in blue

New text in the manuscript in red

Reviewer #3:

This paper estimates the number of excess deaths in the 20 years following a tropical cyclone exposure. The premise of this paper is significant. However, this paper has several limitations that need further analysis. Below please find my specific comments.

Our reply:

We thank the reviewer for these comments and questions. Our responses and clarifications in the manuscript are below and we hope these changes will reduce any future confusion.

1. A few times in the manuscript the authors state the assumption that TC exposure is random. This is clearly not true, though, as TCs have very distinct spatial and temporal patterns. For example, June in Texas is not exchangeable with August in North Carolina. More sensitivity analyses to evaluate the robustness of the results to this assumption are definitely warranted.

Our reply:

We thank the reviewer for this comment. This is an important point that we tested for in the original manuscript. Though there are TC seasons in particular areas, TC exposure appears to have a distribution that is as good as random conditional on the rich set of fixed effects and trends that we account for. For example, recall that we account for state-by-month-of-year fixed effects, such that we are effectively identifying the effect of storms by only comparing specific months at specific locations (e.g. June in Texas) to that same month in that location in other years. Thus we agree with the reviewer that August in North Carolina is not exchangeable with June in Texas, which is why mortality in those state-by-months-of-year they are treated as coming from different distributions.

We explicitly test whether numerous patterns in the non-random distribution of storms could bias our results by performing four different randomization placebo tests. These randomization tests are designed to explicitly test whether specific patterns in the distribution of TC exposure could be biasing our estimates. As our original Methods section explained (lines 913-923):

“A strong test designed to avoid such artifacts is to ensure that model estimates of TC impacts on mortality are unbiased in a variety of situations where the structure of the association has been manipulated. In four tests, we shuffle the true TC data in different ways. In each case, this shuffling should break any correlation between TC incidence and mortality such that an unbiased estimate of the effect of shuffled TCs on mortality is zero. However, in each case, some of the structure in the original TC data is allowed to remain in the shuffled TC data. For example, randomization within a state over time retains the average cross-sectional patterns of TC incidence, but destroys any time- series structure. Thus, these tests allow us to examine whether, in each case, the remaining structure generates artifacts in the model that would produce a spurious result. Any non-zero correlation, on average, would indicate a biased model where the bias is driven by the non-randomized components of the original TC data.”

These randomizations included (line 930-947)

“Within State Randomization” - shuffles the sequencing of TCs that a state experiences over time. TCs are always assigned to the correct state, but the month and year assigned to each storm is random. The cross-sectional average pattern of storm incidence is preserved in the data. Thus, this tests whether time-invariant cross-sectional patterns across states generate spurious correlations...

“Within Month Randomization” - shuffles the TC incidence across states within each month-of-sample. TCs are always assigned to the correct month and year, but the state assigned to each storm is random. The average time-series structure of TC incidence nationwide is preserved. Thus, this tests whether national or seasonal trends, which are nonlinear, could bias estimates produced by this model...

“Across State” - shuffles complete TC times-series across states, keeping the timing and sequence of storms correct as blocks. TCs are always assigned to the correct month and year, and the sequence of storms experienced by a state is always a continuous sequence that is observed in the data. However, the state that is assigned that sequence is randomly chosen. This tests whether trends within a state and within the sequence of storms that a state experiences could generate bias. This test differs from the within month randomization because state-level trends often differ across states (see Figure SI2 and SI6) and there are complex seasonal patterns that could potentially affect estimates.

The full results from all these randomization tests are shown in Figure SI12 (formally Figure SI8).

To address the reviewer's concern and avoid any future confusion among readers, we now explicitly connect these tests to the concern raised by the reviewer. The text now clarifies.

Added text from line 954-958:

“These [randomization] results demonstrate that non-exchangeability across states within a month, across months within a state, or across states (conditional on month of sample) does not confound our analysis; indicating that the rich set of fixed-effects and trends successfully adjust for many patterns of TC incidence and/or mortality such that the remaining conditional variation is as good as random.”

2. I am concerned about the statewide analysis. In addition to differences in TC exposure across states, there are distinct patterns within each state as well (e.g., western NC does not experience the same TC exposure as eastern NC, and the populations living in these two areas are quite different as well). I understand that the authors do not have access to finer-resolution mortality data that go back to the beginning of the study period. However, such data (at count and month level) are available since the mid-80's (if not earlier). Therefore, a sensitivity analysis at the county level for those years for which data are available, and comparison with the main statewide analysis, is warranted.

Our reply:

Our results present the average treatment effect across populations, aggregated at the state level. By construction, this must match average effects at finer or coarser levels of aggregation in comparably structured OLS models using data aggregated at these other levels. As shown in the original Figure SI9, this estimated average effect is robust to using population or state-weighting, indicating that county-level results using population weights must also match, due to the linearity of our model and the properties of averages.

Nonetheless, the concern raised by the reviewer could affect our results if eastern coastal counties responded to TC incidence systematically differently than western non-coastal counties. Thus, while an analysis of county-level data is not feasible with publicly available digital records (due the number of periods of overlapping lags required for identification) and digitizing existing paper records is beyond the scope of this analysis, we have devised a test using our available data to address this concern.

To address this possibility, implied by the reviewer’s suggestion, we have developed a new test to evaluate whether the spatial distribution of populations, relative to coast *within* states, affects the mortality response of those states. We first compute the average fraction of each state’s population that lives in a coastal county, ranking states by this measure. We then break our sample into quartiles based on this ranking and evaluate whether the mortality-response to TCs differs across these quartiles (the first two quartiles are combined, since they have no coasts). The results are now presented in Figure S113d (formally Figure S19), pasted below:

We find no systematic nor statistically significant pattern in the state-level response based on the distribution of geographies among counties within each state. This indicates that the average treatment effect is indeed a reasonable approximation for the average effect across county-level locations, regardless of distance from the coast.

3. Fig. 1b: Why do the x-axes only go back to 1950?

Our reply:

We set the x-axes to be consistent across the panels, so they can be matched visually between the two samples. We found that displaying all the TC data compressed the mortality data, making it difficult to read. However, to address the reviewer’s concern, we have now extended the x-axis in Figure S12 to the start of the sample in 1930.

4. Fig. 1, “cumulative monthly TC maximum wind speed”: What does this mean and how was it estimated? And what is its relevance to the rest of the paper? That is, why not show average speeds?

Our reply:

The “cumulative monthly TC maximum wind speed” is the sum of maximum monthly wind speed experienced at each pixel, summed over years in each decade. Plotting the average annual/monthly incidence would produce identical images with the color axis rescaled by the one divided by the number of years/months. We present cumulative values because our findings indicate that the impact of TCs are cumulative, such that the cumulative exposure maps directly to the overall mortality burden, i.e. is the most directly relevant climatological summary statistic.

To address the reviewer’s concern and avoid future confusion among readers, we have now added clarifying text to the caption in Figure 1 and SI1.

Caption Figure 1:

(a) Example *decadal total* monthly TC maximum wind speed ($m s^{-1}$) computed using LICRICE (see Figure SI1 for all decade). Red = 500 ($m s^{-1}$) (maximum) and dark blue = 50 ($m s^{-1}$) (minimum).

Caption Figure SI1:

Tropical cyclone incidence data. Limited Information Cyclone Reconstruction and Integration for Climate and Economics (LICRICE) modeled *decadal total monthly maximum wind speed* from tropical cyclones at each $0.1^\circ \times 0.1^\circ$ pixel, between 1930 and 2018. Red = 500 ($m s^{-1}$) (maximum) and dark blue = 50 ($m s^{-1}$) (minimum).

5. Fig. 1e: What is plotted on the x-axis exactly? What are the “cumulative effects”?

Our reply:

Thank you for pointing out that the x-axis was missing a label. The x-axis is the cumulative all-cause mortality effect (per 100,000 per m/s) of TCs. It is the same as the y-axis in Figures 2b-f. We have added an x-axis title and clarified the caption of Figure 1.

6. Lines 115-116: How common is it for a state to experience monthly average winds of 6.9 m/sec when not exposed to TC? How does this impact interpretation of results?

Our reply:

In the absence of other non-TC extreme events, such high average wind speeds do not occur.

To address this and similar questions, which may arise because readers are unfamiliar with the distribution of monthly state-level wind speeds that we analyze, we have added to the manuscript a new Figure SI7 (pasted below) displaying summary distributions (PDFs and CDFs) of this metric. Grey indicates all state-by-month observations in our sample, and blue indicates observations with no-zero values. Overall, monthly average winds of 6.9 m/s or greater occur in 3.1% of state-by-month observations.

Figure SI7: Density and cumulative density of state monthly maximum wind speed from tropical cyclones. (a) density plots of the LICRICE modeled monthly maximum wind speed from tropical cyclones between 1950 and 2015 for 36 states in sample. Blue plot excludes observations with wind speed incidence equal to zero. (b) Cumulative density plots of the monthly maximum wind speed from tropical cyclones. Blue plot excludes observations with wind speed incidence equal to zero.

7. Lines 166-168: But aren't these results surprising, especially given other recent publications (e.g., Parks et al., JAMA, 2022)?

Our reply:

Parks et al. differs dramatically in sample, methodology and scope. Crucially, Parks et al. only search for effects from storms that emerge within the first 6 months, whereas we find that it takes 172 months for the large impact that we report as a main finding to emerge. This means that only 4.6% of the TC-mortality signal we observe appears in

the time frame that Parks et al. study. Furthermore, Parks et al. explicitly exclude the “other” causes of death that we find is the largest contributor to post-TC mortality, and which the reviewer points out is a surprising finding. Parks et al state (p. 947):

“Underlying causes of death were classified into 7 categories: cancers, cardiovascular diseases, infectious and parasitic diseases, injuries, neuropsychiatric conditions, respiratory diseases, and an aggregate set of other death causes. The results of all of these categories were reported except other causes of death. The “other causes” category was not included because the diversity of causes it captures led to substantial heterogeneity, and the composition varied greatly by age, sex, county, and time.”

This indicates that the conclusions of Parks et al. cannot speak to the findings we present, or the fraction of TC-mortality that occurs in the “other” cause of death category, since that study was structured in such a way that it cannot observe the phenomenon we report.

8. Lines 174-175: But are these estimates statistically different from each other? I am not sure that they are given the 95% CIs. If not, this finding is overinterpreted, please tone down.

Our reply:

The trajectory of these responses are highly statistically different from one another. Performing a t-test on the joint significance of the cumulative effects for months 1 - 240, we do find that they are statistically significantly different ($p < 0.01$). To address the reviewer’s concern, we have now added some additional text to the Figure 2 caption to clarify this finding in the paper.

Revised caption Figure 2:

(f) average TC incidence: low average exposure are the quartile of states with lowest non-zero average TC incidence, high average exposure are the upper three quartiles. Joint significance between high and low exposure for months 1-240 is $p=0.0082$ (see Figure SI13c for quartiles separately).

9. Nonlinear TC effects: It is not clear to me which model (linear vs nonlinear) actually yielded a better fit. How did the authors compare the two models? Whichever model yielded the best fit is the one that should be used for the calculation of excess deaths.

Our Reply:

We find that the model fit across the different specifications are very similar, ranging from an adjusted r-squared between 0.9262-0.9276. We present these values in the table below:

Model	Adjusted R-squared
Main linear model	0.9262
Linear model with adaptation	0.9261
Quadratic model	0.9270
Cubic model	0.9276
Cubic model with adaptation	0.9272

The Cubic model has the best fit, but the difference between the Cubic model and the Cubic model with adaptation is a tiny margin that is not itself statistically significant. In computing excess deaths in the main text (Figures 3-4), we focus on results that use the Cubic model with adaptation because it accounts for patterns of adaptation that we believe are important, especially because it affects the spatial pattern of the mortality burden. However, we present the full range of total mortality estimates that result from all versions of the model in our main conclusions, including those stated in the abstract, and all estimates are reported in Table S11. Thus, to address the reviewer's concern, we have now added a row to Table S1 that reports these R-squared values so that all estimates of the total mortality burden we report can be associated with a value of model fit.

10. What is the predictive accuracy of LICRICE?

We thank the reviewer for this important question. We now have conducted extensive new analysis to address this question.

It is nearly impossible to validate the wind speed data against ground-based wind speed measures, since most wind data collection systems are not designed to withstand/measure TC strength winds. However, we are able to evaluate the extent to which our measure of wind speed correlates with rainfall during storms and overall reported damages from a significant prior study (Nordhaus, 2010). [Although, we note that a similar evaluation of storm surge is beyond the scope of this analysis but is the subject of follow-up work we are conducting along with a large team of collaborators.]

We examine whether wind speed is associated with rainfall by reconstructing the rainfall field for a limited number of TCs where storm-specific gauge data is available. We emulate the approach of NOAA in constructing rain fields by interpolating rainfall, and then we compare these fields against the wind fields of the same storms. Our new Figure S16, pasted below, illustrates this comparison for six example storms and reports state-level correlations between our measure of wind speed exposure and an analogously computed rainfall exposure. The r-squared of these correlations range from 0.36 to 0.74, indicating that at the level of our analysis, wind speed is a fairly good proxy for rainfall within a storm, as hypothesized in our original manuscript.

Figure S16. Tropical cyclone rainfall and wind speed incidence comparison. (a)

Left panel, NCEP tropical cyclone rainfall weather station data (inches) inverse squared distance weighted interpolation at each $1^\circ \times 1^\circ$ pixel. Right panels, LICRICE modeled cumulative maximum wind speed (ms^{-1}) from tropical cyclones at each $0.1^\circ \times 0.1^\circ$ pixel. R-squared values from the linear correlation between state area average rainfall and state maximum wind speed for each storm.

We cannot conduct this analysis for all storms, but we now pool the data from all storms for which rainfall data is available and estimate an average association with r-squared = 0.31, indicating that state-level wind speed is a reasonable, albeit imperfect, proxy for state-level rainfall. This result is now presented in a new figure panel of the main text (Figure 1e):

Figure 1e: Association between state-by-storm average rainfall (inches) from NOAA station data and maximum wind speed (ms^{-1}) computed using LICRICE (263 storms, $N=12,889$, slope=0.116, R-squared=0.307).

This finding is now reflected in the main text, which has been revised to state:

Revised text from line 675-679:

LICRICE does not explicitly model storm surge, rainfall or flooding, however these dimensions of impact are captured in the analysis to the extent that they are correlated with integrated maximum wind speeds. For example, we find that LICRICE wind speed and NCEP rainfall from TCs are correlated within storms at

the pixel level (Figure SI6) and at the state-by-storm level (r -squared = 0.31, Figure 1e) for a limited sample of storms where granular rainfall data is available.

We have also evaluated the extent to which our measure of wind speed is correlated with estimates for total direct TC damage, i.e. monetary losses that occur during the storm event. Total storm damage is notoriously difficult to estimate, since many damages are to public infrastructure, are not insured, or are insured by various disparate entities. Thus, there is no definitive record of total damage and it is widely known that all efforts to compute such damages are coarse approximations. Nonetheless, we obtain total damage estimates from a prominent study by Nordhaus (2010) in which the author undertook extensive efforts to correct for many known sources of bias. We then evaluate the extent to which LICRICE wind metrics predict the log of total damages as a fraction of GDP – a standard, but imperfect, normalization used throughout the literature. Because Nordhaus's estimates of damage are national, we aggregate wind nationally and evaluate associations at the storm level, as well as aggregated across all storms reported by Nordhaus in each year (which is not a complete record).

These associations are shown in new Figure SI5, pasted below. Within a broad class of simple models, we find that the fit is maximized by a log-linear model that is a piecewise linear spline with a single kink. Above the kink, average damages appear to be a constant fraction of GDP, regardless of estimated higher wind speed exposure for a few outlying storms. The model fit is r -square=0.39 across storms and r -square 0.32 across years.

Figure SI5. Tropical cyclone damage and wind speed incidence comparison. (a) Total logarithm tropical cyclone national economic damage per national gross domestic product (GDP), assembled for Nordhaus 2010, of tropical cyclones between 1950-2005 and national average maximum wind speed (ms^{-1}) estimated from Limited Information Cyclone Reconstruction and Integration for Climate and Economics (LICRICE). Discrete spline estimated relationship, split at 1.2 maximum wind speed (ms^{-1}). (b) corresponding annual tropical cyclone log damage per GDP and national average maximum wind speed (ms^{-1}).

We also now present the linearly increasing segment of the storm-level analysis in the main text in a new panel Figure 1d, pasted below:

Figure 1d. Correlation between log TC damage / GDP for each storm computed by Nordhaus (2010) and national average maximum wind speed (m/s) computed using LICRICE (1950-2005, N=89, slope=4.87, R-squared=0.36).

And have updated the main text to report this result:

Revised text from line 679-685:

“Iterations of the LICRICE model has been used to measure various social and economic impacts of TCs including direct deaths and damages (Hsiang & Narita 2012), changes to household income and expenditure (Anttila-Hughes & Hsiang 2013), infant mortality (Anttila-Hughes & Hsiang 2013), Gross Domestic Product growth (Hsiang 2010, Hsiang & Jina 2015), and depreciation (Hsiang & Jina 2014). We also compare our measure of wind speed against total direct national economic damages (normalized by GDP) from a limited sample of TCs estimated by Nordhaus 2010. We find that national wind speed exposure is a meaningful

predictor of total national damages (Figure 1b and Figure SI5), although we note that this outcome is highly uncertain and widely understood to be biased.”

We thank the reviewer for pushing us to investigate these issues, as we believe these new analyses substantially strengthen the manuscript.

11. Building up on comment #2: Assigning a monthly statewide average of TC wind speeds is a bit concerning for me, and I am worried that it may bias results quite a bit.

The reviewer’s concern was best addressed in an earlier analysis by Hsiang & Jina (NBER, 2014), which is pasted below (from their p. 16). The mathematical derivation by Hsiang & Jina is focused on the economic impact of storms in a country-level analysis, but it can be seamlessly applied to health impacts in our state-level analysis by re-assigning the meaning of $f(\cdot)$ to be a health production function and i to be states, rather than countries.

The main result of this derivation by Hsiang & Jina is that the type of regression results that we present in the current analysis are unbiased so long as there is no correlation between the average intensity of a storm and the likelihood that the most intense regions within that storm strike the most economically active / heavily populated / vulnerable pixels within a state. This condition would be violated if, for example, there were systematic patterns such that the eyes of a category 3 hurricanes tended to pass directly over dense cities but the eyes of category 2 hurricanes tended to miss cities. However, to the best of our knowledge, storm trajectories are governed by large scale environmental factors and phenomena that would violate this condition have never been observed.

Thus, to address the reviewer’s concern, we now point readers to the result by Hsiang & Jina (rather than repeating it in our text) and highlight the main conclusion and intuition. The methods section now explains:

Added text from line 695-709:

“Prior analysis by Hsiang & Jina (2014) demonstrated that spatial aggregates of TC exposure can be used as independent variables in a regression framework to obtain unbiased average effects that are expressed at finer spatial resolutions (footnote 13 on p. 16-17 in Hsiang & Jina (2014)). So long as there is no systematic correlation between the average intensity of a storm and the likelihood that the most intense regions within that storm strike the most populated (or economically active or vulnerable) pixels within a state, regression

coefficients will not be biased by spatial aggregations. This condition would be violated if, for example, there were systematic patterns such that the eyes of a Category 3 hurricanes tended to pass directly over dense cities but the eyes of Category 2 hurricanes tended to miss cities. However, given that the path of storms are primarily controlled by random steering winds at high altitude, interacting with the beta-effect induced by the Earth's meridional vorticity gradient, we have strong reason to believe that the spatial distribution of TC incidence within each state is orthogonal to the spatial distribution of underlying populations; and further that this covariance is independent of average TC intensity. To date, we know of no evidence that the trajectory of stronger (or weaker) storms systematically strike more vulnerable locations on land."

Cut and pasted material from Hsiang & Jina below:

¹³ Suppose pixels have heterogenous pre-storm capital K_p (capital could be physical, human, social, political, etc.) which has a long run production $f(K_p)$. Damage to this capital from a storm suffered at p is $D(S_p, K_p)$, a function of storm intensity S_p experienced at pixel p . Anttila-Hughes and Hsiang (2011) find $D(S_p, K_p) = \alpha K_p S_p$, where α is a constant describing the marginal fraction of capital that is destroyed by each additional unit of S_p . Thus, $\alpha S_p \in [0, 1]$ for observed values of S_p . We assume a similar linear form holds generally.

Long-run output lost to a storm is the difference between output with baseline capital when no storm occurs (our simple counterfactual here, but a trend could be accounted for) and output with storm-damaged capital, both summed over all pixels in country i :

$$lost_income_i = \sum_{p \in i} f(K_p) - \sum_{p \in i} f(K_p - \underbrace{\alpha K_p S_p}_{D(S_p, K_p)}).$$

If changes to the total capital stock from a single storm are modest relative to the curvature of $f(\cdot)$, by Taylor's theorem we can linearize $f(K_p - \alpha K_p S_p) \approx f(K_p) - f'(K_p)\alpha K_p S_p$ at each pixel. Letting $g(K_p) = f'(K_p)\alpha K_p$, we write

$$\begin{aligned} lost_income_i &\approx \sum_{p \in i} f(K_p) - \sum_{p \in i} (f(K_p) - f'(K_p)\alpha K_p S_p) \\ &= \sum_{p \in i} g(K_p) S_p \end{aligned}$$

Thus losses are roughly the inner product of storm intensity in each pixel and the marginal effect of storm intensity on production in each pixel, where the latter depends on both the capital density at p and the shape of the production function. Because we do not have observations of $g(K_p)$ for each pixel, we must find some way to estimate aggregate lost growth as a function of wind exposure. As in Equation 1 we denote area averages with a bar such that $\bar{x}_i = \sum_{p \in i} (x_p a_p) / \sum_{p \in i} a_p \approx \sum_p x_p / n_i$. The approximation holds if pixel areas do not vary substantially within a country, which is a reasonable approximation for almost all countries since pixel area is proportional to cosine of latitude and few countries exposed to tropical cyclones span large ranges of latitudes at high latitudes (where the derivative of cosine is large). Because there are many pixels in each country, we rewrite the sum of pixel impacts, i.e. the total lost income, in terms the average over pixels:

$$\begin{aligned} lost_income_i &\approx n_i \overline{(g(K_p) S_p)}_i \\ &= n_i \overline{g(K_p)}_i \bar{S}_i + n_i \text{Cov}_p(g(K_p), S_p) \end{aligned}$$

where the second term is the covariance across pixels between $g(K_p)$ and storm intensity for a specific cyclone event. Because the size of these terms scale with the size of a country n_i , we normalize by the initial size of the economy $n_i \overline{f(K_p)}_i$ so lost income is in terms of lost growth, a scale-invariant economic measure

$$lost_growth_i = \frac{lost_income_i}{initial_income_i} \approx \underbrace{\left(\frac{\overline{g(K_p)}_i}{\overline{f(K_p)}_i} \right)}_{\hat{\beta}} \bar{S}_i + \underbrace{\frac{\text{Cov}_p(g(K_p), S_p)}{\overline{f(K_p)}_i}}_{\varepsilon} \quad (\aleph)$$

where the coefficient of interest, labeled $\hat{\beta}$, does not scale with the size of the country n_i . The form of Equation \aleph is useful because it links a national summary statistic describing area-averaged cyclone exposure \bar{S}_p to a national summary statistic describing economic growth. The factor denoted $\hat{\beta}$ is the coefficient that we will attempt to measure empirically—it is the average marginal effect of cyclone exposure on long-run output in percentage terms. The form of Equation \aleph is what motivates us to use the spatial average of cyclone exposure across pixels to aggregate pixel-level cyclone exposure to the country-year level to match the units of observation in macro-economic data.

The term denoted ε is a residual that is likely mean zero—it is the covariance across pixels of cyclone exposure in a single storm and the marginal effect of cyclone exposure across pixels, normalized by total output of i . Importantly, it is a country-by-storm specific residual. The intuition behind this term is that sometimes a cyclone will cause unexpectedly large damages because the most intense part of the storm will pass directly over a location that has either a high capital density or a large sensitivity to cyclones (e.g. Hurricane Katrina), this will cause covariance between S_p and $g(K_p)$ to be positive and the lost growth from this event to be abnormally large relative to what we expect based on the average intensity of exposure \bar{S}_p . In other cases, the most intense part of a storm may pass over an uninhabited region, in which case this covariance will be negative and the lost growth will be abnormally low relative to expectation. On average across years, we assume ε is approximately zero because cyclone exposure within each storm is unlikely to be systematically correlated with economic activity on the ground.

Importantly, holding other factors constant, we will obtain an unbiased estimate of $\hat{\beta}$ if we estimate the expected value of Equation \aleph using observed values of \bar{S}_i so long as ε is not correlated with \bar{S}_i . Thus ordinary least squares will be

unbiased if

$$\text{Cov}_t \left(\frac{\text{Cov}_p(g(K_p), S_p)}{f(K_p)}, \bar{S}_i \right) = 0$$

where the outer covariance is across years (i.e. different storms). The intuition behind this condition is that Equation 8 is unbiased if there is no correlation between the average intensity of a storm (\bar{S}_i) and the likelihood that the most intense regions within that storm strike the most economically active (or vulnerable) pixels within a country ($\text{Cov}_p(g(K_p), S_p)$).

12. Line 645: Why not 1-18 and 19-44?

Our reply:

Unfortunately, this is a limitation of our data. Age groups 1-18 and 19-44 are readily available for more recent years but it was not possible to study the extended 20-year impact of TCs using these shorter time series. We discuss this limitation briefly on line 345-357. We have added additional text to clarify this point on line 733.

Revised text from line 733-735:

“To account for differences in underlying age-specific mortality we decomposed the effect of TCs on all cause mortality by four age groups in the data: less than 1, 1 - 44, 45-64, and 65-99 years old. We were limited to these age groups because these are the designations in our historical data. We computed mortality with respect to the underlying population by these same four age groups.”

13. Eqn 6: What is K?

Our reply:

K in equation 6 is the same as K in equation 1 and equation 3. It is the (unknown) total number of possible pathways by which TCs (or the intermediate variable x^k) influence mortality. This is documented when Equation 1 is introduced:

Text from line 769-770:

“Here, we let there be K pathways through which y is impacted by intermediary variables (x) and J ways through which determinants unrelated to TCs (w) impact y.”

14. Lines 744-746: Wouldn't that depend on storm intensity and characteristics and could thus vary across storms? Was this taken into account?

Our reply:

In principle, the distribution of impacts could vary enough that the assumption of additive separability is invalid for the strongest storms. However, even the strongest storms have an extremely small impact on mortality within any given moment in time,

such that we think additive separability is a robust approximation even in these cases. To address this concern, we have adjusted the text to include the fraction of deaths that occur for the highest impact storms, rather than just describing the average. The text now reads

Revised text from line 834-837:

“This method assumes that the overlapping responses influencing mortality at a moment in time are additively separable, an assumption that we think is reasonable given the overall small impact that any individual storm event has mortality rates at a moment in time in a particular region (0.019% on average, 0.04% for storms at the 95th percentile).”

15. Why use an OLS for rates and not a (quasi-)Poisson or negative binomial model?

Our reply:

To address this concern, we have rerun our analysis with a Poisson model and now include this result in the manuscript. We find that results using a Poisson model are nearly indistinguishable from results obtained using OLS. This result is now presented in Figure SI13a, pasted below (compare gray dashed line = Poisson model to maroon solid line = OLS unweighted and blue line = OLS population weighted). We now describe this result in the main text:

Added text from line 891:

“These results are unchanged if we use a Poisson regression specification (Figure SI13a).”

Figure SI13: Excess mortality following a tropical cyclone, adjusting for population, time trends, and average wind speed. (a-c) Estimated cumulative excess all cause mortality and 95% C.I. (a) Red = cumulative effect; blue line = cumulative effect using state population as

regression weights; green line = cumulative effect including regional month of sample fixed effects; gray dashed line = cumulative effect using Poisson regression.

16. Why were there not more lags of temperature considered?

Our reply:

Prior literature studying the effect of temperature on mortality has carefully examined the lag structure and determined that that mortality impacts dissipate within 30 days. Thus, to address the reviewer’s concern and clarify this for readers, we have updated the text to note this:

Added text line 857-859:

“In an effort to balance parsimony with model richness, we omit extended lags of temperature based on prior literature demonstrating that impacts on mortality dissipate within a month (Carleton et al. 2023, Gasperinin & Leone 2014).”

17. The temperature-mortality association is not really quadratic, though. Why not use a more flexible form (e.g., a 3- or 4-df natural spline)?

Our reply:

The reviewer is correct that there is a rich and complex literature on mortality and temperature, and there exist numerous possible ways that the association could be modeled statistically, ranging from the suggestions from the reviewer to far more complex functional forms that are also nonlinear functions of covariate information (e.g. Carleton et al., QJE 2022). However, the objective of our analysis is not to develop an advanced model of the temperature-mortality relationship, rather we simply account for the temperature flexibly to ensure its influence does not confound our estimates of the TC-mortality relationship. In reality, local exposure to historical TCs and local exposure to temperature is only very weakly correlated. Thus, we have found that inclusion of state-specific quadratic functions of temperature as a control variable is sufficient to render the remaining variation in residuals stationary and assignment of TC treatment to be as good as random. This is apparent in the randomization tests presented in Figures 1g and SI11 of the original manuscript, where our estimated effects of TCs are unbiased when TCs are randomized in various ways and temperature is accounted for nonlinearly. As noted in the original manuscript, failing to account for temperature causes some randomization tests to fail, however the approach in our model is flexible enough to be sufficient.

Nonetheless, to further address the reviewers concern and demonstrate that our approach to accounting for the confounding effects of temperature is sufficient, we have now added a new diagnostic Figure SI9 that presents the residuals from our main

specification, both the marginal distribution and the distribution over time. The distribution is stationary and essentially normal, with no evidence for seasonal or trend-related errors that would signify that the effects of temperature confound our estimates.

Figure S19: Distribution of the residuals from the main linear model. (a) Density plot of the residuals from the impulse response of mortality rates following state-level wind speed incidence. Gray line = normal distribution. (b) percentiles of residuals across each month between 1950 - 2015.

18. Eqn 8: What is the difference between m_t and t ?

Our reply:

t is an index for the month of sample, taking on values that range from 1 to 1,032. m_t is a vector of twelve indicators that signify the month of the year for an observation (e.g., an indicator variable for whether month = January). We have checked that these are accurately described in the text.

Revised text from line 844:

“ m_t is month-of-the-year dummy (e.g., an indicator variable for whether month = January),”

19. Line 786: What is h: nationwide monthly mortality rate?

Our reply:

h_t denotes a vector of 1032 month-of-sample indicator variables that each have an estimated coefficient. For example $h_t = 1$ if the month and year are January 1930. This control accounts for nation-wide events that could impact health, such as economic recessions, etc.

20. Lines 786-787: But these outbreaks tend to be more localized and not nationwide. Why not use regional terms instead?

Our reply:

Our approach towards modeling these large-scale shocks is similar to that of temperature, described above. Our current approach is sufficiently flexible to remove the confounding effects of these events, based on the same evidence presented above, but retains the parsimony of a model with limited parameters.

Nonetheless, to address this concern, we now also present results that include region-specific month-of-sample fixed effects for 3 regions (noting that our complete sample only contains the 36 eastern states that ever experience a TC) as a robustness check in Figure SI13a, shown above in response to an earlier comment. We have also added new text to describe this result in the main text:

Revised text from line 126-127:

“This result is robust to using population-based regression weights, count-based models, or accounting for region-by-month-of-sample shocks (Figure SI13a).”

And we provide additional interpretation and explanation in the Methods

Added text from line 893-897:

“In a robustness test, we interact the month-of-sample fixed effects with 3 region indicator variations. We continue to obtain our main findings after introducing these additional 2062 parameters to the model, although the estimates become much noisier and attenuate slightly. Both of these effects are well-understood results of including a large number of highly flexible variables that absorb a meaningful fraction of the true variation in the independent variable (Fisher et al. AER, 2012).”

Given the substantial costs of more than doubling the model's complexity without enriching the results, we elect to continue using our more parsimonious model (which passes all diagnostic tests) in our extended calculations.

21. Line 791: How was this R^2 estimated? Cross-validation? If not, it may be just due to overfitting.

Our reply:

The r-squared we report is an in-sample measure, however there remains a very large number of degrees of freedom (25062), indicating that the model is not overfit. We now clarify this in the text.

Revised text from line 887:

*“Overall, the fit for this model is high (in sample **adjusted $R^2 = 0.93$ with 25,062 degrees of freedom**).*”

22. Line 797: Is the period of 72 months prior to TC exposure included as a negative exposure control? Please clarify.

Our reply:

Thank you for this comment. The reviewer is correct that the leads are included as negative exposure controls, since we assume that future idiosyncratic TC incidence cannot affect current health outcomes. We had not used this term because we are more accustomed to the econometrics term “falsification tests”, but have now updated the manuscript so the terminology is accessible to both audiences. In fact, the randomization placebo tests we implement also would be considered negative exposure controls, so we have also updated the descriptions of those calculations as well. Thank you for making us aware of this helpful terminology.

The caption to Figure 1g now clarifies

*“(g) Verifying that the model produces unbiased estimate of TC impacts in four randomization-based placebo experiments (**negative exposure controls**, see Figure SI11).”*

And the caption to Figure 2b now clarifies

*“(b) Green=cumulative effect of raw coefficients in (a) with 95% C.I. shaded. **Leads (negative months) and orange lines are negative exposure controls.** Orange dashed lines are the mean effects in the four randomization-based placebo experiments. ”*

And the Methods section (line 847-849) now describes these leads as:

“We include 72 lead terms in equation 8 as a falsification test, also known as negative exposure controls, since idiosyncratic future TC incidence should not alter current health outcomes.”

And it describes the randomization tests as (line 925-928)

“Thus, these tests allow us to examine whether, in each case, the remaining structure generates artifacts in the model that would produce a spurious result, also known as a negative exposure control (Lipsitch et al 2011). Any non-zero correlation, on average, would indicate a biased model where the bias is driven by the non-randomized components of the original TC data.”

23. Lines 865-878: I am not sure I understand the purpose of this analysis or how it was actually conducted. How is this different from the association being nonlinear?

Our reply:

This analysis is meant to account for any adaptive measures that states that are frequently impacted by TCs may implement, resulting in a possibly lower response to TC incidence compared to states with less frequent exposure (and possibly less adaptation, therefore a higher average response). This is a stratification of the sample, which groups collections of states that are allowed to express different responses. Those responses could themselves be nonlinear (or not), as is the case with our preferred estimates. To address this concern, we now clarify this for readers in the the text, which now states:

Revised text from lines 983-988:

“To evaluate whether there is heterogeneity in the mortality response of states that are frequently exposed to TCs compared to those infrequently exposed, we stratify the sample by the average TC incidence they experience. We allow the mortality impulse-response to differ based on quartiles of states, sorted by their average TC incidence. We implement this by including an interaction with an indicator variable for the quartile of their average wind speed incidence, following the general approach for modeling adaptation developed in refs [41] and [45].”

24. It is not clear to me if there is statistical interaction by quartile of incidence or not in the end.

Our reply:

Yes, there is an interaction with dummy variables that are assigned based on the quartile. We hope the revised text in response to the comment above now clarifies this.

25. Eqn 9: Why cubic and not a more flexible form (e.g., natural or penalized splines)?

Our reply:

We have explored multiple functional forms and found that the results are invariant to this particular choice, so long as the model is sufficiently flexible. To address this concern and demonstrate this point for readers, we have revised Figure SI15 (pasted below) to now also include a restricted cubic spline specification (dashed lines), which is statistically indistinguishable from the cubic specification

Figure SI15: Excess mortality as a cubic function of wind speed. ... Dashed lines are response functions from a cubic spline version of the model.

And the text now states:

Revised text from lines 1012-1014:

“Figure SI15 displays the cumulative impact estimated using both the linear and nonlinear models after 180 months. These results are unchanged if we alternatively use a cubic spline regression specification (Figure SI15).”

We chose to present the cubic model as our main model in the manuscript because it recovered the same results but also has a closed form expression that is simple to communicate for future modeling analyses.

Reviewer Reports on the First Revision:

Referees' comments:

Referee #2 (Remarks to the Author):

I have no further comments for the author. Many thanks for thoroughly attempting to address my comments and those of the other reviewers. I think that the additional analyses that were undertaken are a worthwhile addition to the paper, and I appreciate the many editorial changes that were made in response to our comments, including the acknowledgment of some of the limitations of your data sets.

Referee #3 (Remarks to the Author):

I thank the authors for very carefully addressing my previous comments. I only have two follow-up comments.

1. (on my previous comment #6): so there are no other high-wind-speed events experienced by any of the states during the study period? For example tornadoes? In some states, tornadoes may be more frequent than TCs....

2. (on my previous comment #11): the spatial distribution of TC incidence is not the same as the spatial distribution of Category 2 vs Category 3 hurricane. And in fact, there are established spatial patterns in population density across states that may positively or negatively correlate to TC incidence. Therefore, I think this is a very strong assumption with large probability of violation.

Referee #4 (Remarks to the Author):

I was asked by the editor to provide an additional review of this paper, focusing on the meteorological aspects. As the authors are well aware, the issue here is how well correlated overall damages and mortality are with wind speed, which (in addition to surface pressure) is by the far the best measured meteorological variable quantity in landfalling hurricanes. In response to one of the reviewers, the authors attempted a correlation between hurricane wind and rain, but this is very tricky because rain data are quite spotty and, in addition, it is really flood levels we are interested in.

What is worrying here is that water kills far more people (at least directly) than wind. Rappaport (2014) found that wind is responsible for only 8% of fatalities in U.S. hurricanes between 1963 and 2012, with most of the rest coming from water. (Note: I am not Rappaport.) Not sure about damage, but one could at least try to compare NFIP payouts to those by private insurers; if the former are large compared to the latter, then it makes we wonder whether direct deaths might be a better predictor of indirect deaths. I confess I did not read the paper carefully enough to see if the authors tried that, but I think not.

Aside from the possibility of using direct deaths as a measure of storm severity, I think the authors have done the best they can with the available data, and the conclusions are important and far-reaching. But the relatively poor correlation (poor in my book) between rain and wind speed is a matter of concern (by the way, what is the p value?) and I only wish the authors would state this more strongly as a caveat.

Reference:

Rappaport, E. N., 2014: Fatalities in the United States from Atlantic tropical cyclones: New data and interpretation. *Bulletin of the American Meteorological Society*, 95, 341–346, <https://doi.org/10.1175/BAMS-D-12-00074.1>.

Author Rebuttals to First Revision:

Reviewer comments in black

Our replies in blue

New text in the manuscript in red

Referee #2 (Remarks to the Author):

I have no further comments for the author. Many thanks for thoroughly attempting to address my comments and those of the other reviewers. I think that the additional analyses that were undertaken are a worthwhile addition to the paper, and I appreciate the many editorial changes that were made in response to our comments, including the acknowledgment of some of the limitations of your data sets.

Our reply:

We thank the reviewer for their thoughtful evaluation of our analysis and revisions to the manuscript. We feel that the review process has been extremely productive and we believe it has made the manuscript much stronger.

Reviewer comments in black

Our replies in blue

New text in the manuscript in red

Referee #3 (Remarks to the Author):

I thank the authors for very carefully addressing my previous comments. I only have two follow-up comments.

1. (on my previous comment #6): so there are no other high-wind-speed events experienced by any of the states during the study period? For example tornadoes? In some states, tornadoes may be more frequent than TCs....

Our reply:

We thank the reviewer for this clarification to their previous comment. While we agree that states experience high gusts of wind speeds from tornadoes (and other high wind events), due to the small spatial footprint of tornadoes, they result in extremely small values for monthly average wind speed. These values are so small that they are dwarfed in comparison to the state average wind speed from TCs, even in states with low average TC incidence. Absent a TC, states do not experience monthly average winds >6.9 m/sec.

To specifically compare TC vs. tornado incidence, we have estimated the state average wind speed from all tornadoes between 1950-2022 using data from the NOAA National Weather service. To compare, the minimum non-zero value of monthly average TC incidence in our sample is $3.34e-04$ m/s and the 1st percentile value is $8.4e-03$ m/s. In contrast, the maximum monthly state averaged wind speed from tornadoes is $9.6e-04$ m/s, the 99th percentile is $3.8e-05$ m/s, and the median is $6.3e-08$ m/s. Thus, the maximum tornado events are comparable to the smallest (non-zero) TC events.

We have also examined the highest wind speeds from non-tornado and non-TC “Wind/Hail” events between 1955-2022, again according to NOAA. The maximum monthly average from wind/hail events is $1.1e-03$ m/s and the 99th percentile is $1.3e-04$ m/s. Thus, again, the maximum incidence from these events is below the first percentile of (non-zero) TC events.

We have now added text to our Data section, where wind speed data is presented, to document these comparisons. We hope this may clarify these points for readers who may have similar questions. The updated text now reads:

Added text line 705:

“We note that TC events generate the highest average state wind speeds compared to wind speeds from other intense storm phenomena, such as tornadoes. For reference, the minimum monthly state average wind speed we compute from TCs in our sample is $3.34e-04$ m/s and the 1st percentile is $8.4e-03$ m/s. In contrast, the maximum monthly state average wind speeds from tornadoes in CONUS between 1950-2022 is $9.6e-04$ m/s and the 99th percentile is $3.8e-05$ m/s, and for non-TC and non-tornado wind/hail events the maximum is $1.1e-03$ m/s and the 99th percentile is $1.3e-04$ m/s. Therefore, the maximum non-TC wind events are comparable to the minimum (non-zero) TC events; and absent a TC, states do not experience average wind speeds of a similar magnitude as those from TCs.”

2. (on my previous comment #11): the spatial distribution of TC incidence is not the same as the spatial distribution of Category 2 vs Category 3 hurricane. And in fact, there are established spatial patterns in population density across states that may positively or negatively correlate to TC incidence. Therefore, I think this is a very strong assumption with large probability of violation.

Our reply:

We thank the reviewer for this clarification to their previous comment. To address this concern, we have now developed a direct test to understand whether the spatial correlation between average TC incidence and population density affects our results. We re-estimate our results, but stratify our sample based on the within-state spatial correlation between average TC incidence and the average distribution of the population. To do this, we first compute the spatial correlation between population and wind speed across 0.1×0.1 degree pixels within each state. We then stratify our sample into two groups, above and below the median spatial correlation. Finally, we evaluate whether the mortality-response to TCs differs across these groups.

We find no systematic pattern in the mortality response across these strata that is distinguishable from statistical noise. We find that the impact of TCs on mortality is nearly indistinguishable between these two strata for the first 96 months (8 years) after a TC. After 96 months, the central estimates diverge somewhat but they are not statistically different from one another, and neither is statistically different from the pooled main estimate we present. This leads us to conclude that our results (both average effects and trends) are unlikely to be driven or biased by spatial correlations between average TC incidence and population density within states.

We have updated the text to present these results in Figure SI13e, pasted below:

“Figure SI13: (e) within state correlation between population density and wind speed incidence: light orange below median and dark orange above median. Solid black line = cumulative effect and 95% C.I. shaded.”

This new test and the results are now described in the main text:

Added Text from line 193:

“Lastly, we evaluate whether the overall spatial correlation between populations and average wind speed incidence, within each state, alters the mortality response to TCs. Stratifying states based on within-state spatial correlation across 0.1x0.1 degree pixels, we fail to find evidence that states with higher spatial correlations are systematically different from states with little or negative correlations (Figure SI13e).”

We thank the reviewer for their thoughtful evaluation of our analysis and revisions to the manuscript. We feel that the review process has been extremely productive and we believe it has made the manuscript much stronger.

Reviewer comments in black

Our replies in blue

New text in the manuscript in red

Referee #4 (Remarks to the Author):

I was asked by the editor to provide an additional review of this paper, focusing on the meteorological aspects. As the authors are well aware, the issue here is how well correlated overall damages and mortality are with wind speed, which (in addition to surface pressure) is by the far the best measured meteorological variable quantity in landfalling hurricanes. In response to one of the reviewers, the authors attempted a correlation between hurricane wind and rain, but this is very tricky because rain data are quite spotty and, in addition, it is really flood levels we are interested in.

What is worrying here is that water kills far more people (at least directly) than wind. Rappaport (2014) found that wind is responsible for only 8% of fatalities in U.S. hurricanes between 1963 and 2012, with most of the rest coming from water. (Note: I am not Rappaport.)

Our reply:

We appreciate the additional information about the drivers of direct deaths from TCs. We have revised the manuscript to include additional points along the lines suggested by the reviewer and added a citation to the Rappaport paper. The role of flooding for direct deaths is now described in the introduction

Revised line 69:

Ground-level wind speed exposure is not the only dimension of TCs that impact human outcomes, for example, flooding is a large driver of direct deaths from TCs (Rappaport 2014), however wind speed is the only metric that can be consistently reconstructed for all storms going back throughout our entire sample, to the best of our knowledge.

And the omission of rainfall and flooding variables are now explicitly described in the Discussion caveat that originally discussed weaknesses in the meteorological data:

Revised line 376:

However, storm surges, rainfall, or flooding have important differential effects on direct deaths (Rappaport 2014) and accounting for these effects explicitly would further improve our understanding of TC impacts. If rainfall or flooding that is

uncorrelated with wind speeds causes long-run mortality, the estimates presented here may underestimate the total mortality impacts from TCs.

Not sure about damage, but one could at least try to compare NFIP payouts to those by private insurers; if the former are large compared to the latter, then it makes we wonder whether direct deaths might be a better predictor of indirect deaths. I confess I did not read the paper carefully enough to see if the authors tried that, but I think not.

Our reply:

We appreciate the reviewer's suggestion, but note that it is notoriously challenging to systematically evaluate NFIP payouts and private insurance payouts on comparable footing. FEMA has recently made redacted NFIP data public, which we are analyzing in another major research project, but it is not a trivial task and goes far beyond the scope of this analysis (e.g. it requires years of data cleaning). In addition, there is no public aggregator for private insurance claims or payouts, so there is no definitive record of these numbers that is comprehensive for our sample.

Nonetheless, it is possible to examine some very limited publicly available data to consider estimates of insured losses and NFIP insured losses, as suggested by the reviewer, for a few recent hurricanes. The Insurance Information Institute summarizes non-NFIP insured losses from several of the highest damage Hurricanes, which we tabulate below along with national NFIP data. This sample is extremely small, but we think it addresses the basic question posed by the reviewer. In these six events, NFIP damages range from <1% to 30% of insured losses, averaging 14% across the six events. These quantities are qualitatively different from the 92% of direct mortality caused by flooding that Rappaport (2014) reports. Further, this does not indicate that NFIP damages are generally large compared to non-flood damages, although they can be substantial in some events. Thus, we do not think that these numbers would justify a switch to using direct deaths as the independent variable in our analysis.

Date	Hurricane	Billion dollars		NFIP losses as a percent of insured losses
		Insured losses (non-NFIP)	NFIP insured losses	
Aug. 25-30, 2005	Katrina	\$ 41.10	\$ 16.10	28%
Aug. 24-26, 1992	Andrew	\$ 15.50	\$ 0.17	<1%
Oct. 22-29, 2012	Sandy	\$ 18.75	\$ 8.10	30%
Sep. 12-14, 2008	Ike	\$ 12.50	\$ 2.18	15%
Oct. 24, 2005	Wilma	\$ 10.30	\$ 0.36	3%
Sep. 17-22, 1989	Hugo	\$ 4.20	\$ 0.38	8%

Source: Insured losses from Insurance Information Institute (2013) and NFIP losses from OpenFEMA dataset (2022).

In addition to the limited evidence in the table above, it is worth noting that direct damages are notoriously poorly documented. Official estimates generally only included insured losses, however insurance penetration is not well known and it is widely assumed that roughly half of damages are probably uninsured. Thus, comparisons of NFIP to other damages is a comparison of two incomplete numbers where the missing (uninsured) components are unknown; and to the best of our knowledge, there currently are no reliable techniques to bound these values. One standard practice, which we find deeply unsatisfying, is to simply multiply insured losses by a factor of two to estimate total losses (Nordhaus, *Climate Change Econ.*, 2010). In addition, Corelogic recently estimated losses from Hurricane Ian and reported that uninsured losses were roughly 30% of total losses. Moreover, unearned income is a form of indirect economic damage that is not reported in direct damage estimates, but it is substantially larger than direct damages (Deryugina, *AEJ Econ Policy*, 2015; Hsiang & Jina, *NBER* 2014). Thus, while we could, in principle, attempt to compare the ratio of NFIP to non-NFIP insurance payouts for a larger number of storms, we believe they would still not be sufficiently reliable to be meaningful.

Even though NFIP damages do not appear to be a large fraction of total insured damage, one could, in principle, still use direct deaths (or damages) instead of maximum wind speed as the independent variable in our analysis (as suggested by the reviewer as an additional check). However, there are two reasons why presentation of such a calculation would be uninformative, to the point of being misleading, regardless

of the result. Thus, we respectfully refrain from including this calculation in the manuscript.

First, we have no evidence to believe that long-term indirect mortality from TCs is related to or proportional to direct mortality, and we have reason to believe that they are fundamentally different. In our Discussion, we hypothesize multiple possible pathways that TCs could generate indirect mortality, including changing social networks, economic impacts, changing disease ecology, state and local fiscal adjustments, and heightened physical and mental stress; all described in the Discussion of our manuscript. But there is nothing in our theory to suggest that direct deaths should predict indirect deaths (e.g. See Eq. 3). Yet there are many reasons to think that direct deaths should *not* predict indirect deaths. Specifically, we allow for there to be K channels of influence (denoted by \mathbf{x} in Eq. 3; which we are not required to know or identify explicitly) that link TCs to indirect mortality but which are not reflected in direct mortality (i.e. direct mortality is not impacted via \mathbf{x}). For example, changes to social networks a year after a TC could contribute to indirect mortality, but is unlikely related to direct mortality during the TC itself.

Second, there are critical statistical reasons why using direct deaths as a measure of storm incidence (instead of wind speed) would result in a biased estimate of the relationship between TCs and long-run mortality in a regression framework. Using direct deaths from TCs to measure TC incidence violates the “unconfoundedness assumption”, a necessary condition for estimating unbiased causal effects (e.g. see Imbens & Rubin, 2015), because it is not randomly assigned and is likely correlated with both unmeasured variables and potential outcomes. Specifically, the unconfoundedness assumption requires that our measure of hurricane incidence be plausibly exogenous and random. This means that the treatment assignment (i.e., whether a location is impacted by a TC) is conditionally independent (or orthogonal) of the potential outcomes (i.e., long-run mortality in a world where a state was impacted by a TC and in a world where a state was not impacted by the TC). Stated another way, it must be the case that populations that generate the outcome (here, indirect mortality) cannot intentionally affect or control the independent variable (e.g. direct mortality). The challenge with using direct mortality as the independent variable is that it clearly is affected by many confounding variables that are controlled by the subject population (e.g. early warning systems, healthcare, economic inequality).

A classic example of this assumption being violated is to examine whether some optional “healthy” activity, such as getting more exercise, causes individual life expectancy to increase. A challenge with studying this question in observational data is that individuals who exercise more also probably engage in other healthy non-exercise

activities that also increase their life expectancy, thereby confounding an analysis of the relationship between exercise and life-expectancy. Individuals that voluntarily exercise are not comparable to individuals that do not. In our context, the concern is that populations that exhibit high direct mortality from TCs conditional on wind speed (e.g. the Ninth Ward of New Orleans), are fundamentally different from populations that exhibit low direct mortality for similar wind speeds (e.g. Manhattan, NYC), and the underlying factors that differentiate these responses (e.g. healthcare, racial inequality, infrastructure) might also confound the indirect mortality responses that we measure. For example, states can anticipate TC incidence through meteorological forecasting and implement emergency management protocols which change the counts of direct deaths from evacuations. We may also think that states with more effective emergency response systems may also be more equipped across a number of other healthcare settings in the years after a TC, which would bias our estimate.

In econometrics, this type of violation of the unconfoundedness assumption is known as “selection bias” and would, in this context, be described as caused by “selection into treatment” (e.g. see Angrist & Pischke, 2009), since here the “treatment variable” (direct deaths) would be an outcome that is influenced by confounding variables. To avoid this issue in the analysis of environmental impacts, it is now generally understood that outcomes should be conditioned on physical measurements (here, wind speed) rather than socially-generated outcomes (deaths) from each event (e.g. see Hsiang, *Annual Review of Res. Econ.*, 2016). This is the primary reason that we use wind speed as the measure of TC incidence. Even though, as the reviewer notes, wind speed incidence does not fully capture the complex variation in storm surges, rainfall, or flooding that occurs within storms; it is sufficiently exogenous that we can obtain unbiased estimates of long term mortality impacts.

Thus, to respond to the reviewer’s request, we have now revised the methods section of the manuscript to briefly explain that using direct deaths as a measure of storm incidence would violate the unconfoundedness assumption, and we point readers to additional literature that discusses this point more fully:

Added Text Line 841:

Econometric implementation

Identification

Our econometric analysis exploits the quasi-random variation in the location and intensity of TC incidence to estimate the impact of TCs on mortality separately from other known and unknown factors that affect mortality across locations and

over time....Because the location, timing and intensity of TC incidence is determined by oceanic and atmospheric conditions that are beyond the control of individual states, we assume mortality TC incidence is as good as randomly assigned (Angrist & Pischke, 2009).

We note that some early analyses of natural disaster impacts utilized social outcomes (e.g. direct economic damage or direct mortality), as a proxy measure of physical hazard severity. However, it is now understood that use of these metrics as independent variables may confound estimated treatment effects, since they are endogenously determined by many of the same underlying covariates (e.g. healthcare, infrastructure, inequality, institutions) that mediate other outcomes from disasters (Kahn, 2005; Hsiang & Narita, 2011; Carleton et al 2022). Thus, use of these proxy measures for hazard severity exposes analyses to selection biases, since population characteristics may cause observational units to “select” into more or less severe treatment (Angrist & Pischke, 2009). We therefore focus this analysis strictly on independent variables that are physical measures of TC incidence (wind speed), because they are exogenous and cannot be influenced by the populations that are impacted (Hsiang, 2016).

Aside from the possibility of using direct deaths as a measure of storm severity, I think the authors have done the best they can with the available data, and the conclusions are important and far-reaching. But the relatively poor correlation (poor in my book) between rain and wind speed is a matter of concern (by the way, what is the p value?) and I only wish the authors would state this more strongly as a caveat.

Our reply:

We find that state average wind speed and rainfall have a highly statistically significant correlation (p-values for each of the 6 storm examples in Figure SI6 are all ≤ 0.001 and the p-value of the correlation across all storms is < 0.001). We have now added this to the text:

Revised line 678:

For example, we find that LICRICE wind speed and NCEP rainfall from TCs are correlated within storms at the pixel level (Figure SI6) and at the state-by-storm level (r -squared = 0.31 and $p < 0.001$, Figure 1e) for a limited sample of storms where granular rainfall data is available.

In addition, we have now expanded our original (first) caveat on the challenge of missing rainfall and flood data to now also describe the implication of the imperfect correlation between rainfall/flooding and wind speed for our results. The Discussion now reads (pasted again here):

Revised line 376:

Our analysis faces several limitations due to our data that should be improved upon in future work. First, we focus on measuring incidence to TCs by reconstructing wind incidence, but we do not explicitly model or otherwise parameterize storm surges, rainfall, or flooding that occurs within storms. The impact of these other variables are likely represented in our estimates to the extent that they are correlated with the maximum winds experienced at each location (Figure 1e and Figure SI6). However, storm surges, rainfall, or flooding have important differential effects on direct deaths (Rappaport 2014) and accounting for these effects explicitly would further improve our understanding of TC impacts. If rainfall or flooding that is uncorrelated with wind speeds causes long-run mortality, the estimates presented here may underestimate the total mortality impacts from TCs.

We thank the reviewer for their thoughtful evaluation of our analysis and revisions to the manuscript. We feel that the review process has been extremely productive and we believe it has made the manuscript much stronger.

Reviewer Reports on the Second Revision:

Referees' comments:

Referee #4 (Remarks to the Author):

I very much appreciate the exceptionally thorough response to my review. For a meteorologist with little experience in economics, it has been a great educational opportunity. I have no further concerns and recommend that the paper be published in its present form.